# Macrophage deletion of Noc4l triggers endosomal TLR4/TRIF signal and leads to insulin resistance

Yongli Qin[1,15], Lina Jia[1,15], Huijiao Liu[1,15], Wenqiang Ma[1,15], Xinmin Ren[1], Haifeng Li[1], Yuanwu Liu[1], Haiwen Li[2], Shuoqian Ma[1], Mei Liu[3], Pingping Li[4], Jinghua Yan[5], Jiyan Zhang[6], Yangdong Guo[7], Hua You[8], Yan Guo[1], Nafis A. Rahman[9,10], Sławomir Wołczyński[10], Adam Kretowski[11], Dangsheng Li[12], Xiru Li[13], Fazheng Ren[14] & Xiangdong Li[1,10 ✉]

In obesity, macrophages drive a low-grade systemic inflammation (LSI) and insulin resistance (IR). The ribosome biosynthesis protein NOC4 (NOC4) mediates 40 S ribosomal subunits synthesis in yeast. Hereby, we reported an unexpected location and function of NOC4L, which was preferentially expressed in human and mouse macrophages. NOC4L was decreased in both obese human and mice. The macrophage-specific deletion of Noc4l in mice displayed IR and LSI. Conversely, Noc4l overexpression by lentivirus treatment and trans-genic mouse model improved glucose metabolism in mice. Importantly, we found that Noc4l can interact with TLR4 to inhibit its endocytosis and block the TRIF pathway, thereafter ameliorated LSI and IR in mice.

[1] State Key Laboratory of Agrobiotechnology, College of Biological Sciences, China Agricultural University, Beijing, China. [2] Agricultural Research Station, College of Agriculture, Virginia State University, Petersburg, VA, USA. [3] Department of Pathology, Chinese PLA General Hospital, Beijing, China. [4] Academy of Medical Sciences & Peking Union, Medical College, Beijing, China. [5] CAS Key Laboratory of Pathogenic Microbiology and Immunology, Institute of Microbiology, Chinese Academy of Sciences, Beijing, China. [6] Department of Molecular Immunology, Institute of Basic Medical Sciences, Beijing, China. [7] State Key Laboratory of the Agro-Biotechnology, College of Horticultural Science, China Agricultural University, Beijing, China. [8] Affiliated Cancer Hospital & Institute of Guangzhou Medical University, Guangzhou, China. [9] Department of Physiology, Institute of Biomedicine, University of Turku, Turku, Finland. [10] Department of Reproduction and Gynecological Endocrinology, Medical University of Bialystok, Bialystok, Poland. [11] Department of Endocrinology, Diabetology, and Internal Medicine, Medical University of Bialystok, Bialystok, Poland. [12] Shanghai Institutes for Biological Sciences, Chinese Academy of Sciences, Shanghai, China. [13] Department of Surgery, Chinese PLA General Hospital, Beijing, China. [14] Department of Nutrition and Human Health, China Agricultural University, Beijing, China. [15]These authors contributed equally: Yongli Qin, Lina Jia, Huijiao Liu, Wenqiang Ma. ✉email: xiangdongli@cau.edu.cn

Obesity and metabolic syndrome, including insulin resistance (IR), type 2 diabetes (T2D), fatty liver disease, and atherosclerosis currently present a major threat to global health[1,2]. A growing number of studies suggest that in obesity, there is an increased accumulation of proinflammatory macrophages in insulin target tissues, including adipose[3], muscle[4], and liver[5], thus driving low-grade chronic systemic inflammation (LSI)[6,7]. The increased production of proinflammatory cytokines, such as TNFα and IL-6, was linked to IR via serine phosphorylation of IRS-1[3,8–10]. However, clinical studies have found that the inhibition of TNFα or IL-6 to improve insulin sensitivity exert insignificant effects[11,12]. Thus, factors that regulate inflammatory response-induced IR require to be further studied.

Nucleolar complex associated 4 homolog (NOC4L), also known as NOC4, is a homolog of yeast Noc4p. Noc4p and Nop14p formed a complex, which was mainly involved in the assembling and transporting of ribosome 40 S subunit in yeast[13,14]. The normal tissue RNA-seq data mined from the public database showed that NOC4L is highly expressed in testis, fat and immune organs of human[15]. FACS-based full-length transcript analysis showed that Noc4l is highly expressed in marrows[16]. We have demonstrated that deletion of Noc4l leads to embryonic lethality in mice[17]. A recent study demonstrated the critical role of Noc4l in activation of regulatory and conventional T cells (Tregs and Tconvs). Noc4l-deficient T cells have a smaller 40 S peak, which is involved in selectively controlling protein translation in Tregs and Tconvs[18].

In the current study, we found that NOC4L expression was decreased in both obese subjects and mice. By using lentivirus-Noc4l (Lv-Noc4l), myeloid-specific deletion, and overexpression of Noc4l transgenic mouse models, as well as experiments in vitro, we clarified an unexpected function of NOC4L, in inhibiting TLR4 internalization and the TLR4/TRIF pathway in endosomes of macrophages involved in LSI and IR. We further found that overexpression of Noc4l can prevent LSI and IR induced by high-fat diet (HFD).

## Results

### NOC4L decreased in obese subjects and mice and Lv-Noc4l treatment could improve IR and inflammation in mice.
Noc4l was preferentially expressed in the testis, lung, white adipose tissue (WAT), and some immune organs of the mice (Fig. 1a). The high expression of Noc4l in WAT prompted us to explore the functional roles of Noc4l within the adipose tissue.

Adipose tissue plays a vital role in regulating energy balance and metabolism and its dysfunction is closely associated with metabolic diseases such as obesity, IR and LSI[19,20]. To assess the potential relevance of NOC4L in WAT, NOC4L mRNA and protein were quantified in WAT of diet-induced obese (DIO) mice and genetically diabetic mice (db/db), as well as obese humans. The expression of NOC4L was decreased in WAT in obese mice and humans (Fig. 1b–e).

To investigate the physiological role of Noc4l in vivo, Noc4l was overexpressed in DIO mice by tail intravenous injection (IV) of Lv-Noc4l. The expression of Noc4l in liver and WAT were increased after 3 weeks of injection of Lv-Noc4l (Fig. 1f, g). Lv-Noc4l led to decreased glucose intolerance and improved IR (Fig. 1h–i) and reduced total fat, including epididymal WAT (eWAT), inguinal WAT (iWAT), and perinephric fat (pWAT) weight (Fig. 1j–l). The expression of Noc4l was no difference between these adipose tissues (Supplementary Fig. 1a). Symptoms of reduced fatty liver in the mice injected with Lv-Noc4l were evident (Fig. 1m, n).

In addition, we mined the expression of Noc4l from ATTIE LAB DIABETES DATABASE, which is an interactive database of gene expression and diabetes-related clinical phenotypes (http://diabetes.wisc.edu/index.php). This database allows researchers to retrieval genes within a particular tissue that correlate with specific diabetes-related clinical traits. The glucose level was identified as the factor most negatively correlated to Noc4l transcripts vs. all diabetes-related clinical traits (Fig. 1o). The combined results showed that Noc4l could improve IR in vivo and may be a candidate target for IR treatment.

### Noc4l was mainly expressed in both mouse and human adipose tissue macrophages and regulated by Lipopolysaccharide (LPS) and Palmitic Acid (PA).
For further experiments, mouse monoclonal antibody (3L7) and rabbit polyclonal antibody (6 R) were prepared. To confirm the specificity of NOC4L antibodies prepared in this study, we used NOC4L-Flag vector to overexpress NOC4L and detected the expression of NOC4L. As mentioned below, Noc4l expression was further detected in Noc4l-ablated bone-marrow-derived-macrophages (BMDMs). The results confirmed the specificity of NOC4L antibodies (Supplementary Fig. 1b, d, e). The localization of NOC4L was assessed in mouse and human WAT. Double immuno-fluorescence analyses of NOC4L and macrophage markers (F4/80 or Mac-2) showed that NOC4L co-localized with F4/80 and Mac-2, which indicated that NOC4L was mainly expressed in both mouse and human adipose tissue macrophages (ATMs) (Fig. 2a). We also analyzed the mRNA expression of Noc4l in the adipocyte and stromal vascular fractions (SVFs) of visceral adipose tissue (VAT). Consistent with the immunofluorescence results, Noc4l was highly expressed in SVFs of mice (Supplementary Fig. 1c).

Whether Noc4l in macrophages plays an important role in obesity-associated IR and inflammation remains undetermined. To address this question, we evaluated Noc4l expression in macrophages upon treatment with LPS and PA, which are related to obesity-induced inflammation and IR[21–24]. LPS treatment could reduce Noc4l mRNA expression in a time-dependent manner and PA treatment also reduced Noc4l mRNA expression in a dose-dependent manner in the murine macrophage cell line RAW264.7 (Fig. 2b, c). Consistent with the results of mRNA expression, LPS treatment reduced the Noc4l protein level in a time- and dose-dependent manner in RAW264.7 cells (Fig. 2d, e). Similarly, treatment with PA reduced the Noc4l protein levels in a dose-dependent manner in RAW264.7 cells (Fig. 2f). We also detected reduced NOC4L protein expression in macrophages derived from THP1 (a human monocyte cell line) after LPS treatment (Fig. 2g).

### Deletion of Noc4l in macrophages affected insulin sensitivity and energy metabolism.
To further investigate the physiological role of Noc4l in inflammation and IR in vivo, Noc4l[LKO] mice whose Noc4l was genetically and selectively inactivated in myeloid cells were prepared for further experiments. Specifically, Noc4l[LKO] mice were prepared using Noc4l[fl/fl] mice[17] crossed with lysozyme M (LysM) cyclization recombinase (cre) mice. Noc4l was effectively and selectively deleted from BMDMs both at the mRNA and protein levels (Supplementary Fig. 1d, e). The expression of Noc4l was also decreased in ATMs of Noc4l[LKO] mice (Supplementary Fig. 1f). Histopathological analyses of kidney, liver, lung, spleen, and eWAT revealed no abnormalities in mice aged 2 months (Supplementary Fig. 1g). Macrophage-specific deletion of Noc4l exerted no effect on the body weight of the chow diet (CD) mice (Supplementary Fig. 1h). However, the Noc4l[LKO] mice gained more weight compared with the age-matched HFD control (Fig. 3a). Consistent with the results for body weight, the epididymal fat mass of HFD-fed Noc4l[LKO] mice was considerably higher than that of the age-matched Noc4l[fl/fl]

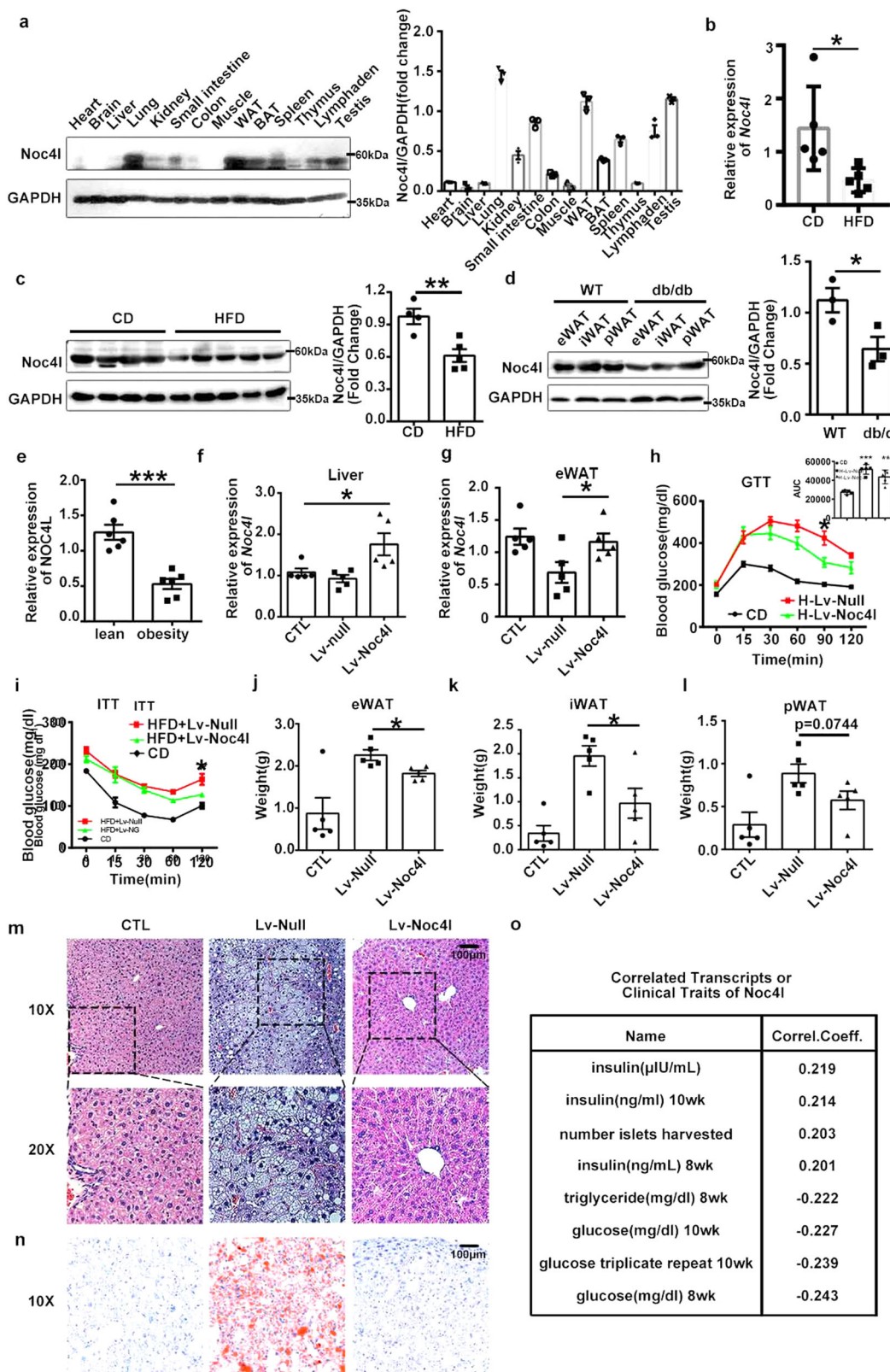

mice (Supplementary Fig. 1i). In addition, the weights of liver and spleen have no difference between Noc4l$^{LKO}$ and Noc4l$^{fl/fl}$ mice (Supplementary Fig. 1j). Fasting blood glucose concentration was considerably higher in the HFD-fed Noc4l$^{LKO}$ mice, whereas no difference was observed between the two groups with CD (Fig. 3b). Moreover, the serum concentrations of insulin, free fatty acids (FFAs), triglycerides (TGs), and cholesterol (CHOL)

were significantly increased in the HFD-fed Noc4l$^{LKO}$ mice (Fig. 3c). Accordingly, Noc4l deletion in macrophages led to metabolic abnormalities in mice. Histopathological analysis of pancreas and the insulin secretion of the islet were no different between the Noc4l$^{fl/fl}$ and Noc4l$^{LKO}$ mice (Supplementary Fig. 1k, l), indicating that Noc4l exerted no effect on the function of the pancreas.

**Fig. 1 NOC4L was lower in obese subjects and mice and Lv-Noc4l treatment could alleviate IR on mice. a** Protein levels of Noc4l (AV41005, Sigma) in various tissues from C57/BL6 mice aged 8 weeks. Quantification of Noc4l/GAPDH ratio was shown in right panel. Data are representative of three independent experiments. **b** Relative mRNA expression of Noc4l in epididymymal WAT (eWAT) from mice on chow diet (CD) or high fat diet (HFD) for 12 weeks ($n = 5$ per group). $*p = 0.0287$. **c** Protein expression of Noc4l in eWAT from both lean and obese mice. Mice were fed with CD ($n = 4$) or HFD for 12 weeks ($n = 5$). Quantification of Noc4l/GAPDH ratio was shown in right panel. $**p = 0.0055$. **d** Protein expression of Noc4l in eWAT, inguinal WAT (iWAT) and perinephric fat (pWAT) from C57/BL6 and db/db mice. Quantification of Noc4l/GAPDH ratio in total fat tissue was shown in right panel. Data are representative of independent three experiments. $*p = 0.0469$. **e** Quantitative analysis of the mRNA expression levels of *NOC4L* in fat tissues from lean and obese patients, measured by qRT-PCR ($n = 6$ per group). $***p = 0.0002$. **f–g** Relative mRNA expression levels of *Noc4l* in liver (**f**) and eWAT (**g**) from lean and obese mice treated with lentivirus-null (Lv-null) or lentivirus-Noc4l (Lv-Noc4l) ($n = 5$ per group). $*p = 0.0191$ (**f**); $*p = 0.0497$ **g**. **h–i** GTT (**h**) and ITT (**i**) of lean and obese mice treated with Lv-Noc4l or Lv-null after 3 or 4 weeks. Mice were fed with CD or HFD for 20 weeks before treated ($n = 5$ per group). Quantification of the area under curve (AUC) of GTT was shown in top-right panel. $*p = 0.0202$, $***p = 0.0001$, $**p = 0.0014$ (**h**); $*p = 0.0102$ (**i**). **j–l** Weight of eWAT, iWAT, and pWAT from lean and obese mice described in (**h**) ($n = 5$ per group). $*p = 0.0164$ (**j**); $*p = 0.0304$ (**k**). **m, n** Histological analysis by H&E staining (**m**) and Oil red oxygen dyeing (**n**) of liver tissues from CD and HFD C57/BL6 mice described in (**h**). Data are representative of independent three experiments. (**o**) Interactive database of Noc4l expression and diabetes-related clinical phenotypes. All data are presented as mean ± SEM. Statistics in **b–g**, **j–l** represent an unpaired, two-tailed Student's t test. Statistical analysis in h-i was performed with two-way ANOVA with Tukey's test. Quantification of AUC from (**h**) was analyzed by two-tailed Student's t test. $*p < 0.05$, $**p < 0.01$, $***p < 0.001$. Source data are provided as a Source Data file.

Considering that the Noc4l$^{LKO}$ mice exhibited increased concentrations of blood glucose, insulin, FFA, TG, and CHOL, glucose and insulin tolerance tests (GTT and ITT) were performed to assess the systemic insulin sensitivity. Glucose tolerance of Noc4l$^{LKO}$ mice was significantly impaired relative to the Noc4l$^{fl/fl}$ mice with CD; however, no significant difference in insulin response was detected in the CD-fed mice (Fig. 3d). The HFD-fed Noc4l$^{LKO}$ mice were considerably more intolerant to glucose than the Noc4l$^{fl/fl}$ mice (Fig. 3e). The insulin sensitivity of HFD-fed Noc4l$^{LKO}$ mice was also significantly impaired (Fig. 3e). To directly investigate the effect of Noc4l on insulin signaling, HFD-fed mice with Noc4l$^{fl/fl}$ and Noc4l$^{LKO}$ at aged 20 weeks were intraperitoneally injected with insulin or PBS, after an overnight fast, the mice were removed liver, adipose tissue, and muscle quickly for biochemical analyses. The results indicated that insulin-stimulated AKT (Ser473) phosphorylation was significantly attenuated in the WAT and muscle of Noc4l$^{LKO}$ mice relative to that in their age-matched controls (Fig. 3f–h). These findings suggested that a lack of Noc4l in the myeloid compartment impaired glucose metabolism in the context of chronic exposure to HFD.

To further evaluate the role of myeloid Noc4l in energy balance, the metabolic rates in age-matched Noc4l$^{fl/fl}$ and Noc4l$^{LKO}$ mice on either CD or HFD for 10 weeks were measured using metabolic chambers. Compared with Noc4l$^{fl/fl}$ mice, the Noc4l$^{LKO}$ mice had the decreased oxygen consumption (VO$_2$), VCO$_2$ production, and energy expenditure (EE) both on CD and HFD (Fig. 3i, j). Meanwhile, there was no change of locomotor activity between Noc4l$^{fl/fl}$ and Noc4l$^{LKO}$ mice (Supplementary Fig. 2). Taken together, deletion of Noc4l in myeloid lowered the energy expenditure under both CD and HFD conditions.

To confirm the functions of Noc4l in glucose metabolism, we generated mice exhibiting myeloid-specific overexpression of Noc4l (OE) (Supplementary Fig. 3a). Glucose homeostasis was improved in HFD-fed OE mice compared with the HFD-fed WT mice, although no significant difference in insulin sensitivity was observed (Supplementary Fig. 3b, c).

**Macrophage-specific deletion of Noc4l aggravated HFD-induced inflammation in mice.** To determine whether macrophage-specific deletion of Noc4l inducing IR is associated with increased inflammation, the levels of plasma and tissue proinflammatory cytokines were analyzed. The plasma concentrations of IL-6 and TNFα were significantly higher in the HFD-fed Noc4l$^{LKO}$ mice than in the age-matched Noc4l$^{fl/fl}$

(Fig. 4a). The expression levels of the proinflammatory genes in the WAT and liver were measured by qRT-PCR analysis. We detected higher expression levels of *TNFα*, *IL-6*, and *MCP1* in the WAT and *MCP1* in the liver of the HFD-fed Noc4l$^{LKO}$ mice than in their Noc4l$^{fl/fl}$ counterparts (Fig. 4b). In addition, macrophage infiltration was increased in the WAT of Noc4l$^{LKO}$ mice relative to Noc4l$^{fl/fl}$ controls on HFD, as determined by increased levels of the macrophage transcript markers *CD68* and *F4/80* (Fig. 4b). Consistent with this, histopathological analysis indicated that the infiltration of inflammatory cells in the epididymal fat of the HFD-fed Noc4l$^{LKO}$ mice was increased (Fig. 4c). Moreover, immunohistochemical analysis of the macrophage marker F4/80 revealed more crown-like structures in the adipose tissue of the HFD-fed Noc4l$^{LKO}$ mice than that of the Noc4l$^{fl/fl}$ counterparts (Fig. 4d), indicating a greater abundance of macrophages in the adipose tissue of Noc4l$^{LKO}$ mice. And flow cytometric analysis confirmed the increased presence of F4/80$^+$ macrophages in the WAT of Noc4l$^{LKO}$ mice (Fig. 4e and Supplementary Fig. 6a). The combined results indicated that Noc4l deletion in macrophages exacerbated the HFD-induced systemic and local inflammation in mice.

**Noc4l inhibited M1-like macrophage differentiation and activation ex vivo.** Macrophages that are recruited to adipose tissue and polarize from anti-inflammatory M2 state to pro-inflammatory M1-like state produce some source of proinflammatory cytokines during obesity. ATMs are derived from bone marrow precursors that migrate from the peripheral circulation[3]. In order to study the mechanism underlying macrophage specific deletion of Noc4l inducing inflammation, BMDMs were isolated from Noc4l$^{fl/fl}$ and Noc4l$^{LKO}$ mice and treated with LPS or IL-4 to promote polarization of M1 or M2 macrophages, respectively[25–27]. To rule out deletion of Noc4l in macrophages may affect the LPS receptor and regulator, we detected the expression of TLR4[28] and CD14[29] in BMDMs of Noc4l$^{fl/fl}$, Noc4l$^{LKO}$, and OE mice. CD14 and TLR4 expression were not altered among three types of mice (Supplementary Fig. 4a, b and Supplementary Fig. 6b). Next, the expression of specific markers of M1 and M2 macrophages were detected by qRT-PCR. Notably, BMDMs from Noc4l$^{LKO}$ mice had significantly increased responses to LPS stimulation; the enhanced transcriptional expressions of *IL-6*, *TNFα*, *MCP1*, and *IL-1β* were evident (Fig. 4f and Supplementary Fig. 4f). Consistent with the transcription level, the protein levels of IL-6 and TNFα were elevated in the supernatant of Noc4l$^{LKO}$ BMDMs compared with the age-matched Noc4l$^{fl/fl}$ controls (Fig. 4g, h). We further

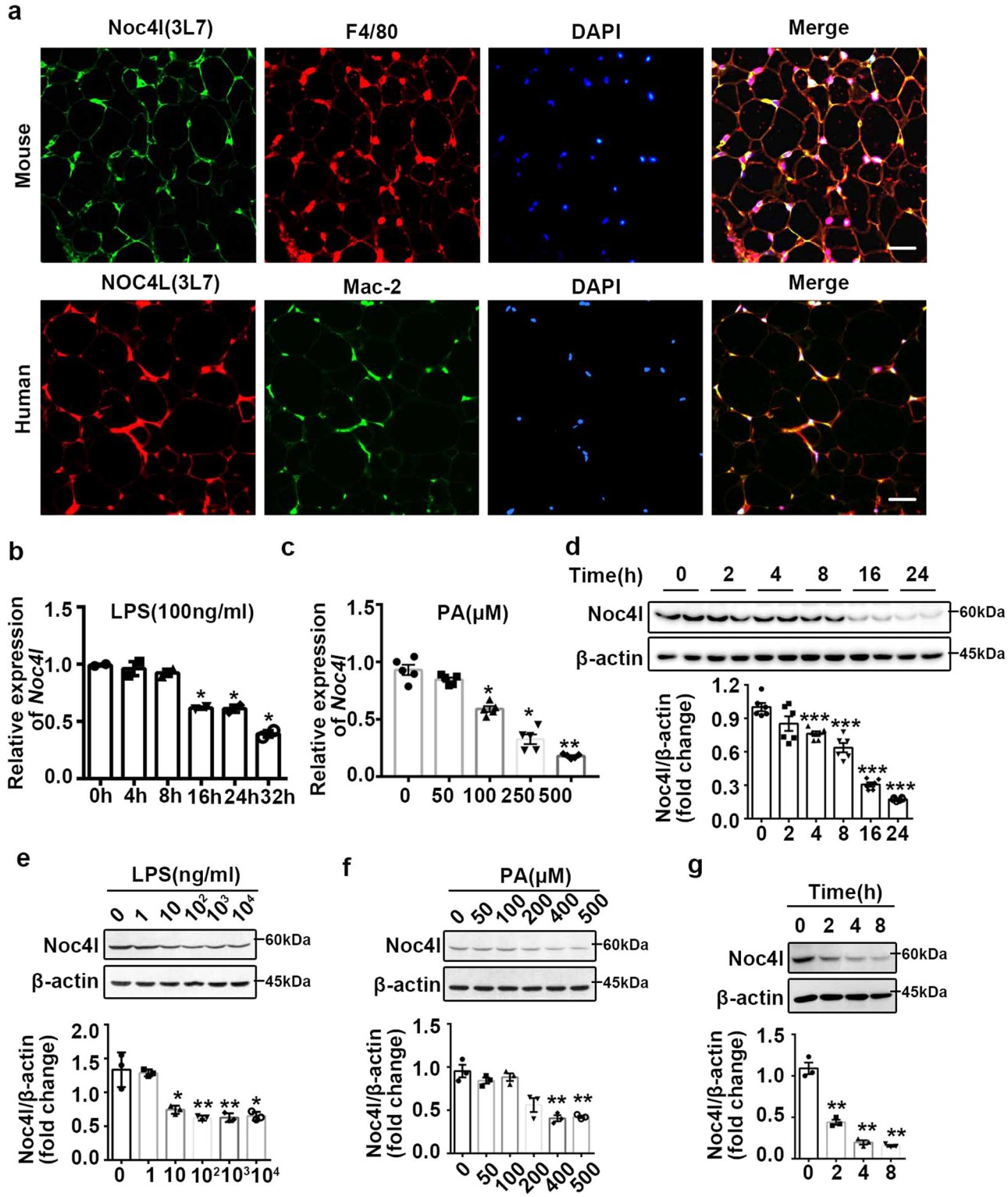

analyzed gene expression profiling in unstimulated BMDMs from Noc4l^LKO mice by microarray assay. Of the annotated genes, approximately 200 genes were upregulated upon the deletion of Noc4l (Supplementary Fig. 4c). Pathway analysis revealed the enrichment of pro-inflammatory signaling pathways in Noc4l^LKO *vs.* Noc4l^fl/fl BMDMs (Supplementary Fig. 4d). Notably, proinflammatory cytokines among the upregulated genes such as *IL-6* and *TNFα* (Supplementary Fig. 4e) further confirmed our qRT-PCR results. Moreover, consistent with the results from BMDMs

the transcriptional expressions of *IL-6* and *TNFα* in ATMs of Noc4l^LKO mice were significantly higher than Noc4l^fl/fl controls on HFD (Fig. 4i). Furthermore, we found that Noc4l^LKO BMDMs potently decreased the M2 macrophage marker *Arg1* mRNA expression after treating with IL-4, and the basal expression levels of the M2 macrophage markers *Arg1, Mrc1* were significantly reduced in unstimulated Noc4l^LKO BMDMs (Fig. 4j). Moreover, the mRNA expression of anti-inflammatory marker *IL-10* was significantly decreased upon LPS stimulation (Supplementary

**Fig. 2 Noc4l was located in ATMs and downregulated by LPS and PA. a** Immunofluorescence staining for Noc4l (green), F4/80 (red), and DAPI (blue) in the eWAT of mice (top) and NOC4L (red), Mac-2 (green) and DAPI (blue) in fat tissue of human patients (bottom). Data are representative of independent three experiments. Scale bar, 50 μm. **b, c** Relative mRNA expression of *Noc4l* determined by qRT-PCR assay in RAW264.7 cells treated with 100 ng/mL LPS at different time points (**b**) and different doses of PA (**c**) for 24 h. *n* = 5 biological replicates. *\**p* = 0.0152 (16 h), *\**p* = 0.0161 (24 h), *\**p* = 0.029 (32 h) **b**; *\**p* = 0.02 (100 μM), *\**p* = 0.01(250 μM), \*\**p* = 0.002 (500 μM) **c**. **d–f** Protein expression of Noc4l in RAW264.7 cells treated with LPS (100 ng/mL) at different time points **d**, different doses of LPS **e**, and PA **f** for 24 h. \*\*\**p* = 0.0002 (4 h), \*\*\**p* = 0.0001 (8 h, 16 h, 24 h) **d**; *\**p* = 0.0179 (10 ng/ml), \*\**p* = 0.0088 (100 ng/ml), \*\**p* = 0.0098 (1000 ng/ml), *\**p* = 0.0110 (10000 ng/ml) **e**. \*\**p* = 0.0025 (400 μM), \*\**p* = 0.0021 (500 μM) **f**. **g** Protein expression of the Noc4l in phorbol 12-myristate 13-acetate (PMA)-differentiated THP-1 cells treated with 100 ng/mL LPS at different time points. \*\**p* = 0.0076 (2 h), \*\**p* = 0.0021 (4 h), \*\**p* = 0.0016 (8 h). **d–g** Quantification of the Noc4l/β-actin ratio were shown in bottom panel, respectively. Scale bar, 50 μm. Data are representative of independent three experiments. All data are presented as mean ± SEM. Statistics in **b–g** represent an unpaired, two-tailed Student's *t* test. *\**p* < 0.05, \*\**p* < 0.01, \*\*\**p* < 0.001. Source data are provided as a Source Data file.

Fig. 4f). Additionally, PA-treated BMDMs from Noc4l[LKO] mice also expressed much higher levels of *IL-6* and an upward trend of *TNFα* compared to the age-matched Noc4l[fl/fl] mice (Fig. 4k, l). OE mice, however, had no difference with WT in terms of *IL-6* and *TNFα* expression after LPS treatment (Supplementary Fig. 4g) and this was consistent with the ITT result. Taken together, these results demonstrated that Noc4l deficiency promoted M1-like macrophage polarization and which in turn, increased inflammatory state.

To further examine the role of inflammation in the induction of IR and its dependence on Noc4l, a nonlethal dose of LPS was used to induce acute inflammation. Mice were injected with LPS (1 mg/kg) intraperitoneally and GTTs were performed following an overnight fast for 14 h. The Noc4l[LKO] mice were more glucose intolerant than Noc4l[fl/fl] mice after glucose loading (Fig. 4m). Thus, deletion of Noc4l in macrophages leads to IR may involve increased proinflammatory factor production.

Meanwhile, ITT was performed to assess the systemic insulin sensitivity on body-weight matched Noc4l[LKO] and Noc4l[fl/fl] mice on HFD. Insulin tolerance of Noc4l[LKO] mice was higher than the Noc4l[fl/fl] weight matched mice with HFD (Fig. 4n). These results suggested that the metabolic abnormalities were not driven by the increased body weight of Noc4l[LKO] mice on HFD.

**Noc4l inhibited MyD88-independent TLR4/TRIF pathway in macrophages**. Based on the IR and inflammatory response of Noc4l[LKO] mice, we next explored how Noc4l regulates this process. To assess whether Noc4l was involved in ribosome biogenesis in macrophages, ribosomal profiles in BMDMs of Noc4l[fl/fl] and Noc4l[LKO] mice were analyzed by sucrose density ultracentrifugation. As shown in Supplementary Fig. 4h, Noc4l-deficient BMDMs have a similar 40 S peak with Noc4l[fl/fl].

In addition, LPS triggers innate immune responses via TLR4[30], which contains two major pathways: the MyD88-dependent and MyD88-independent pathways (TRIF-dependent pathway)[31–34]. To investigate potential changes in the signaling pathway by which Noc4l regulates macrophage functions, BMDMs were stimulated with LPS in a time-course study and evaluated several signaling parameters. Almost no differences in the MyD88-dependent signaling pathway, including IκBα, JNK, and Erk in response to LPS were observed in both Noc4l[fl/fl] and Noc4l[LKO] BMDMs (Fig. 5a). Then MyD88-induced NF-κB activation was analyzed by luciferase assay. MyD88-induced NF-κB activation was not affected by NOC4L expression (Fig. 5b). Similarly, no such effect was exerted on TRAF6 or P65, which functions downstream of MyD88 in TLR4 signaling (Fig. 5b). These results indicated that Noc4l was not involved in the MyD88-dependent signal pathway.

Meanwhile, we evaluated the effects of Noc4l deficiency on the activation of TRIF-dependent signaling, which mainly activates the transcription factor IRF3 in BMDMs by LPS treatment. Noc4l deficiency significantly increased the phosphorylation of IRF3 relative to that of the control (Fig. 5c). Activation of IRF3 can induce the expression of type I IFNs as well as IFN-inducible genes, such as *CCL5*[31,32,35,36]. Notably, the transcription levels of *IFNβ* and *CCL5* were significantly elevated in the BMDMs from the Noc4l[LKO] mice (Fig. 5d). Consistently, NOC4L expression significantly inhibited the TRIF-induced activation of the interferon-stimulated response element (*ISRE*) (Fig. 5e). In addition to IRF3 activation, TRIF-dependent signaling can trigger the activation of NF-κB, referred to as "the late NF-κB" [28,32,37,38]. TRIF-induced NF-κB activation was measured by luciferase assay. In contrast to the lack of effect on MyD88-induced NF-κB activation, TRIF-induced NF-κB activation was inhibited by NOC4L expression in a dose-dependent manner (Fig. 5f). Meanwhile, the isolated BMDMs from Noc4l[fl/fl], Noc4l[LKO], TRIF KO, and Noc4l[−/−]/TRIF[−/−] double KO (dKO) mice were treated with LPS. The mRNA expressions of *TNFα*, *IL-6*, and *IFNβ* from Noc4l[LKO] BMDMs were higher than Noc4l[fl/fl] when stimulated with LPS, but had no changes in BMDMs of Noc4l[−/−]/TRIF[−/−] dKO mice in comparted to Noc4l[fl/fl] mice (Fig. 5g–i).

Moreover, to assess the systemic insulin sensitivity, the GTT assay was performed in Noc4l[−/−]/TRIF[−/−] dKO mice. Glucose tolerance of Noc4l[−/−]/TRIF[−/−] dKO mice was ameliorated compared to Noc4l[LKO] mice, and had no significant difference compared with TRIF KO mice on CD (Supplementary Fig. 4i). Thus, Noc4l deletion induced IR may involve the activation of TRIF signal.

Besides TLR4, stimulation of TLR3 can activate IRF3 via a TRIF-dependent pathway[39,40]. By contrast, other TLRs such as TLR2 and TLR7, trigger downstream signals only via the MyD88-dependent pathway[41–43]. Then the inflammatory cytokines production by BMDMs in response to Poly I:C, LTA, and Poly U ligands for TLR3, TLR2, and TLR7, were analyzed, respectively. Poly I:C could potentially induce higher transcription levels of *TNFα* and *IFNβ* in BMDMs from the Noc4l[LKO] mice relative to age-matched Noc4l[fl/fl] mice (Supplementary Fig. 4j). However, LTA and Poly U induced similar levels of *IL-6* and *TNFα* in BMDMs from the Noc4l[LKO] and Noc4l[fl/fl] mice (Supplementary Fig. 4k). Overall, these results demonstrated that Noc4l exerted no effect on the MyD88-dependent signaling pathway but inhibited the TRIF pathway of TLR3/4 in macrophages.

**Noc4l localized on the early endosome-associated subcellular compartments and inhibited TLR4 endocytosis**. We subsequently determined the precise mechanism by which NOC4L regulates the TRIF-dependent signaling pathway. TRIF-mediated signal transduction occurs on the endosome[28,44]; thus, the subcellular localization of NOC4L was assessed. In order to systematically study the localization of NOC4L, cytoplasmic and nuclear extraction assay was performed on several types of cells, including NIH3T3, 3T3-L1, GC-1, GC-2, and BMDM cells. To our surprise, NOC4L was located differently in various cells. Noc4l existed mainly in the nucleus of NIH3T3, 3T3-L1, GC-1

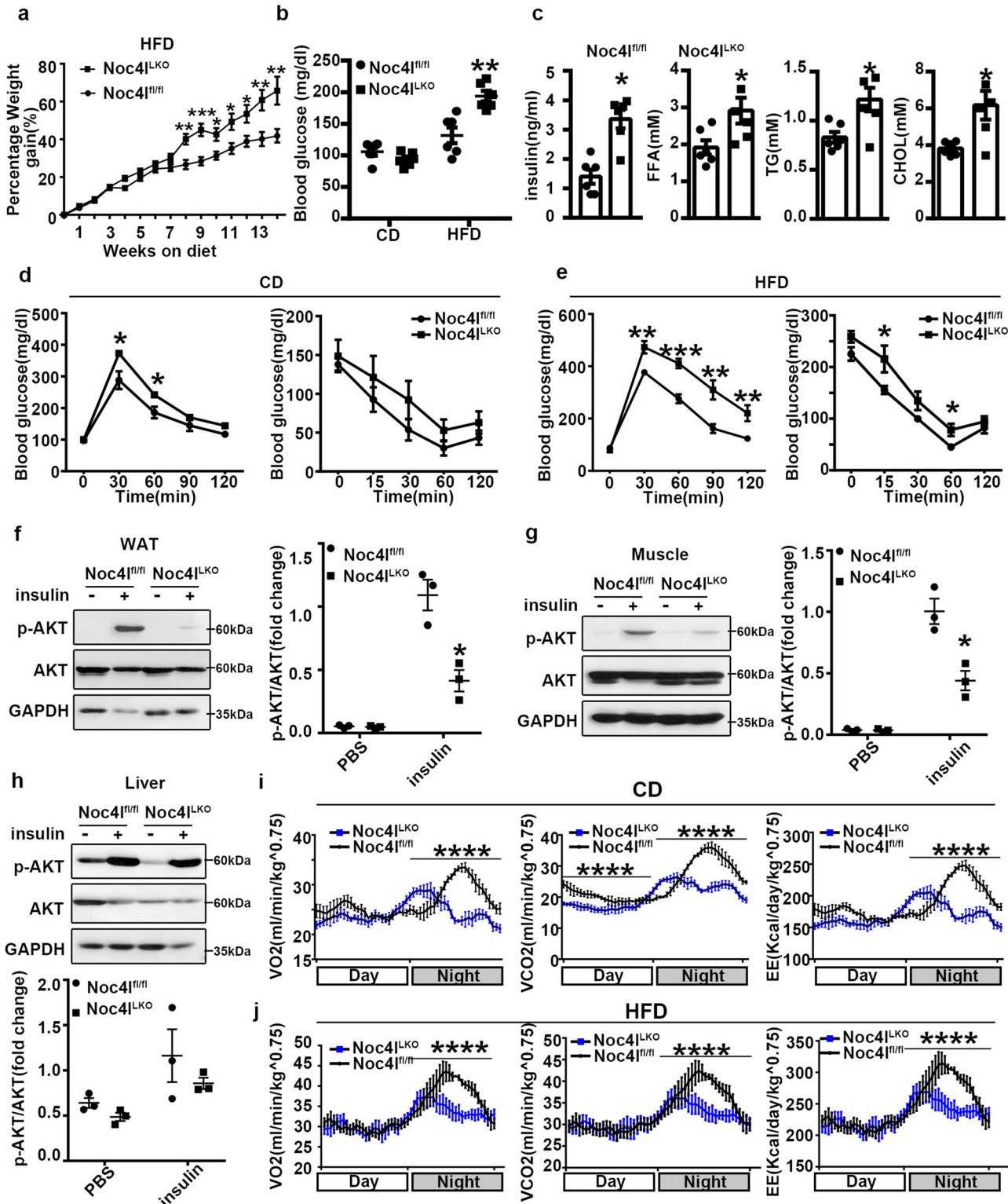

and GC-2 cells (Supplementary Fig. 5a), but predominantly in the cytoplasm of primary BMDM cells (Fig. 5j). The subcellular localization of NOC4L was also examined in situ. Considering that BMDMs is difficult to be transfected, lentivirus combined with green fluorescent (GFP) labeled Noc4l were used to detect its location. The fluorescence signals were observed in the cytoplasm of the BMDM cells (Fig. 5k). To further investigate its location, HeLa cells were transfected with Flag-NOC4L plasmid and subjected to immunofluorescence by Flag antibody, NOC4L

fluorescence distributed predominantly in nucleus, but also some punctate fluorescence signals in the cytoplasm (Supplementary Fig. 5b). Furthermore, NOC4L was co-stained with the early endosomal marker Rab5-positive compartments in HeLa cells (Fig. 5l). Moreover, NOC4L was not co-stained with Rab7 (late endosome) or Rab11 (cycling endosome)-positive compartments (Fig. 5l). In addition, NOC4L staining was not detected on other subcellular organelles, such as LysoTracker-positive (acidic compartment, lysosomal marker) or endoplasmic reticulum (ER)

**Fig. 3 Deletion of Noc4l in macrophages affected insulin sensitivity and energy metabolism. a** Weight gain percentage of Noc4l[fl/fl] ($n = 10$) and Noc4l[LKO] mice ($n = 7$) with HFD for 14 weeks. Statistical analysis was performed with two-way ANOVA with Tukey's test. **$p = 0.003$ (8w), ***$p = 0.0005$ (9w), *$p = 0.01$ (10w), *$p = 0.012$ (11w), *$p = 0.014$ (12w), **$p = 0.005$ (13w), **$p = 0.006$ (14w). **b, c** Blood glucose concentrations (**b**), plasma concentrations of insulin, free fatty acids (FFA), triglycerides (TG), and cholesterol (CHOL) (**c**) of Noc4l[fl/fl] and Noc4l[LKO] mice on CD or HFD for 20 weeks ($n = 6$ per group). Two-tailed Student's t test. **$p = 0.0022$ (**b**); *$p = 0.0246$ (c, insulin), *$p = 0.032$ (**c**, FFA), *$p = 0.0156$ (c, TG), *$p = 0.0141$ (c, CHOL). **d, e** GTT and ITT of Noc4l[fl/fl] and Noc4l[LKO] mice on CD (Noc4l[fl/fl] mice $n = 11$, Noc4l[LKO] mice $n = 7$) (**d**) or HFD (GTT:Noc4l[fl/fl] mice $n = 11$, Noc4l[LKO] mice $n = 11$; ITT: Noc4l[fl/fl] mice $n = 10$, Noc4l[LKO] mice $n = 7$.) **e** for 20 or 21 weeks. Statistical analysis was performed with two-way ANOVA with Tukey's test. *$p = 0.039$ (30 min), *$p = 0.037$ (60 min) (**d**); **$p = 0.0023$ (GTT 30 min), ***$p = 0.0008$ (GTT 60 min), **$p = 0.0033$ (GTT 90 min), **$p = 0.0074$ (GTT 120 min); *$p = 0.020$ (ITT 15 min), *$p = 0.015$ (ITT 60 min) (**e**). **f–h** Protein expression of p-AKT and AKT in eWAT (**f**), muscle (**g**), and liver tissues (**h**) from Noc4l[fl/fl] and Noc4l[LKO] mice on HFD for 20 weeks injected intraperitoneally with insulin for 10–15 min. Quantification of the p-AKT/AKT ratio was shown in the right (**f, g**) or bottom (**h**) panel. Data represented three independent experiments. Two-way ANOVA. *$p = 0.0103$ (**f**); *$p = 0.0125$ (**g**). **i–j** Oxygen consumption (VO2), carbon dioxide production (VCO2) and energy expenditure (EE) of Noc4l[fl/fl] ($n = 3$) and Noc4l[LKO] ($n = 4$) mice on CD (**i**) and HFD (**j**) for 10 weeks. ****$p < 0.0001$ (Noc4l[LKO] vs Noc4l[fl/fl]). Statistical analysis was performed with two-way ANOVA with Tukey's test. All data are presented as mean ± SEM. *$p < 0.05$, **$p < 0.01$, ***$p < 0.001$. Source data are provided as a Source Data file.

Tracker-positive compartments (Supplementary Fig. 5c). To further confirm these results, co-localization of Noc4l with the Rab5 was detected in BMDMs by immunofluorescence assay (Fig. 5m). The co-localization of Noc4l with early endosome marker EEA was also detected in RAW264.7 cells (Supplementary Fig. 5d). These results suggested that the localization of NOC4L in the nucleus and cytoplasm may play different roles in biological process of different cells, especially in macrophages, which was predominantly detected in the cytoplasm.

LPS induces the endocytosis of TLR4 in macrophages[28]. Considering that NOC4L was located in endosomes, we hypothesized that NOC4L could regulate the ligand-induced TLR4 endocytosis. TLR4 surface staining, a readout for TLR4 endocytosis[28,29], was reduced in the unstimulated Noc4l[LKO] PMs compared relative to that in the Noc4l[fl/fl] PMs controls (Fig. 5n and Supplementary Fig. 6b). The reduction in TLR4 surface staining was considered a *bona fide* endocytic event because it was inhibited by dynasore (Supplementary Fig. 5e), an inhibitor of dynamin GTPases that was essential for TLR4 internalization[28]. Loss of TLR4 surface expression was enhanced over time in Noc4l[LKO] PMs compared relative to that in the WT PMs upon LPS treatment (Fig. 5o and Supplementary Fig. 6b). By contrast, TLR4 internalization was decreased in the PMs from Noc4l[LOE] mice relative to that in the WT controls (Supplementary Fig. 5f). The levels of IFNβ and TNFα were decreased after inhibiting the internalization of TLR4 by dynasore, and there was no difference between Noc4l[LKO] and Noc4l[fl/fl] BMDMs (Fig. 5p, q). These results indicated that Noc4l could block the endocytosis of TLR4, thus reducing the production of IFNβ.

**NOC4L directly interacted with the TIR of TLR4.** Based on the results that NOC4L blocked the endocytosis of TLR4, we explored whether NOC4L could interact with TLR4 to regulate this process. Co-immunoprecipitation (co-IP) in HeLa cells which were overexpression of FLAG-NOC4L and TLR4-EGFP revealed the interaction of exogenous NOC4L and TLR4 (Fig. 6a). Furthermore, the co-IP assay showed that endogenous Noc4l was interacted with endogenous TLR4 in HeLa cells (Supplementary Fig. 5g) and RAW264.7 cells (Fig. 6b). Moreover, endogenous Noc4l and TLR4 interacted physically in BMDMs from Noc4l[fl/fl] but not from Noc4l[LKO] mice by immunoprecipitation (Fig. 6c). Additionally, the interaction of Noc4l and TLR4 was enhanced in RAW264.7 cells stimulated with LPS (Fig. 6d). The interaction between TLR4 and Noc4l was further confirmed in the peritoneal macrophages (PMs) of Noc4l[fl/fl], Noc4l[LKO], and Noc4l[LOE] mice by proximity ligation assay (PLA) (Fig. 6e), which critically depends on the close proximity (<40 nm) between interaction partners[45]. The quantitative results showed that almost no fluorescence signal was detected in PMs of Noc4l[LKO] mice, and

the fluorescence signal increased significantly in PMs of Noc4l[LOE] mice (Fig. 6f). In order to rule out the false positive results due to non-specificity of TLR4, we performed PLA in PMs of TLR4 mutant mice, and the result showed no fluorescent signal was detected (Fig. 6g). To determine whether there is a direct interaction between TLR4 and Noc4l, full-length Flag-NOC4L-His and GST-TLR4 (1–4, truncated according to its functional domains) were expressed and purified from Sf9 cells and *E. coli* strain BL21. Results from GST pulldown assays showed that the purified Flag-NOC4L-His strongly interacted with the TIR domain of TLR4 (Fig. 6h).

**Molecular changes in adipose tissues of T2D patients were similar to that of Noc4l[LKO] mouse.** To test whether this Noc4l[LKO] mouse model reflects T2D in humans, the selected important inflammatory genes found in the Noc4l[LKO] mice were validated by cDNA microarray on adipose tissues from T2D patients (NCBI, GEO accession GSE29231). Consistently, *NOC4L* expression was significantly reduced, whereas some inflammatory genes, such as *CCL2, IL-6, CD69, MSR1, IL-1β*, and *TRIF* were increased in the visceral adipose tissue of diabetic subjects ($n = 12$) *vs.* non-diabetic subjects ($n = 12$) (Fig. 7a). Overall, these results suggested that a pattern of molecular markers was observed in T2D patients similar to those in the Noc4l[LKO] mice.

## Discussion

Noc4l has been considered localized in the nucleus of cells and plays a fundamental role in ribosome biosynthesis for a long time. Hereby, we reported an unexpected location and function of NOC4L in macrophages, demonstrated that NOC4L was decreased in obese subjects and mice. By using the Lv-Noc4l, macrophage-specific deletion and overexpression of Noc4l mouse models, and several studies in vitro, we clarified a previously unidentified biological function of NOC4L, which directly interacted with TLR4 to inhibit its endocytosis and blocked the TRIF pathway. Consequently, the production of inflammatory cytokines was reduced, which in turn ameliorated local and systemic inflammation and influenced insulin sensitivity. Data mined from the diabetic-related public database and the experiments from diabetic patients confirmed our findings that the expression of Noc4l was negatively correlated to glucose levels in mice.

NOC4L, the ortholog of yeast Noc4p, was first identified by functional proteomic analysis of HeLa cell nucleoli in 2002[46]. Less than 20 published articles referring to "NOC4L" have been introduced in PubMed, but the functional studies on NOC4L have been rarely reported. General Noc4l deficiency leads to preimplantation embryonic lethality in mice[17], similar to other proteins critically involved in ribosomal RNA synthesis or

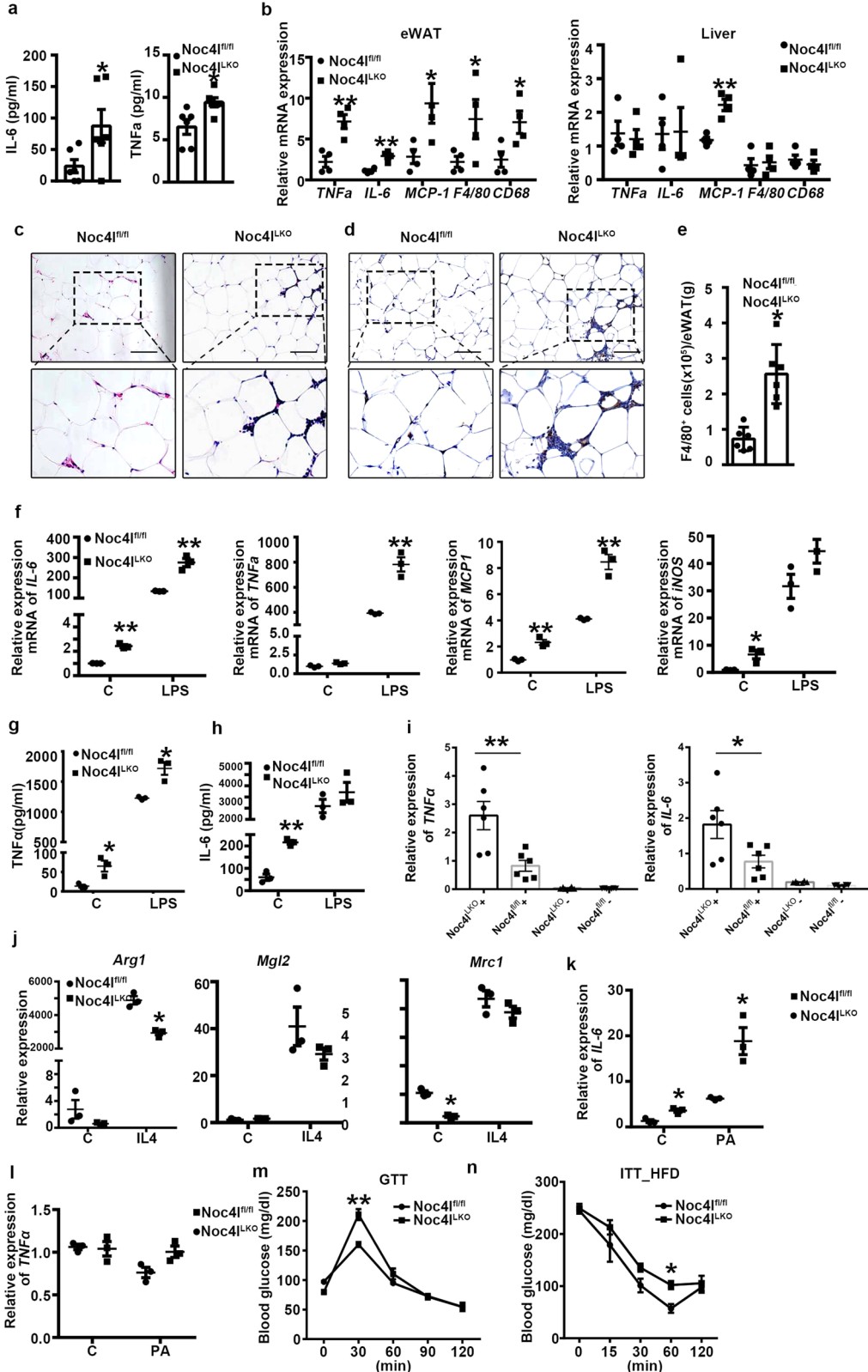

processing the deficiency of which leads to embryonic lethality, such as RBM19 and Fad24[47,48]. We found that NOC4L was located differently in various cells. Noc4l was located in the nucleus of NIH3T3, 3T3-L1, GC-1, and GC-2 cells, but in the early endosome of BMDMs. The localization of NOC4L in the nucleus and cytoplasm may play different roles in biological process of different cells, especially in macrophages, which are

only detected in the cytoplasm. Specific ablation of Noc4l in Tregs resulted in a severe autoimmune disease through mediating ribosome biogenesis. However, the author did not study the location of Noc4l in Tregs. In our study, macrophages specific deletion of Noc4l resulted in IR through triggering endosomal TRIF/IRF3 signal. Zhu et al. reported that Noc4l deficiency resulted in impairing the activation of Tregs by interfering the

**Fig. 4 Macrophage-specific deletion of Noc4l aggravated HFD-induced inflammation and promoted proinflammatory M1-like Macrophage polarization. a** Cytokine concentrations of IL-6 and TNFα in plasma from Noc4l[fl/fl] and Noc4l[LKO] mice on HFD for 20 weeks ($n = 6$ per group). Two-tailed Student's t test. *$p = 0.0491$(IL-6); *$p = 0.0179$ (TNFα). **b** Relative mRNA levels of inflammation markers from eWAT (left) and liver (right) of Noc4l[fl/fl] and Noc4l[LKO] mice on HFD for 20 weeks ($n = 4$ per group). Two-tailed Student's t test. eWAT: **$p = 0.0041$ (TNFα), **$p = 0.0015$ (IL-6), *$p = 0.0452$ (MCP-1), *$p = 0.0498$ (F4/80), *$p = 0.0345$ (CD68); Liver: **$p = 0.0015$ (MCP-1). **c** H&E staining of eWAT of Noc4l[fl/fl] and Noc4l[LKO] mice on CD or HFD for 16 weeks. Data are representative of independent three experiments. **d** Immunohistochemical staining for F4/80 in eWAT from Noc4l[fl/fl] and Noc4l[LKO] mice on CD or HFD for 16 weeks. Data are representative of independent three experiments. **e** Flow cytometric analysis of F4/80[+] cells of the SVF isolated from the eWAT of Noc4l[fl/fl] and Noc4l[LKO] mice on HFD for 16 weeks ($n = 6$ per group). Gating strategy for flow cytometry was shown in Supplementary Fig. 6a. Two-tailed Student's t test. *$p = 0.0118$. (**f**) Relative mRNA expression of IL-6, TNFα, MCP1 and iNOS of BMDMs from Noc4l[fl/fl] and Noc4l[LKO] mice incubated with 100 ng/mL LPS for 24 h ($n = 3$ biological replicates). Two-tailed Student's t test. **$p = 0.0028$ (C-IL-6), **$p = 0.0022$ (LPS-IL-6); **$p = 0.0024$ (LPS-TNFα); **$p = 0.0034$ (C-MCP1), **$p = 0.0017$ (LPS-MCP1); *$p = 0.0235$ (C-iNOS). **g, h** Concentration of TNFα (**g**) and IL-6 (**h**) in the supernatant secreted by BMDMs from Noc4l[fl/fl] and Noc4l[LKO] mice after LPS stimulation (100 ng/mL) for 24 h ($n = 3$ biological replicates per group). Two-tailed Student's t test. *$p = 0.0238$ (C-TNFα), *$p = 0.0103$ (LPS-TNFα); **$p = 0.007$ (C-IL6) (**h**). **i** Relative mRNA expression of IL-6 and TNFα of ATMs from Noc4l[fl/fl] and Noc4l[LKO] mice on HFD for 16 weeks. Noc4l[fl/fl]+ and Noc4l[LKO]+ represent CD11b-positive cells derived from Noc4l[fl/fl] and Noc4l[LKO] mice, respectively; and Noc4l[fl/fl]- and Noc4l[LKO]- represent CD11b-negative cells derived from Noc4l[fl/fl] and Noc4l[LKO] mice, respectively. ($n = 6$ per group). Two-tailed Student's t test. **$p = 0.0076$ (TNFα); *$p = 0.0356$ (IL-6). (**j**) Relative mRNA expression levels of Arg1, Mgl2 and Mrc1 of BMDMs from Noc4l[fl/fl] and Noc4l[LKO] mice incubated with 10 ng/mL IL-4 for 48 h ($n = 3$ biological replicates). Two-tailed Student's t test. *$p = 0.0216$ (Arg1); *$p = 0.0117$ (Mrc1). (**k-l**) Relative mRNA expression of IL-6 and TNFα of BMDMs from Noc4l[fl/fl] and Noc4l[LKO] mice stimulated with 500 μM palmitate (PA) for 24 h ($n = 3$ biological replicates). Two-tailed Student's t test. *$p = 0.01$(C-IL-6), *$p = 0.0134$ (PA-IL-6). (**m**) GTT of Noc4l[fl/fl] ($n = 3$) and Noc4l[LKO] ($n = 6$) mice after injected with LPS (1 mg/kg) and overnight fast for 14 h. Statistical analysis was performed with two-way ANOVA with Tukey's test. **$p = 0.0036$ (30 min). **n** ITT of Noc4l[fl/fl] mice placed on the HFD until their average body weight matched that for Noc4l[LKO] mice on HFD for 10 weeks ($n = 4$ per group). Statistical analysis was performed with two-way ANOVA with Tukey's test. *$p = 0.0308$ (60 min). All data are presented as mean ± SEM. *$p < 0.05$, **$p < 0.01$, ***$p < 0.001$. Scale bar, 50 μm. Source data are provided as a Source Data file.

ribosome 40 S subunit formation. In contrast to their study, we did not observe significant changes in ribosome 40 S subunit formation in Noc4l-deficient macrophages, which may implicate other molecular function of Noc4l in macrophages.

Accumulation of ATMs in obese adipose is partly derived from bone marrow precursors[3,49]. By total body irradiation mouse model, Hill and colleagues showed local ATMs can be reconstituted by BMDMs after total body irradiation[50]. According to the limitations of ATM such as relatively small number and isolation techniques difficulties, therefore in this study BMDMs were used to test proinflammatory cytokine expression ex vivo. Importantly, we further confirmed inflammatory cytokines expression in ATMs of Noc4l[LKO] were higher than those of Noc4l[fl/fl], which is consistent with the results of BMDMs.

ATMs are a heterogeneous population of immune cells with a highly plastic phenotype according to microenvironment and exhibit a diverse spectrum of metabolic characteristics. In general, ATMs include alternatively activated M2-like macrophages and classically activated M1-like macrophages. M1-like and M2-like represent the proinflammatory state of recruited ATMs and the anti-inflammatory state of resident ATMs, respectively. Interestingly, Kratz et al. identified a distinct population of metabolically activated macrophages (MMe), following palmitate, glucose, and insulin challenge which was distinct from bacterially activated M1 macrophages[25]. In our study, we also measured the effect of Noc4l on MMe after the treatment of palmitate. The result showed that proinflammatory cytokines expressions were increased in BMDMs of Noc4l[LKO] mice compared to WT mice. These findings may suggest the role of Noc4l in regulating MMe functions during obesity. However, the mechanism of how Noc4l regulating MMe needs to be studied further.

Diet-induced metabolic endotoxemia is an important factor for the low-grade inflammation in obesity and metabolic diseases[21]. The circulating LPS levels are increased owing to diet or obesity[51,52]. In contrast to acute inflammatory response, LPS produced from the gut microbiota of obese subjects and diet-induced animal models binds to its receptor TLR4 to induce the production of pro-inflammatory cytokines leading to low-grade systemic inflammation[53]. In our case, consistent with HFD-induced obesity model we observed that glucose tolerance of

Noc4l[LKO] mice was significantly impaired relative to the Noc4l[fl/fl] mice with a nonlethal dose of LPS injection intraperitoneally. Besides LPS, the serum levels of saturated free fatty acids such as palmitate are elevated in obesity, which might be an important player to activate proinflammatory pathways and induce cytokines expression. There exist a number of mechanisms about how palmitate regulates obesity-associated inflammation[1,54,55]. However, recent studies showed that palmitate does not directly bind TLR4, although Tlr4[−/−] attenuates inflammation in palmitate-treated macrophages[56]. So how Noc4l regulates palmitate-associated signaling is still required to be further studied.

The present study showed that the expression levels of Noc4l in macrophages were downregulated by LPS and saturated free fatty acids. Indeed, the HFD-fed Noc4l[LKO] mice exhibited accelerated systemic/local inflammation and macrophage infiltration in adipose tissues; metabolic alterations, including obesity, glucose intolerance, and IR, occurred more rapidly in the HFD-fed Noc4l[LKO] mice than in the WT mice. In addition, the Noc4l[LKO] mice exhibited a proinflammatory phenotype characterized by increased M1 (LPS/TLR4) and decreased M2 (IL-4) responses. However, in contrast to the macrophage specific Noc4l deficiency mice, no obvious difference of inflammatory cytokines expression in Noc4l macrophage-overexpressed mice compared to WT mice was observed. Consistent with the important role of inflammation in obesity-induced insulin resistance, therefore, we verified that insulin response is similar between WT and OE mice under HFD fed. Nevertheless, OE mice improved glucose metabolism with better glucose clearance in glucose tolerance test, likely by enhancing the insulin sensitivity of peripheral tissues under higher glucose concentration.

It is known that diet can induce thermogenesis to keep energy balance. Noc4l deficiency reduced energy expenditure to cause increased weight gain upon HFD through a non-activity-based mechanism. Previous studies reported that alternatively activated macrophages express tyrosine hydroxylase (TH) and produce catecholamines to regulate HFD-induced obesity[57]. However, this point is still a controversial issue. Some studies indicated that macrophages isolated from adipose tissue of cold challenged mice did not express TH[58,59]. Therefore, we speculated that Noc4l deletion in macrophage may inhibit HFD-induced thermogenesis

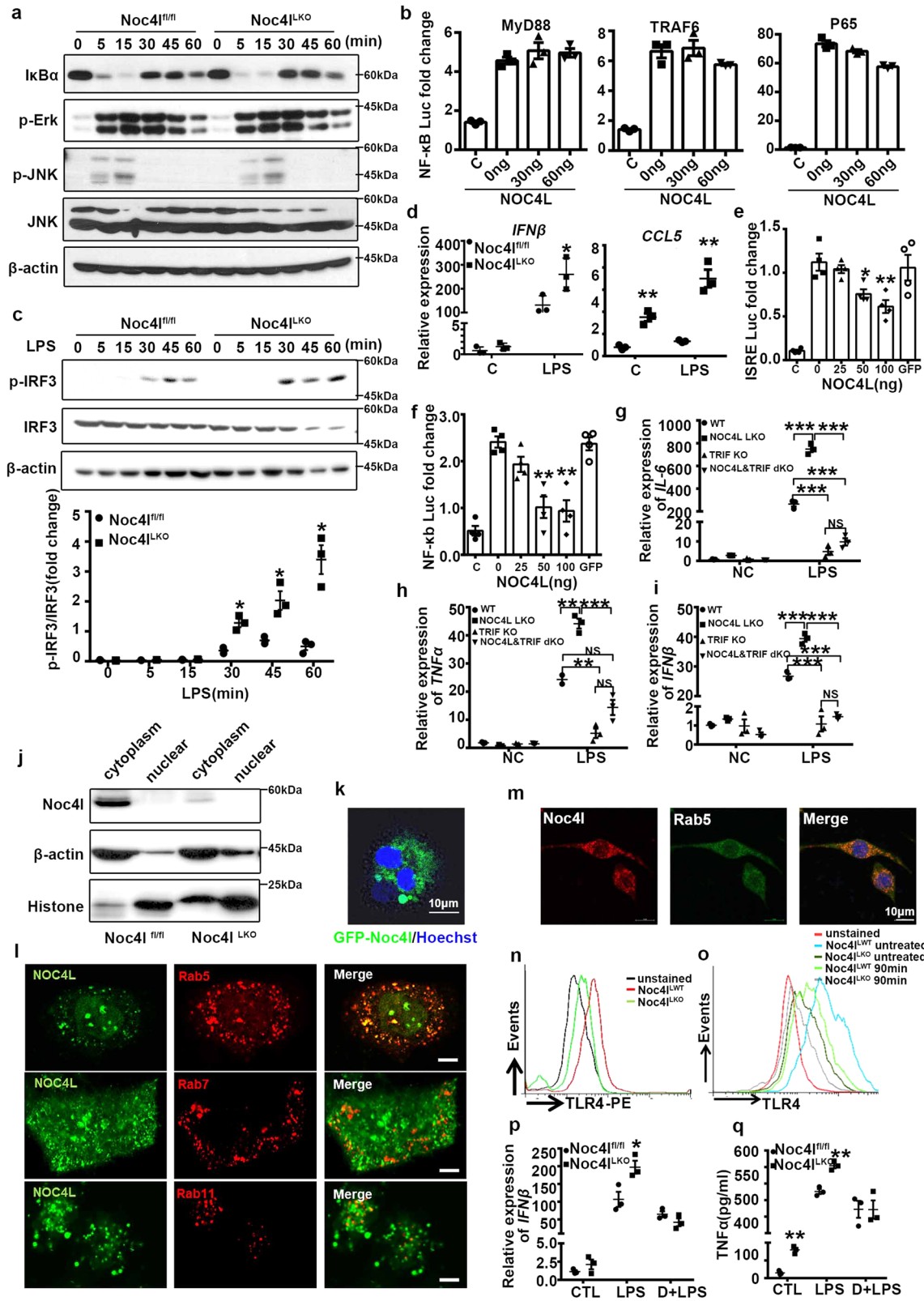

in the white adipose tissues to suppress energy expenditure. However, the mechanism of how Noc4l regulating energy balance needs to be studied further.

Activation of TLR4 signaling pathways triggered by LPS plays a predominant role in obesity-induced inflammation[23,60]. As a receptor of LPS, TLR4 induces two distinct signaling pathways[28,30]. On one hand, the TLR4 signaling pathway is activated from the plasma membrane after coupling with LPS[61]. This pathway is mediated by TIRAP and MyD88[62], which induces proinflammatory cytokines expression by activating NF-κB and AP-1[63]. On the other hand, the LPS-coupled TLR4 is then internalized into the endosome and triggers TRAM and the TRIF signaling pathway[28,30] which mediates the activation of IRF3 by regulating Type I IFN expression[63]. We found that Noc4l

**Fig. 5 Noc4l restrained TRIF pathway by inhibiting TLR4 endocytosis. a** Protein expression of IκBα, p-Erk, p-JNK, and JNK. Total cell lysates from Noc4l[fl/fl] and Noc4l[LKO] BMDMs were extracted at 0, 5, 15, 30, 45, and 60 min after treatment with 100 ng/mL LPS. Data represented two independent experiments. **b** Luciferase assay of HeLa cells transfected with the NF-κB reporter plasmid and MyD88, TRAF6 or P65 expression plasmid, co-transfected with the increasing amount of NOC4L plasmid ($n = 3$ biological replicates). Data represented three independent experiments. **c** Protein expression of IRF3 and p-IRF3 in BMDMs from Noc4l[fl/fl] and Noc4l[LKO] mice. Quantification of the phosphorylation of IRF3 (p-IRF3) was shown in the bottom panel. Data represented three independent experiments. Two-tailed Student's t test. *$p = 0.036$ (30 min), *$p = 0.014$ (45 min), *$p = 0.042$ (60 min). **d** Relative mRNA expression of IFNβ and CCL5 of BMDMs from Noc4l[fl/fl] and Noc4l[LKO] mice incubated with 100 ng/mL LPS for 24 h ($n = 3$ per group). Two-tailed Student's t test. *$p = 0.0462$ (IFNβ); **$p = 0.0020$ (C- CCL5), **$p = 0.0021$(LPS- CCL5). **e, f** Luciferase assay of HeLa cells transfected with the ISRE **e** and NF-κB **f** reporter plasmid and TRIF, co-transfected with the increasing amount of NOC4L plasmid or GFP plasmid (negative control) ($n = 4$ biological replicates). Data represented three independent experiments. Two-tailed Student's t test. *$p = 0.0176$ (50 ng), **$p = 0.0062$ (100 ng) **e**; **$p = 0.0017$ (50 ng), **$p = 0.0012$ (100 ng) **f**. **g–i** Relative mRNA expression of IL-6 **g**, TNFα **h**, and IFNβ **i** of BMDMs from Noc4l[fl/fl], Noc4l[LKO], TRIF KO and Noc4l&TRIF dKO mice incubated with 100 ng/mL LPS for 6 h ($n = 3$ per group). Statistical analysis was performed with two-way ANOVA. ***$p = 0.0002$ (Noc4l[LKO] vs WT), ***$p = 0.0003$ (TRIF KO vs WT), ***$p = 0.0003$ (Noc4l&TRIF dKO vs WT) **g**; **$p = 0.0019$ (Noc4l[LKO] vs WT), **$p = 0.0020$ (TRIF KO vs WT), *$p = 0.0432$ (Noc4l&TRIF dKO vs WT) **h**; ***$p = 0.0006$ (Noc4l[LKO] vs WT), ***$p = 0.0002$ (TRIF KO vs WT), ***$p = 0.0004$ (Noc4l&TRIF dKO vs WT), ***$p = 0.0003$ **i**. **j** Nuclear and cytosolic extracts were prepared from BMDMs of Noc4l[fl/fl] and Noc4l[LKO] mice and then analyzed for the expression levels of Noc4l, Histone and β-actin by immunoblot analysis. **k** Fluorescence analysis of BMDMs transfected with lentivirus of EGFP-Noc4l. **l** Fluorescence microscopy analysis of HeLa cells co-transfected by EGFP-NOC4L and mCheery-Rab5/7/11. Scale bar, 5 μm. **m** Immunofluorescence staining for Noc4l (3L7, red), Rab5 (green), and DAPI (blue) in BMDMs. Data are representative of independent three experiments (Fig. 5j—m). **n, o** Flow cytometry of TLR4 surface staining on PMs **n** or treated with 100 ng/mL LPS for 0, 90 min **o** from Noc4l[fl/fl] and Noc4l[LKO] mice ($n = 3$ per group). Gating strategy for flow cytometry was shown in Supplementary Fig. 6b. **p** Relative expression of IFNβ of BMDMs from Noc4l[fl/fl] and Noc4l[LKO] mice pre-treated with dynasore (80 μM) for 1 h and stimulated with LPS (100 ng/ml) for 6 h ($n = 3$ biological replicates). Two-way ANOVA. *$p = 0.0287$. **q** Concentration of TNFα in the supernatant secreted by BMDMs from Noc4l[fl/fl] and Noc4l[LKO] mice pre-treated with dynasore (80 μM) for 1 h and stimulated with LPS (100 ng/ml) for 6 h ($n = 3$ biological replicates). Two-way ANOVA. **$p = 0.006$ (CTL), **$p = 0.002$ (LPS). Data represented three independent experiments. All data are presented as mean ± SEM. *$p < 0.05$, **$p < 0.01$, ***$p < 0.001$. Source data are provided as a Source Data file.

negatively controlled TLR4 internalization by directly interacting with TLR4. By blocking the TLR4-mediated MyD88-dependent pathway, that mice lacking TLR4 or MyD88, or JNK or IKKβ are protected from HFD-induced IR and inflammation[23,60,64,65]. Our results indicated that by blocking the TRIF-dependent pathway inflammatory signaling, the myeloid-specific overexpression of Noc4l improved glucose homeostasis in the HFD-fed mice relative to that of Noc4l[fl/fl]. Noc4l deletion increased the internalization of TLR4, resulting in the activation of the TRIF signaling pathway and increasing the production of IFNβ, which led to IR. The inflammatory reaction of Noc4l[LKO] BMDMs was remitted and blood glucose was abrogated by TRIF deletion. Both the results of the ITT of weight matched Noc4l[fl/fl] and Noc4l[LKO] mice and the GTT of LPS challenged mice suggested that it was the change in inflammation but not in body weight that is causing IR.

Overall, we demonstrated here a possible mechanism explaining the molecular function of NOC4L in insulin sensitivity (Fig. 7b). NOC4L was localized in the endosome of BMDMs and played a potential role in TRIF-dependent pathway by directly interacting with TLR4. It might inhibit the endocytosis of TLR4, thus reducing the production of IFNβ and proinflammatory cytokines, ameliorating the LSI and IR. NOC4L deficiency in macrophage activated the transcription factor IRF3 and increased proinflammatory cytokines, which in turn lead to IR.

## Methods

**Mice.** Noc4l[fl/fl] mice were provided by Prof. George Fu Gao (Institute of Microbiology, Chinese Academy of Sciences, China). TRIF KO mice were provided by Prof. Feng Shao (Beijing's National Institute of Biological Sciences, China). TLR4 mutant mice and db/db mice were purchased from Jackson Laboratory. Noc4l[fl/fl] mice were bred to LysM-cre mice in order to generate myeloid-specific Noc4l knockout (Noc4l[LKO]) mice. Genotyping was performed by PCR[17]. Myeloid-specific Noc4l overexpressing mice, referred to as "Noc4l[LOE]," were generated by Nanjing Biomedical Research Institute of Nanjing University. The human lysozyme promoter was fused to the 5′-end of the Noc4l sequence, and the polyadenylation (Poly A) sequence was attached at the 3′-end of the Noc4l sequence. A chicken beta-globin insulator was constructed in front of the human lysozyme promoter and behind the Poly A sequence to prevent position effects. Under the human lysozyme promoter, Noc4l was overexpressed in mouse myeloid cells[66]. The Noc4l[LKO] mice

were bred with TRIF KO mice to generate Noc4l[−/−]/TRIF[−/−] dKO mice. The genotyping primers of each KO mice are listed in Supplementary Table 1.

The animals were housed under specific pathogen-free conditions. All mice were maintained under a 12 h light/12 h dark cycle with free access to food and water in a temperature (22 ± 1 °C) and humidity (50 ± 5%) controlled room. All experiments were conducted in accordance with the principles for the care and use of laboratory animals and approved by the ethics committee of China Agricultural University (reference number SKLAB-2014-01-13). Male mice were fed with CD or HFD (60%, D12492, Research Diets Inc., USA) ad libitum from 6 weeks of age. The body weights were measured weekly.

**Human subjects.** Visceral adipose tissues (omental and mesenteric) were collected from patients who underwent laparotomy for neoplastic disease at a site remote from the collection of visceral adipose tissues. Patients diagnosed with metabolic syndrome, including IR, T2D, fatty liver disease, and atherosclerosis excluded. The lean patients had a body mass index (BMI) ≤ 25 kg/m². Patients with BMI ≥ 28 kg/m² were considered obese[67]. Difference definitions of obesity have been used in different countries[68]. In China, the two most commonly used BMI classification for adults is the WHO standard (BMI ≥ 30 kg/m²) and the recently developed Chinese standard (BMI ≥ 28 kg/m²)[67]. The lower BMI is recommended in China because a growing body of evidence suggests that people in China, as well as several other Asian Pacific populations, have an elevated risk for obesity-related diseases or conditions at a lower BMI than Caucasians[68]. Patients' information was listed in Supplementary Table 2. The study was conducted with approval from the Ethics Committee of the 306th Hospital of PLA (Beijing, China) with approval No.2014-KLS-03. The study was carried out in accordance with the Declaration of Helsinki. Written informed consent was obtained from each patient.

**Cell lines.** The mouse cell lines RAW 264.7 (ATCC® TIB-71™), L929 (ATCC® CCL-1™), GC-1(ATCC® CRL-2053™), GC-2(ATCC® CRL-2196™), NIH3T3 (ATCC® CRL-6442), 3T3-L1(ATCC® CL-173™), the human cell line THP-1 (ATCC® TIB-202), HeLa (ATCC® CCL-2) and HEK293T (ATCC® CRL-11268) and the insect cell line Sf 9 (ATCC® CRL-1711™) were purchased from ATCC. THP-1 cells were differentiated using PMA (Sigma-Aldrich, Dorest, UK) at a final concentration of 100 ng/mL and incubated for 48 h. THP-1 cell differentiation was enhanced by removing the PMA-containing media and adding fresh media for a further 24 h. Mycoplasma contamination was assessed using a MycoFluor™ Mycoplasma Detection Kit (M7006, Invitrogen™, USA), and cells used for the experiments were free of mycoplasma contamination. All experiments were three independent replicates.

**Expression vectors.** Human pcDNA3.1/FLAG-MyD88, pcDNA3.1/FLAG-TRAF6, and pcDNA3.1/FLAG-p65 plasmids were provided by Professor Zhijie Chang (Tsinghua University, China). Human NOC4L were amplified from cDNA templates of HeLa cells and were further subcloned into pcDNA4.0 (Invitrogen, USA) and PC2AOE-3×FLAG expression vectors. The PC2AOE-3×FLAG vector

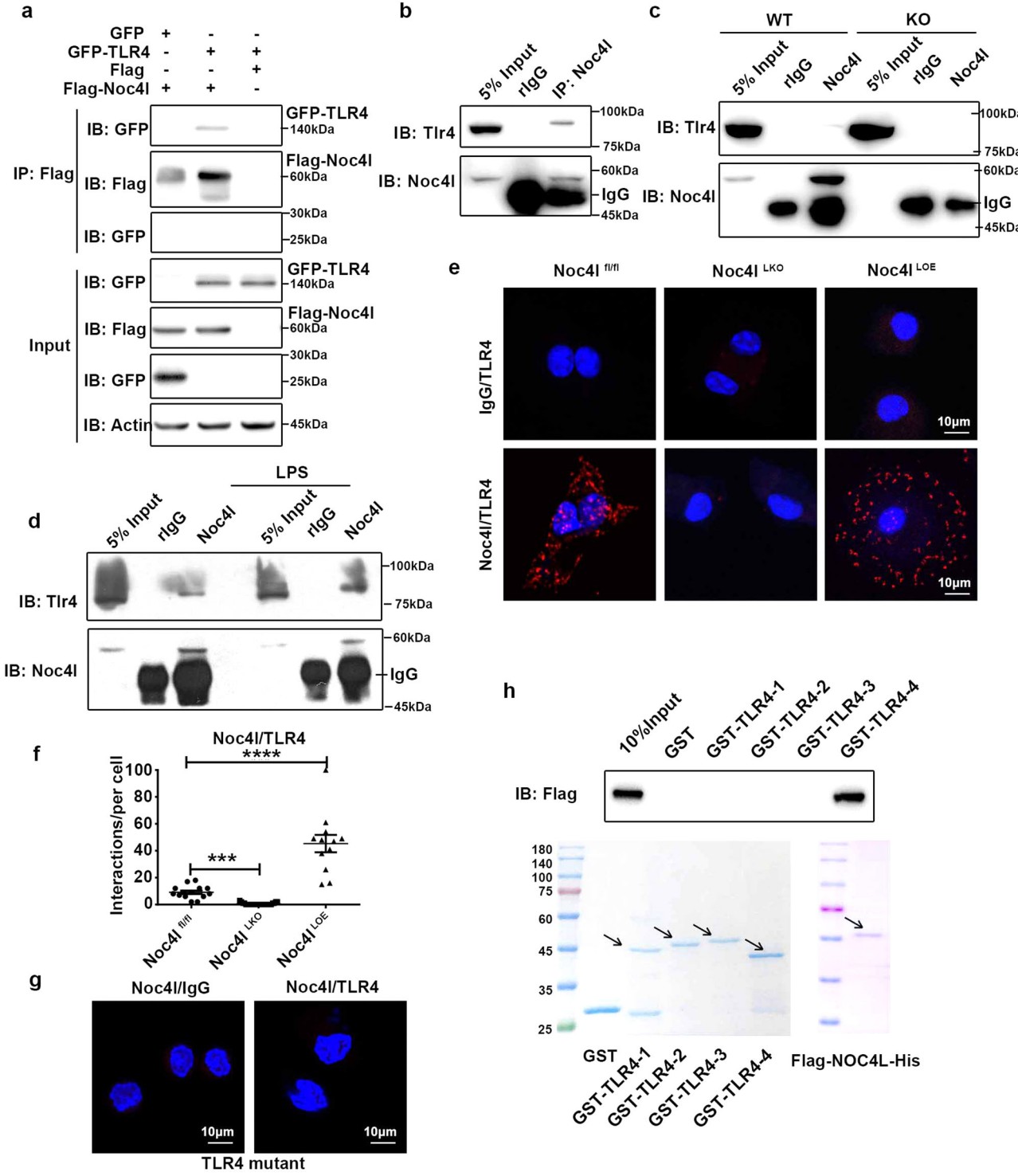

was constructed using the overlap extension PCR method as described in a previous study[69], containing a 3×FLAG sequence following a CMV promoter. The pGEX and pNF-κB-Luc vectors were purchased from BD Biosciences-Clontech. Human TLR4 was amplified from the cDNA template of HepG2 cells and then subcloned into the pEGFP-C1 vector (Invitrogen, USA).

**Generation of monoclonal and polyclonal antibody against NOC4L.** The VHRPQGPELDADPYD Peptides were synthesized (Abclonal Company, China) and were used for immunization. New Zealand white rabbits and Balb/c mice were immunized with the VHRPQGPELDADPYD peptide coupled with Freund's complete adjuvant. We administered four booster doses at two-week intervals. Blood was drawn from the immunized rabbit and mice two weeks after the fourth booster, and the rabbit polyclonal antibody was obtained after affinity purification by using the VHRPQGPELDADPYD peptide. The titer of the purified rabbit

polyclonal antibody and mouse antibody were tested by ELISA using the VHRPQGPELDADPYD peptide. Selected the mouse with the best results of endogenous WB test to fuse spleen cells and myeloma cells, and hybridoma cells were screened by HAT. The positive monoclones were expanded and inoculated into mice to prepare ascites for purification.

**Immunofluorescence staining.** Formalin-fixed paraffin-embedded mouse and human adipose tissue sections were deparaffinized and rehydrated prior to antigen retrieval by boiling in 10 mM sodium citrate with pH 6.0 for 20 min and then cooled at RT. Samples were incubated with 10% normal goat serum (ZSGB-BIO, China) in PBS for 1 h at RT to block unspecific binding sites. After blocking, the sections were incubated with the anti-Noc4l antibody (3L7, 1:100), anti-F4/80 (1:50 dilution, Biolegend, USA), or anti-Mac-2 (1:100 dilution, Biolegend, USA) overnight at 4 °C. After three washes in PBS, secondary antibodies, Alexa Fluor594 or

**Fig. 6 NOC4L directly interacted with TLR4. a** HeLa cells were co-transfected with NOC4L plasmids with 3×FLAG tag and EGFP–TLR4 plasmids, and cell lysates were prepared for co-immunoprecipitation. Cell lysates (Input) and FLAG immunoprecipitation were determined by immunoblot analysis for FLAG and GFP. **b** Endogenous Noc4l and TLR4 interaction in RAW264.7 cells. Cell lysates (Input) and Noc4l (6 R) immunoprecipitation analyzed by immunoblot analysis for TLR4 and Noc4l. **c** Endogenous Noc4l and TLR4 interaction in BMDMs isolated from Noc4l^fl/fl and Noc4l^LKO mice. Cell lysates (Input) and Noc4l (6 R) immunoprecipitation analyzed by immunoblot analysis for TLR4 and Noc4l. **d** Endogenous Noc4l and TLR4 interaction in RAW264.7 cells stimulated with LPS (200 ng/mL). Cell lysates (Input) and Noc4l (6 R) immunoprecipitation analyzed by immunoblot analysis for TLR4 and Noc4l. **e** Representative images of Proximity Ligation Analysis (PLA). PMs of Noc4l^fl/fl, Noc4l^LKO, and Noc4l^LOE mice were stained with the rabbit anti-TLR4 antibody and mouse anti-Noc4l antibody (3L7) or IgG antibody, and in vivo protein–protein interaction between TLR4 and Noc4l (denoted by a red dot) was detected with secondary proximity probes, Anti-Rabbit MINUS and Anti-Mouse PLUS, using the Duolink protein–protein interaction detection kit in vivo. **f** Quantification of positive signals in each cell of each group in **e**. Two-tailed Student's t test. ****p < 0.0001. **g** PLA was performed in PMs of WT and TLR4 mutant mice as described in **e**. **h** Indicated truncates of TLR4 were constructed according to their functional domains. Full-length Flag-NOC4L-His and GST-TLR4 were expressed and purified from Sf9 cells and BL21 respectively, and then GST pulldown assays were performed. Purified GST-TLR4 proteins and Flag-NOC4L-His proteins were incubated with GST beads overnight at 4 °C. NOC4L was analyzed by Western blotting using anti-Flag antibody. GST-TLR4-1: 1–199aa, GST-TLR4-2: 200–423aa, GST-TLR4-3: 424–673aa, GST-TLR4-4: 674–839aa. Data represented three independent experiments. All data are presented as mean ± SEM. *p < 0.05, **p < 0.01, ***p < 0.001. Source data are provided as a Source Data file.

---

FITC-conjugated goat anti-mouse IgG (Thermo fisher, USA, 1:500), and FITC or Alexa Fluor594-conjugated goat anti-rat IgG (Thermo fisher, USA, 1:500) were applied for immunofluorescence staining. After the tissues were washed with PBS, they were mounted in the anti-fading medium containing 4′,6′-diamidino-2-phenylindole (DAPI) (Santa Cruz, USA). The stained sections were photographed under a confocal microscope (Olympus, Japan).

**Isolation of adipocytes, SVFs, and ATMs**. The mouse epididymal adipose tissue was isolated and minced in PBS with calcium chloride containing 0.5% BSA. Collagenase II (1 mg/mL, C6885, Sigma, USA) was added, and the tissue was incubated at 37 °C with shaking at 75 rpm for 30–60 min. Tissue suspensions were filtered using 250 μm nylon sieves to remove larger particles, and the filtrates were centrifuged at $500 \times g$ for 5 min to separate floating adipocytes. Pelleted SVFs were suspended in erythrocyte lysis buffer (155 mM NH4Cl, 10 mM KHCO₃, and 0.1 mM EDTA) and then incubated at RT for 5 min. After erythrocyte lysis, the SVFs were spun at $500 \times g$ for 5 min. The adipocytes and the pelleted SVFs were dissolved in TRIzol (Invitrogen, USA) for RNA isolation.

To isolate ATMs, the CD11b microbeads (MACS, Germany) were used according to the manufacturer's instructions. In brief, the pelleted SVFs were suspended in PBS containing 0.5% BSA and 2 mM EDTA and incubated with CD11b microbeads (10 μL/$10^7$ cells) for 15 min at 4–8 °C. Then, cell suspensions were passed through MS Column which was placed in the magnetic field of a MACS Separator. CD11b- cells were then collected in the both flow-through and column washing buffer. CD11b + cells were collected from the column removed from the MASC Separator. The purities of isolated cells were more than 90% by flow cytometric analysis.

**GTT and ITT**. GTT and ITT were conducted following the procedure described[70]. In Brief, GTT was performed in the CD-fed or HFD-fed mice for 20 weeks. The same mice were used for ITT after 1 week. For GTT, mice were fasted overnight. After measuring the baseline blood glucose level via a tail nick using a glucometer (ACCU-CHEK, Switzerland), 2 g Kg⁻¹ BW of 20% glucose was intraperitoneally injected into mice. Blood glucose levels were then measured at 30, 60, 90, and 120 min after glucose injection. For ITT, mice were fasted for 5 h and then injected intraperitoneally with recombinant human insulin (Eli Lilly, USA) at 0.75 U Kg⁻¹ BW. Their blood glucose concentrations were determined in a time-course period from tail blood by using a glucometer (pre-insulin administration and 15, 30, 60, and 120 min post-insulin administration).

**Indirect calorimetry**. After 10 weeks of HFD or CD feeding, the Noc4l^fl/fl and Noc4l^LKO mice were housed individually in indirect calorimeter chambers (Panlab, Spain). Calorimetry, daily body weight and daily food intake data were collected during a 3-day acclimation period, and then during a 1-day experimental period. Oxygen consumption (VO2), carbon dioxide production (VCO2), and EE were evaluated. EE was calculated as described below and normalized relative to the body mass of each mouse:

$$RQ = VCO_2/VO_2$$

$$EE = (3.815 + (1.232 \times RQ)) \times VO_2 \times 1.44$$

**Analysis of adipose tissue sections**. Mouse epididymal adipose tissues were fixed in 4% PFA, dehydrated, and embedded in paraffin. Sections (5 μm) were cut and stained using hematoxylin and eosin (H&E). To detect crown-like structures in adipose tissues, immunohistochemical staining of F4/80 was performed. Adipose tissue sections were deparaffinized, rehydrated, and treated for antigen retrieval following a previously described procedure[71]. Endogenous peroxidase activity was

blocked for 10 min at RT by using 3% H₂O₂. After washes, the sections were blocked in 10% normal goat serum for 30 min at RT and then incubated with the primary antibody against F4/80 (1:100, Biolegend, USA) overnight at 4 °C. The secondary antibody biotinylated goat anti-rat IgG (1:200, ZSGB-BIO, China) was incubated for 30 min at RT, followed by incubation with horseradish peroxidase–streptavidin (ZSGB-BIO, China). Specific binding was visualized using 3′,3′-diaminobenzidine tetrahydrochloride, and the reaction was stopped by rinsing in deionized water. Sections were slightly counterstained with Mayer's hematoxylin following dehydration and then sealed with neutral resins. All images were visualized and captured with a Zeiss microscope (Imager ZI, Germany) equipped with a digital camera (Axiocam, Germany).

**qRT-PCR**. Total RNA from mouse tissues and cultured cells was extracted with TRIzol (Invitrogen, USA) in accordance with the instructions provided by the manufacturer. cDNA was prepared from 1 μg total RNA, using the oligo dT and M-MLV reverse transcriptase (TAKARA, Japan) as instructed by the manufacturer. Real-time qRT-PCR was performed using SYBR Green I master mix (Roche, Switzerland) in accordance with standard protocols on the LightCycler 480 (Roche, Switzerland). The cycling conditions were 95 °C for 5 min, followed by 40 cycles at 95 °C for 10 s, 60 °C for 20 s, and 72 °C for 10 s. Gene expression levels were normalized to GAPDH. Fold change in mRNA expression was calculated using the comparative cycle method ($2^{-\Delta\Delta t}$). The primers used are listed in Supplementary Table 3.

**Immunoblot analysis of insulin signaling in WAT, liver, and muscle tissues**. The Noc4l^fl/fl and Noc4l^LKO mice fed with HFD for 20 weeks were fasted overnight[60]. Vehicle saline or 5 U kg⁻¹ BW of recombinant human insulin (Eli Lilly, USA) was injected intraperitoneally. The liver, epididymal fat, and muscle tissues were immediately collected in 10–15 min and snap-frozen in liquid nitrogen. Tissues were stored at −80 °C until analysis. For analysis of AKT (Ser473) phosphorylation, tissues were homogenized in RIPA lysis buffer containing phenylmethylsulfonyl fluoride (PMSF, Roche, Switzerland) and protease inhibitor cocktail (P2714, Sigma, USA), lysed for 15 min on ice, followed by centrifugation at $15,000 \times g$ for 15 min at 4 °C. Supernatants were collected, and tissue protein concentrations were quantified by bicinchoninic acid kit (Pierce, USA). The extracted proteins were separated by 10% SDS–polyacrylamide gel electrophoresis (SDS–PAGE). Proteins were transferred to a polyvinylidene difluoride membrane by electroblotting. The membrane was blocked with 5% nonfat milk in Tris-buffered saline (TBS) containing 0.05% Tween-20 (TBST) overnight at 4 °C. The membrane was incubated with the primary antibody, anti-phospho-AKT antibody (Thr308, 9331 S, Cell Signaling Technology, USA), and anti-AKT antibody (9272 S, Cell Signaling Technology, USA) at 1:2,000 dilution, respectively, overnight. After washing with TBST, the membrane was incubated with the secondary antibody horseradish peroxidase-conjugated goat anti-rabbit IgG (1:5000) (ZSGB-BIO, China) for 1 h at RT. After incubation with enhanced chemiluminescence (Pierce, USA), the membranes were exposed for image development. Image J 1.8.0 software was used to quantify bands of western blotting.

**Isolation of PMs and BMDMs**. To isolate PMs, CD-fed mice aged 8–12 weeks were injected with 3 mL 3% thioglycolate (Sigma, USA). After 4 d, the mice were anaesthetized and injected intraperitoneally with 5 mL ice-cold phosphate-buffered saline (PBS, Sigma, USA). The elicited PMs were collected and centrifuged at $300 \times g$ for 10 min at 4 °C. Cell pellets were resuspended and seeded in six-well plates in RPMI 1640 medium (Sigma, USA) containing 10% fetal bovine serum (FBS, Gibco, USA), 100 U mL⁻¹ penicillin, and 100 μg mL⁻¹ streptomycin (Sigma, USA). After 2 h, the cells were washed twice with PBS, and adherent macrophages were used for experiments.

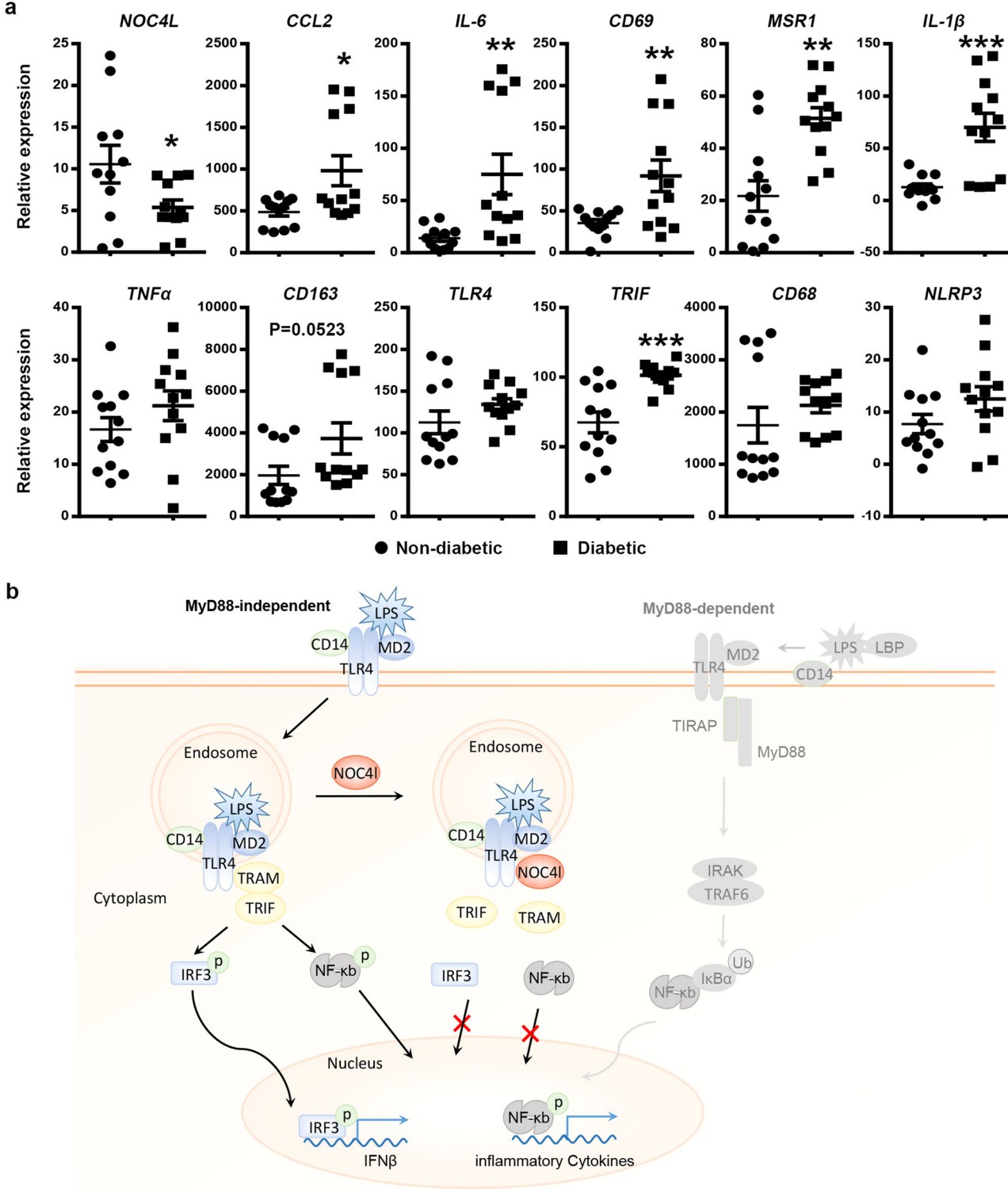

**Fig. 7 Hypothetical working model of Noc4l and its clinical association with diabetes. a** The mRNA expression levels of *NOC4L* and selected inflammatory genes, *CCL2, IL-6, CD69, MSR1, IL-1β, TNFα, CD163, TLR4, TRIF, CD68* and *NLRP3* were determined in the visceral adipose tissues of patients with diabetes ($n = 12$ per group) vs. non- diabetes ($n = 12$ per group). All data are presented as mean ± SEM. Two-tailed Student's t test. *$p = 0.0381$ (*NOC4L*), *$p = 0.0146$ (*CCL2*), **$p = 0.0049$ (*IL6*), **$p = 0.008$ (*CD69*), **$p = 0.004$ (*MSR1*), ***$p = 0.0004$ (*IL-1β*), ***$p = 0.0004$ (*TRIF*). *$p < 0.05$, **$p < 0.01$, ***$p < 0.001$. **b** NOC4L was located in the early endosome of macrophages. It blocked the endocytosis of TLR4, thereby inhibiting TRIF signaling, reducing IFNβ production, and reducing systemic inflammation. Source data are provided as a Source Data file.

Isolation of BMDMs was conducted using the procedure described in previous studies[72,73]. The mice were sacrificed by cervical dislocation, and both the femur and tibia were separated. After all muscle tissues were removed from the bones, the bones at both ends were cut, and the bone marrows were crushed with a 25-gauge needle attached to a 10 mL syringe. After washing three times with ice-cold PBS and centrifugation at 500 g for 5 min at 4 °C, the bone marrows were filtered through a 500 μm nylon mesh. The bone marrow-derived cells were centrifuged, resuspended in differentiation media (DMEM supplemented with 30% L929 supernatant, 10% FBS, 100 U·mL$^{-1}$ penicillin, and 100 μg mL$^{-1}$ streptomycin) and plated onto 10 cm cell culture plates. Fresh differentiation media were added to the cultured cells on Day 3 and Day 5. The medium was changed to fresh DMEM containing 10% FBS on Day 6, and BMDMs were attached to the bottom of the culture dishes. The cells were plated at a density of $1.2 \times 10^6$ cells/well in six-well plates and then cultured in the RPMI medium containing 10% FBS. Studies were performed using BMDMs incubated with 100 ng/mL LPS for 16 h, 10 μg/mL Poly I:C for 8 h, 0.5 mM palmitate, 5 μg/mL LTA, 10 μg/ml Poly U for 24 h or 10 ng/mL IL-4 for 48 h. LPS, Poly I:C, Palmitate, LTA and Poly U were purchased from Sigma (Sigma, USA), IL-4 was from PeproTech, USA.

**Cytokine secretion**. Plasma levels of cytokines (IL-6 and TNFα) were subjected to measurements using Mouse IL-6 and TNFα ELISA Kit (Invitrogen, USA), following the instructions provided by the manufacturer. Supernatants of BMDMs after treatment with LPS were collected and then centrifuged at 1,000 × g for 10 min to remove cell debris. The concentration of IL-6 and TNFα in the supernatant was measured using Mouse IL-6 and TNFα ELISA Kit (Invitrogen, USA).

**Transfection and luciferase reporter assay**. HeLa cells ($2 \times 10^4$ cells per well) were seeded into 96-well plates and were transfected after 24 h with expression vectors and luciferase reporter genes by using Lipofectamine 3000 (Invitrogen, USA). The transfection procedure was performed as described in a previous study[74]. In brief, 20 ng/well pGL3/NF-κB-luc reporter gene combined with 25 ng/well pcDNA3.1/FLAG-MyD88, or 10 ng/well pcDNA3.1/FLAG-TRAF6, or 10 ng/well pcDNA3.1/FLAG-p65 plasmids, and PC2AOE-3×FLAG-NOC4L expression plasmids in different amounts (0, 30, 60 ng) or PC2AOE-GFP plasmid, was co-transfected into the cells. Similarly, 20 ng/well pGL3/NF-κB reporter genes combined with 20 ng/well pcDNA4.0-TRIF and PC2AOE-3×FLAG-NOC4L expression plasmids in different amounts (0, 25, 50, and 100 ng) or PC2AOE-GFP plasmid were co-transfected into the cells. In addition, 10 ng/well of the phRL-TK reporter plasmid (Promega, USA) was co-transfected to allow normalization of data for transfection efficiency. The total amount of DNA per transfection was kept constant at 150 ng by adding the empty vector PC2AOE-3×FLAG. After transfection for 24 h, luciferase activity was measured using a Dual-Luciferase reporter assay system (Promega, USA) following the instructions provided by the manufacturer. The luciferase activity was normalized against *Renilla* luciferase activity. All reporter assays were completed in 4 repeat wells, and data were presented as mean ± SEM.

**Nuclear and cytoplasmic extraction**. Nuclear and cytoplasmic extraction were performed with the nuclear and cytoplasmic extraction reagents kit (Invent, USA), following the instructions provided by the manufacturer. Briefly, grow adherent cells to 90–100% confluence and wash the cells twice in the tissue culture plates with cold PBS, aspirate the buffer completely. Add the cytoplasmic extract buffer and let stand on ice for 5 min. After blowing several times with a suction head, it is transferred to the pre-cooled 1.5 ml centrifugal tube. Shake violently for 15 s. Centrifuge for 5 min at 4 °C, 14000 × g. The supernatant is the cytoplasmic component. Add the nucleus lysis buffer to the sediment. Shook vigorously for 15 s and incubated on ice for 1 min. Repeat four times. Quickly transfer the nuclear extract into the pre-cooled centrifugal tube casing, 14000 × g, centrifuge for 30 s. Storage of nucleoproteins at −80 °C. Cell lysates were analyzed by Western blot analysis for the indicated proteins.

**Noc4l subcellular localization**. The fragments for NOC4L, the early endosome marker Rab5, the late endosome marker Rab7, and the recycling endosome marker Rab11 were amplified from cDNA templates of HeLa cells. The NOC4L fragments were integrated into the 3′-end of the pEGFP-C1 vector[17], Rab5, Rab7, and Rab11 were constructed into the 3′-end of pmCherry-C1 plasmids[75]. Equal amounts of pEGFP-NOC4L and subcellular localization plasmids Rab5, Rab7, and Rab11 were co-transfected into HeLa cells by Lipofectamine 3000 (Invitrogen, USA). After transfection for 24 h, the cells were washed with PBS and directly imaged under a confocal microscope. In addition, after pEGFP-NOC4L plasmid transfection for 24 h, HeLa cells were stained with ER-Tracker Dye and LysoTracker Dye (Invitrogen, USA) in accordance with the instructions provided by the manufacturer and then observed by confocal microscope.

**Flow cytometric analysis**. To detect the cell surface expression of TLR4, PMs were incubated with the phycoerythrin-labeled antibody against mouse TLR4 (Biolegend, USA) or unstained for 30 min at RT under dark conditions. After PMs were

washed twice, they were analyzed using a FACS calibur flow cytometer (Beckman, USA). For analysis of TLR4 internalization after LPS stimulation, the PMs were treated with 100 ng/mL LPS for 0 and 90 min and then immediately cooled on ice. The remaining TLR4 on the plasma membrane was evaluated by FACS. The expression levels of CD14 in the Noc4l$^{fl/fl}$ and Noc4l$^{LKO}$ PMs were also detected by FACS as earlier described. Data were processed with the FlowJo software (TreeStar, USA).

**Immunoprecipitation**. The HeLa cells were co-transfected with the PC2AOE-3 × FLAG-NOC4L and pEGFP-TLR4 plasmids. After transfection for 48 h, the HeLa cells were lysed in lysis buffer containing 10 mM Tris-HCl (pH 7.8), 0.5% NP-40, 150 mM NaCl, 1 mM EDTA, 1 μM PMSF, and 1 μg/mL proteinase inhibitor cocktail (Sigma, USA). The cells were then incubated for 30 min on ice and then centrifuged at 15,000 × g for 15 min at 4 °C. The whole cell extracts (input) in the supernatant were collected and immunoprecipitated with anti-FLAG M2 magnetic beads (Sigma, USA) in accordance with the instructions provided by the manufacturer. The bound proteins were separated by SDS–PAGE, and immunoblotting was performed using anti-FLAG and anti-GFP (Santa Cruz, USA) antibodies in accordance with standard procedures.

RAW 264.7 and BMDMs cells were lysed in lysis buffer as earlier described. The whole cell extracts were immunoprecipitated with anti-NOC4L (6 R) or rIgG overnight at 4 °C and then incubated with proteinA/G (Santa Cruz, USA) for 2 h. The bound proteins were separated by SDS–PAGE, and immunoblotting was performed using anti-Noc4l in accordance with standard procedures.

**GST-pulldown assay**. GST-TLR4 fusion proteins (GST-TLR4-1 (1-199 aa), GST-TLR4-2 (200-423 aa), GST-TLR4-3 (424-673 aa), GST-TLR4-4 (674-839 aa)) were purified in BL21 following standard protocol. Flag-NOC4L-His protein was purified from Sf9 cells. Glutathione Sepharose 4B was prepared with 0.2% BSA in PBS on a rotator for 1 h at 4 °C. Purified GST-TLR4 proteins and Flag-NOC4L-His proteins were incubated with GST beads overnight at 4 °C. After washed by PBS, the bound proteins were separated by SDS–PAGE.

**Proximity ligation assay (PLA)**. The interaction was detected using the proximity ligation assay kit Duolink (PLA probe Anti-Rabbit MINUS, Cat. #90602; PLA probe Anti-Mouse PLUS, Cat. #90701; Dection Kit 563, Cat. #90134 Olink Bioscience, Uppsala, Sweden). The PLA probe Anti-Rabbit Minus binds to the TLR4 antibody, whereas the PLA probe Anti-Mouse PLUS binds to the Noc4l antibody (3L7). The DuoLink proximity ligation assay secondary antibodies only generate a signal when the two PLA probes have bound, which only occurs if both proteins are closer than 40 nm, indicating their interaction. Methanol-fixed air-dried samples were per-incubated with a blocking agent for 1 h. After washed in PBS for 10 min, the primary antibodies to TLR4 (1:100) and Noc4l (3L7, 1:200) were applied to the samples. Incubation was conducted overnight in a pre-heated humidity chamber. Slides were washed three times in PBS for 10 min. The Duolink PLA probes detecting rabbit or mouse antibodies were diluted in the blocking agent at a concentration of 1:5 and then applied to the slides, followed by incubation for 2 h in a pre-heated humidity chamber at 37 °C. Unbound PLA probes were removed by washing three times in PBS for 10 min. For hybridization of the two Duolink PLA probes, the Duolink hybridization stock was diluted 1:5 in high-purity water, and the slides were incubated in a pre-heated humidity chamber for 15 min at 37 °C. The slides were washed in TBS-T for 1 min under gentle agitation. The samples were incubated in the ligation solution consisting of the Duolink Ligation stock (1:5) and Duolink Ligase (1:40) diluted in high-purity water for 90 min at 37 °C. The amplified probe was detected using the Duolink Detection kit. Duolink Detection stock was diluted 1:5 in high-purity water and applied for 1 h at 37 °C. Final washing was performed using SCC buffer and 70% ethanol.

**Lv-Noc4l transfection and treatment**. The sequence of mouse Noc4l CDS was inserted into pLV-EGFP-2A-puro, which was verified by sequencing. The plasmid was then co-transfected with psPAX2 and pMD2.G into HEK-293T cells. We collected the lentivirus after transfection for 48 and 72 h. The lentivirus was filtered with a 0.45 μm filter to remove the cellular debris. The mice were injected with lentiviruses (10E9 TU/ml) expressing either EGFP alone or mouse Noc4l.

**Oil red oxygen dyeing**. Oil red oxygen dyeing was performed with the modified oil red oxygen staining kit (Solarbio Life Sciences, Beijing, China), following the instruction provided by the manufacturer. Firstly, the 7 μm thickness sections of frozen livers were washed after 10 minutes of 10% formalin immobilization. Second, slices were washed with water and soaked in 60% isopropanol for 20–30 s. Third, slices were stained in modified oil red O staining solution (capped) for 10–15 min and washed slightly in 60% isopropanol to remove the dye solution. At last, Mayer staining solution was re-stained for 1–2 min and the slices were washed with water and sealed with glycerol.

**Polysome profiling**. Methods were adapted and modified from previous studies (Khajuria et al., 2018; Poria and Ray, 2017; Zhu et al., 2019). Cells were collected

and lysed in lysis buffer (10 mM Tris-HCl, pH 7.4, 5 mM $MgCl_2$, 100 mM KCl, 2 mM DTT, 100 μg/mL cycloheximide, 0.5% NP-40, 200 U/mL RNase inhibitor, protease inhibitor cocktail). Supernatants were separated on a 10–50% linear sucrose gradient (20 mM Tris-HCl, pH 7.4, 5 mM $MgCl_2$, 100 mM KCl, 2 mM DTT, 100 μg/mL cycloheximide) and separated by centrifugation at $260,000 \times g$ for 2 h at 4 °C in an SW-41 rotor (Beckman Coulter). Gradients were analyzed by passage through a UV monitor (Biocomp), and fixed-volume fractions were collected with a density gradient fractionator.

**Statistics**. Data are presented as mean ± SEM unless otherwise specified. Data were analyzed for statistical significance by using SPSS 12.0.1 and GraphPad Prism 6 (GraphPad Software Inc., USA). Data for all groups were first tested for normality with the Shapiro–Wilk test. If the group data were normally distributed, they were compared using Student's t test or two-way ANOVA, Tukey was used to perform the multiple comparisons test. All figures were generated using GraphPad Prism 6. All reported $p$ values are two-tailed unless stated otherwise. $P < 0.05$ was considered to indicate statistical significance.

**Reporting summary**. Further information on research design is available in the Nature Research Reporting Summary linked to this article.

## Data availability

Microarray data that support the findings of this study (Supplementary Fig. 4) have been deposited in Gene Expression Omnibus (GEO) with the primary accession code GSE141447. Figure 7a was obtained from publicly available data (GSE29231). The authors declare that the data supporting the findings of this study are available within the paper and its Supplementary information files. All datasets generated and/or analyzed during the current study are available from the corresponding author on reasonable request. Source data are provided with this paper.

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

## Acknowledgements

This study was supported by grants from the National Key Research and Development Project (Grant/Award Numbers: 2018YFC1004702), National Natural Science Foundation of China (Grant/Award Numbers: 31970802), Beijing Municipal Natural Science Foundation (Grant/Award Numbers: 7202099), Medical University of Bialystok, Poland (Grant/Award Numbers: SUB/1/DN/20/006/1104). We thank George Fu GAO for kindly providing the Noc4l<sup>fl/fl</sup> mice and Feng Shao for providing the TRIF KO mice.

## Author contributions

X.L. conceived the project and designed the experiments. Y.Q. and L.J. performed the experiments and wrote the manuscript. F.R., D.L., Y.G., D.Y.G., and N.R. analyzed the data and modified the manuscript. W.M. and X.R. prepared for Lv-Noc4l to treat the IR mouse model. F.H.L., Y.L., W.H.L., S.W., and A.K. contributed analytical tools. H.Y., X.R.L., and M.L. collected human subjects. P.L. collected db/db mouse tissues. S.M., W.M., X.R., and F.H.L. raised the mice. J.Z., and Y.G. contributed new reagents. J.H.L. performed the GST-pull down assay. All authors discussed the results and prepared the manuscript.

## Competing interests

The authors declare no competing interests.
