## [Peer Review File · Nature Communications]

Reviewer comments, first round –

Reviewer #1 (Remarks to the Author):

GENERAL:

The authors identify a very novel link between NOC4L (a protein involved in 40S ribosome biosynthesis/nuclear export) and macrophage inflammatory responses, particularly in the context of obesity and metabolic disease. The study represents a very significant body of work, with analyses spanning in vivo models to detailed molecular mechanisms. In general, the in vivo findings relating to obesity and inflammation appear to be clear and appropriately interpreted. Some of the in vitro observations are also novel e.g. cytoplasmic localization of NOC4L in macrophages. However, I have several concerns with the mechanistic data linking NOC4L to TLR4 and TRIF signalling – for these studies, many controls and details are lacking, and more robust data are required to support the author's claims about mechanisms. There are many other concerns relating to data, methods etc that should also be thoroughly addressed by the authors. Finally, although this study is very likely to open up exciting new areas of research with respect to macrophage biology and inflammation, it is also noted that a role for NOC4L in regulating inflammation via control of Tregs was very recently described by Zhu et al (Cell Reports, April 2019; PMID: 31018134). This new study should be described and discussed, given its relevance to the current work (though it is acknowledged that the Treg paper was likely published after the current manuscript was submitted). Nonetheless, the novelty of the link between NOC4L and macrophage inflammatory responses is a key strength of this manuscript.

MAJOR:

1. Very little information is provided in the Introduction about how NOC4L regulates 40S ribosome biosynthesis/nuclear export (e.g. pathway, mechanisms, interacting proteins, key domains of NOC4L and how these function). Providing such information in the introduction would provide more context for the studies that were undertaken. Moreover, assessing ribosome/polysome profiles in wild type versus Noc4l knock-out macrophages (e.g. by sucrose gradient) would seem like an important control for the studies on the myeloid-deleted and over-expressing mice.

2. The TLR4/NOC4L interaction data are, in my view, much less convincing than much of the other data in the manuscript. Firstly, the rationale for looking at this potential interaction (bottom of pg 13) is very weak, and a more logical/stronger case should be presented (e.g. data on cell surface TLR4 – flow cytometry – could be presented first, which might then lead on to TLR4-NOC4L interaction studies). Secondly, the data in Figure 5 that forms the basis for the conclusions are not compelling and are lacking important controls:

(a) in Fig 5a, the authors should show the FLAG blot of the actual IP (not just the input) to confirm that NOC4L has actually been pulled down in the GFP control sample, and they should also show the full gel for the IP. GFP alone will run much lower than TLR4-EGFP (see input), and the authors have cropped the blot – so we don't know if any GFP has been pulled down in the control sample (i.e. GFP alone). This is a major concern. Moreover, another EGFP fusion protein (e.g. similar size to TLR4-EGFP) should be used as an additional control in these experiments to show that it is not pulled down with FLAG-NOC4L (C-terminus of NOC4L-FLAG could also be used as an additional IP control, since the data in Fig 5d suggest that this domain does not interact with TLR4). Finally, with respect to Fig 5a, the argument that 24h LPS treatment downregulates the interaction seems strange – if this is the case, I would have thought this would be seen at a much more acute time point. More concerning is that HEK293T cells should not be LPS responsive unless both TLR4 and MD2 are transfected in to cells – so it is difficult to make sense of these data. Could this apparent decrease simply reflect sample to sample variation and/or background issues?

(b) In relation to the above concerns, Fig 5b shows an endogenous IP, but there is a clear signal with the IgG control IP, and the inputs do not show the IgG control lane. Since the authors have Noc4l myeloid k/o mice available, they could perform IPs for the endogenous interaction on wild type vs knock-out macrophages. This would provide a much cleaner system for these studies i.e. an anti-NOC4L antibody should pull down TLR4 in wild type but not Noc4l knock-out cells.

(c) The PLA data in Fig 5c lacks controls (e.g. an antibody of the same isotype as the anti-NOC4L

antibody, but targeting another cytoplasmic protein), and again, PLA on wild type versus Noc4l k/o macrophages would provide reassurance that the PLA signal represents a bona fide NOC4L/TLR4 interaction. The PLA images should also include scale bars, as well as graphical quantification (not a single cell) i.e. quantification of PLA puncta/cell vs controls across multiple experiments.

(d) As far as I can tell, the methodology for Fig 5d has not been included in the manuscript, so it is very difficult to make any informed comments on these data (this information should be provided in the methods).

3. If the argument is that NOC4L constrains the TRIF pathway, then this could be tested by examining inflammatory responses in macrophages from wild type, Noc4l k/o mice, Trif k/o mice and Noc4l/Trif double k/o mice (the authors have generated these mice - Extended Data Fig 2h). If the proposed model is correct, then amplified inflammatory responses that are apparent in Noc4l k/o macrophages should be abrogated by TRIF deletion. If this is not the case, then the authors would need to revise their model. In fact, the authors present data in Extended Data Fig 2h that are not supportive of their model i.e. the effect of Noc4l deletion on blood glucose should be abrogated by TRIF deletion – but it is not. I am confused by the authors attempts to interpret these data in the results, as they do not seem consistent with the presented data. Also, there is no figure legend presented for the data in Extended Data Fig 2h.

4. Relating to the above, the authors show that macrophages from NOC4L over-expressing mice have elevated cell-surface TLR4 (Fig 6n). So, one would expect that LPS-induced IFN-beta and CCL5 expression, as well as IRF3 activation, should be reduced in macrophages from these mice versus controls. Have the authors assessed this? Perhaps this phenotype will only be revealed at sub-maximal LPS concentrations. Such experiments would provide stronger support for the proposed mechanism of NOC4L action.

5. Most of the macrophage studies are on BMM or peritoneal macrophages. Since the focus of the study is on obesity and metabolic disease, the authors should isolate adipose tissue macrophages from their Noc4l k/o mice and Noc4l fl/fl mice and confirm some of the key findings (e.g. surface expression of TLR4, TLR4 internalization, TRIF pathway activation, inflammatory genes etc). Such experiments are required to validate the authors claims that NOC4L does indeed limit adipose tissue macrophage inflammation.

6. The authors attempt to connect the amplified TRIF signalling response to the in vivo phenotype, but they don't provide any data to support this connection. They should tone down their claims about the role of NOC4L in constraining TRIF signalling as the mechanism being responsible for the metabolic/inflammatory features that they observe in vivo (particularly in the Discussion) OR provide in vivo data to support their claims e.g. abrogation of the metabolic/inflammatory phenotypes in Noc4l k/o mice after crossing on to Trif k/o mice – see point #3 above. Moreover, there is clearly a hyper-inflammatory phenotype in Noc4l-deficient macrophages in the absence of LPS (e.g. IL-6, iNOS, MCP1: Fig 4f-g), so it would seem that control of TRIF signalling is unlikely to be the sole mechanism by which NOC4L acts – unless the authors are proposing that TRIF is constitutively active in the absence of NOC4L (in which case, the basal inflammatory phenotype should be lost in the absence of TRIF).

7. Issues with Figure 1:

(a) Fig 1b – Noc4l expression in adipocytes vs SVF should be performed by qRT-PCR, rather than the non-quantitative gel that is shown.

(b) Fig 1c – can the authors provide images with an increased magnification? In particular, better quality images (more focused) of F4/80 (mouse) and Noc4l (human) is highly desirable.

Quantification of the co-localization data would also be helpful.

(c) Fig 1f – the blot showing that LPS decreases Noc4l expression over a time course is not convincing. The band on the blot is barely visible. Can the authors provide a blot with a stronger exposure, so that the downregulation is clearly apparent?

(d) Fig 1i – were THP-1 cells PMA-differentiated? If so, this should be stated in describing the data, and details provided in the methods (THP-1 cells are generally not LPS responsive unless they have been differentiated with PMA).

8. Issues with Figure 2:

- (a) Fig 2b – it would be preferable to re-run these samples on the one gel for a direct comparison. Also, a graph quantifying a larger number of samples (e.g. n=5 mice for each group) with appropriate statistical analysis is highly desirable (rather than just a representative blot with only n=2 per group).
- (b) Fig 2e – the authors should confirm overexpression of Noc4l in adipose tissue (and other tissues, e.g. liver) with Lv-Noc4l delivery. And were there any differences in the response to insulin (i.e. insulin tolerance tests)?
- (c) Fig 2i – were there any difference in oil-red-O staining or similar in the liver?

9. Issues with Figure 4:

- (a) Fig 4f – the definition of an “M1” macrophage is IFN-gamma+LPS-treated – the authors should revise their nomenclature in describing these data (e.g. M1-like) or perform the experiments using IFN-gamma+LPS.
- (b) Fig 4g – it would be appropriate to examine basal and LPS-induced levels of other inflammatory mediators at the protein level (not just IL-6).

10. Issues with Figure 6:

- (a) Fig 6c – the representative blot of total IRF3 looks like there’s a decrease in total protein of IRF3 in response to LPS in Noc4l k/o cells at 45 and 60 min post-stimulation – is this real? Is IRF3 being degraded in response to LPS treatment? This could affect interpretation of the data in terms of increased phospho-IRF3, as assessed by quantification of pIRF3/total IRF3.
- (b) Fig 6d – the authors state that IFN-beta expression is significantly increased in Noc4l k/o macrophages, but this is not the case when looking at the graph. Is there an asterisk missing?
- (c) Fig 6e – this data set requires expression of a control protein to show that the effect of NOC4L on TRIF responses is specific (i.e. not apparent with a control protein) – to exclude issues such as promoter competition. Perhaps the C-terminus of NOC4L that doesn’t interact with TLR4 could be used here? The figure also requires a no TRIF control to be included, so that one can see the baseline level of NF-kappaB activation.
- (d) Fig 6f - the above issue (i.e. expression of a control protein) also applies to this figure panel.
- (e) Fig 6h-I require scale bars.
- (f) Fig 6j – ideally, co-localization studies with Rabs should be performed in BMMs, since the localization of NOC4L varies depending on the cell type (nuclear vs cytoplasmic). Could this variation in localization reflect primary cells (e.g. BMMs) versus cell lines (mainly used for showing nuclear localization)? Have the authors investigated NOC4L localization in RAW264.7 (macrophage cell line) and/or non-macrophage primary cells?

11. Issue with Fig 7:

- (a) Fig 7a – what are the expression levels of TRIF and CD68 in these T2D patients? Were these T2D patients obese or lean? Also, the IL-1beta data appear to be particularly striking in this data set. It is somewhat surprising that IL-1beta (and other mediators that are typically increased/decreased in obesity/metabolic disease and linked to disease processes e.g. IL-1alpha, IL-10 etc) were not examined in the in vitro and in vivo mouse studies using NOC4L loss-of-function and gain-of-function.

12. Issues with Extended data Fig 2:

- (a) Extended data Fig 2g - requires unstimulated controls so that one can see the baseline levels of inflammatory gene expression.
- (b) Extended data Fig 2h – see concerns above re: interpretation and lack of figure legend.

13. Issues with Extended data Fig 3:

- (a) Extended data Fig 3b-c – why switch between cell types for panel b vs c? It would be much more preferable to perform staining for lysosomes, ER and EEA all on RAW264.7 and/or BMM (not switch between HeLa cells and macrophages).

- 14. Figure legends – all figures should indicate how many experiments individual data sets are combined from (for each panel). This is not apparent for some of the panels in some of the figures (especially in vitro data). Please also thoroughly check all legends to ensure that there are no mistakes in referring to panels, and in providing necessary information to understand each panel.

15. Discussion – The authors should discuss the differences in GTT vs ITT data for in vivo studies. See also point #6 relating to modifying the Discussion to avoid over-interpretation of in vivo mechanisms without supporting data.

16. The authors need to cite and discuss the recent paper by Zhu et al (Cell Reports, 2019; PMID: 31018134). Since Tregs are also present in adipose tissue and since there is obvious overlap with the current study (but in a different cell type), this published study has implications for understanding mechanisms by which NOC4L regulates inflammatory responses.

17. Some of the details in Methods are either very brief (e.g. monoclonal/polyclonal Ab generation, nuclear/cytoplasmic extract generation) or are missing altogether (e.g. 35S labelling/GST pull downs). The Methods section should be carefully reviewed and updated.

MINOR:

1. Reference(s) are required for the description of RNAseq data for various tissues (lines 109-111).
2. Fig 1d-e – include legend on x-axes (time for d, micromolar for e).
3. Fig 2a & b – which fat pad from mice were used to assess Noc4l?
4. Fig 2c –What was the sex, age and background of these patients? This information should be included in the methods. Also, a BMI>30 is generally considered obese, not BMI>28. The authors should justify their BMI criteria.
5. Fig 2d – it is unclear how many mice were used in these comparisons and whether the quantification of Noc4l/GAPDH was an average of all WAT tissues or specific ones (e.g. eWAT, iWAT and pWAT). It would be interestingly to know if there was differential expression of Noc4l in different fat depots.
6. Fig 2e – the authors should show the AUC quantification graph (not just state a 16% decrease in the text).
7. Fig 3h – insert legend with symbols for WT and OE.
8. Extended Fig 2d – the authors should provide the list of 200 genes and/or upload the data in GEO database or similar.
9. Fig 4h – is there statistical significance between 2 groups in control for Arg expression (as per text)? If so, the authors should include statistical significance on the graph.
10. Please indicate protein sizes on Western blots.

Reviewer #2 (Remarks to the Author):

General Comments:

This manuscript by Qin et al investigates the contribution of NOC4L to adipose tissue macrophage (ATM) inflammation and insulin resistance during obesity. The authors demonstrate that NOC4L is decreased in obese humans and in mice and that NOC4L mRNA and protein levels in macrophages are decreased in response to obesity and inflammatory stimuli respectively. They show that lentiviral tail vein injection of NOC4L decreases weight gain during high fat diet challenge and that knocking out NOC4L in macrophages improves glucose tolerance and reduces inflammation during diet induced obesity. They go on to show that NOC4L interacts with toll-like receptor 4 (TLR4) to inhibit endocytosis and block the TRIF pathway to attenuate inflammatory cytokine production. Based on these findings, the authors conclude that NOC4L is a potential biomarker or therapeutic target for insulin resistance. Although the effects of NOC4L on the diet-induced phenotype are clear, and NOC4L is a novel player in regulating inflammation, there are numerous major concerns in the model systems and experimental rigor that lower the impact and interpretation of their results.

Major comments:

- 1) A central claim of this paper is that NOC4L knockout increases inflammation to worsen insulin resistance in mice during diet induced obesity. However, the perturbations used to alter NOC4L (lentiviruses or -/-) also significantly affect body weight. What is the basis for this increase in adiposity? Metabolic cage experiments should be performed. Moreover, the relationship between increased body weight and insulin resistance is well documented in both mice and humans. Thus,

how can the authors rule out the possibility that it's the change in body weight and not changes in ATM inflammation that are causing change to insulin resistance? Weight matching experiments, where insulin resistance is measured in wt and Noc4l^{-/-} mice of similar weight, must be included to rule out this possibility. This is particularly important because macrophage inflammatory pathways in the brain are known to regulate feeding behavior.

2) Although ATM inflammation is the main focus of the paper, there are no measurements that directly demonstrate that ATM inflammation is altered in Noc4l^{-/-} mice. Inflammatory cytokine measurements are made on whole adipose tissue or plasma, neither of which directly demonstrates that ATMs are actually inflamed. Moreover, the results of the whole adipose tissue qRT-PCR (Fig. 4b) shows increases in F4/80 and CD68, raising the possibility that an increase in ATM number maybe more important than an increase in ATM inflammatory state per se (which is predicted by their mechanistic work). The authors should isolate ATMs from mice and quantify inflammatory cytokine expression within them under the various perturbations.

3) The authors use two in vitro inflammatory perturbations: lipopolysaccharide (LPS) and the fatty acid palmitate to study macrophage inflammation. LPS is a well-established method to classically activate (M1) macrophages to mimic their phenotype during infection, while palmitate is a more physiologically relevant stimulus to study macrophage inflammation during diet-induced obesity. The authors treat these two stimulations as equal for much of the manuscript, and then focus their more detailed mechanistic work on LPS signaling. Recent studies suggest that palmitate induces a metabolically activated (MMe) phenotype that is dramatically different from M1 macrophages both in terms of protein expression and the mechanisms supporting inflammation (Kratz et al, Cell Metab, 2014; Coats et al, Cell Rep. 2017). Indeed, recent studies showed that although Tlr4^{-/-} attenuates inflammation in palmitate-treated macrophages, TLR4 does not directly bind palmitate and influences its signaling indirectly (Lancaster et al., Cell Metab., 2018). These papers bring to question the strategy of using LPS as a model system to uncover mechanisms that are at play during diet-induced obesity. Based on this rationale, it would be more appropriate for the authors to determine mechanisms of how Noc4l^{-/-} impacts inflammation in palmitate-challenged macrophages.

4) The authors present nice data to show the mechanism underlying NOC4L-TLR4 interactions, as it pertains to inflammatory signaling. However, these data were acquired with cells lines, most of which are not of the monocyte/macrophages lineage, and with a stimulus (LPS) that is not physiologically relevant to studying diet-induced obesity insulin resistance. Moreover, there are no confirmatory data in ATMs in vivo. It is therefore recommended for authors to confirm their mechanism in palmitate-challenged macrophages in vitro, and to provide evidence of similar effects in ATMs isolated from lean and obese wt and Noc4L^{-/-} mice.

5) One of the major claims of this manuscript is that the interaction between TLR4 and NOC4L is required to inhibit internalization of TLR4 and downstream inflammatory signaling. The authors use pulldowns and a proximity ligation assay to demonstrate that TLR4 and NOC4L interact via the N-terminal domain of NOC4L and the TIR domain of TLR4. However, the necessity of this interaction remains unexplored by the authors, leaving questions as to its relevance to the biology. Does inhibiting the interaction between these regions mimic the phenotype of the knockout mice/cell lines?

Other comments:

1) Lentiviral overexpression of proteins is a well characterized way to perform in vivo modifications, in the same way the LysMCre model has been used to study the relevance of proteins in myeloid cells. However, in both cases the specificity of the modifications may vary in different macrophages types. To provide more rigor to their perturbations, the authors need to confirm their overexpression and knockouts in isolated ATMs.

2) Many of the figures and figure legends show a lack of attention to detail on the part of the authors. There are several instances where the cell lines being studied, the n for the experiment, the antibodies used for staining for IHC are not clear. It is recommended for the authors to revise all figure legends to improve clarity.

Reviewer #3 (Remarks to the Author):

In this manuscript 'Macrophage deletion of Noc4l triggers endosomal IRF3/IFN β signal and leads to insulin resistance', the authors determine the role of Noc4l in regulation macrophage responses in adipose tissue during obesity. They identify regulation of TLR4 by Noc4l followed by control of INF β activity as a crucial pathway mediating adipose tissue inflammation and ultimately regulating insulin sensitivity during obesity.

Using different approaches ranging from in vitro to in vivo animal models and human adipose tissue, the authors conclude that Noc4l protects against adipose tissue inflammation and improves insulin sensitivity.

The manuscript reads well and contains an impressive amount of experiments. However, this also represents somewhat of a weakness, as the authors have attempted to cover many different aspects that sometimes leads to incomplete results and preliminary conclusions.

Please find below my comments and suggestions that may help to improve the quality of the current manuscript.

1. The authors show that noc4l controls TLR4 signalling by regulating its internalization. This also implies that TLR4-ligands are the main drivers of adipose tissue inflammation. In addition to fatty acids that are being debated as true TLR4 ligands, what other potential ligands may exist in adipose tissue? Also, LPS and PA treatment leads to downregulation of noc4l expression in macrophages. Although downregulation of noc4l is also observed in vivo in obese adipose tissue, do the authors have any data available related to dynamics of noc4l expression? In other words, how is the expression regulated during DIO and are effects on gene expression seen early or late after the start of the HFD-intervention?

Also, obesity promotes the influx of macrophages into the adipose tissue. However, noc4l expression is reduced. Would this mean that expression is lowered within macrophages residing in adipose tissue or that the infiltrating macrophages are of a different phenotype with low expression levels of noc4l?

Also, the authors should provide more information about the study participants that were included. Age, BMI, medication, Hba1c, other relevant parameters?

2. The results section is very long and contains a long list of different experiments that sometimes lack coherence. I would also suggest to carefully reconsider the order of presenting the results. For example, I would first present results on regulation of noc4l in adipose tissue during obesity, before presenting data related to the regulation of noc4l by LPS and PA. Also, data on the interaction between noc4l and TLR4 presented in figure 5 is of interest from a biochemical point of view, yet is difficult to translate to in vivo relevance. Also, how would these results fit with the qPCR data on inflammatory gene expression in untreated ko BMDMs?

3. Based on the data presented in the second paragraph ending on page 9, the authors conclude that noc4l could directly improve IR in vivo. I think the authors should rephrase this text. Based on the findings, one could conclude that noc4l is downregulated in adipose tissue during obesity and that overexpression using a lentivirus, that primarily leads to an increase in hepatic expression of noc4l, lowers glucose intolerance. However, overexpression also reduces BW and fat mass, both known to impact on glucose tolerance. Hence, noc4l may not directly improve IR, but more indirectly via affecting the BW of the animals. Btw, any information on food intake or activity levels that may explain the difference in weight gain? Same holds true for the higher BW in the knockout animals. The authors need to provide more data related to a potential cause for the differences in BW. Again, by impacting on BW/fat mass all effects of noc4l on IR and adipose tissue inflammation could be largely independent from its effects on TLR4 signalling.

4. Page 9, line 183. What type of macrophages have been isolated and tested? ATMs?

5. The authors claim that INF β signalling is the key pathway impacting on insulin sensitivity, as stated in the title. However, no evidence in the context of the adipose tissue is presented in the manuscript. Although data presented in figure 6 show a role of noc4l in controlling INF β expression levels in BMDMs, what about INF β levels in vivo in adipose tissue in WT vs. myeloid-specific noc4lko animals? In addition, how do the authors conclude that INF β contributes to the development of insulin resistance in vivo?

In general, the authors should tone down some of the conclusions made in the manuscript and also

adapt the title. In addition, more work is needed to decipher the mechanism of action leading to differences in BW due to noc4l.

6. The authors show and discuss the relevance of noc4l in controlling m1 vs. m2 polarization. Although this is of importance, the relevance of these phenotypes in adipose tissue in vivo may be somewhat limited. What about a role of noc4l in controlling metabolic activation of macrophages using triggers that are more relevant for adipose tissue?

Responds to the reviewer's comments:

Reviewer #1:

1. Very little information is provided in the Introduction about how NOC4L regulates 40S ribosome biosynthesis/nuclear export (e.g. pathway, mechanisms, interacting proteins, key domains of NOC4L and how these function). Providing such information in the introduction would provide more context for the studies that were undertaken. Moreover, assessing ribosome/polysome profiles in wild type versus *Noc4l* knock-out macrophages (e.g. by sucrose gradient) would seem like an important control for the studies on the myeloid-deleted and over-expressing mice.

Response: We thank for the Reviewer 1's constructive suggestion. According to his/her advice, we have reviewed the literature in detail, and explained the key domains of NOC4L, the interacted proteins and how they regulate 40S ribosome biosynthesis/nuclear export in the introduction. The details of this section in introduction are as follows: Nucleolar complex associated 4 homolog (NOC4L), also known as NOC4, NET49, UTP19, is a homologue of yeast *Noc4p*. *Noc4p* and *Nop14p* form a complex, which was mainly involved in the assembly and transport of ribosome 40S subunit in yeast^{1,2}. The normal tissue RNA-seq data mined from the public database show that NOC4L is highly expressed in testis, fat and immune organs of human³. We also demonstrated that deletion of *Noc4l* lead to embryonic lethality in mice⁴. A recent study demonstrates the critical role of NOC4L in activation of regulatory and conventional T cells. NOC4L-deficient T cells have a smaller 40S peak, which is involved in selectively controlling protein translation in Tregs and Tconvs⁵.

Moreover, we also assessed the ribosome/polysome profiles in macrophages of *Noc4l*^{fl/fl} vs. *Noc4l*^{LKO}. To further confirm whether *Noc4l* was involved in regulating the peak of 40S, 60S and 80S, we carried out five independent experiments (Please see the Figure results from five times in below, Fig. 1). Based on these results from five times of experiments, we concluded that deletion of *Noc4l* in macrophages had no influence on the peaks of 40S, 60s and 80s (Extended Data Fig. 2h of the revised manuscript). Meanwhile, we also addressed this issue in the discussion section of the revised manuscript in line 340-342.

Fig. 1

2. The TLR4/NOC4L interaction data are, in my view, much less convincing than much of the other data in the manuscript.

Firstly, the rationale for looking at this potential interaction (bottom of pg 13) is very weak, and a more logical/stronger case should be presented (e.g. data on cell surface TLR4 – flow cytometry – could be presented first, which might then lead on to TLR4-NOC4L interaction studies).

Response: We thank for the constructive comments and suggestions from Reviewer 1. According to the reviewer's suggestion, we reformed the manuscript and made it more logical and following the thinking flow. The logic of the revised manuscript is:

- 1) The expression of NOC4L was decreased in both diet-induced obese (DIO) mice and genetically diabetic mice, as well as obese humans.
- 2) Noc4l is highly expressed in testis, fat and immune organs of human and mice.
- 3) More precisely, Noc4l was mainly expressed in both mouse and human macrophages of adipose tissue and immune organs like spleen.
- 4) The expression of Noc4l in macrophages was regulated by Lipopolysaccharide (LPS) and Palmitic Acid (PA)
- 5) Based on the high expression of Noc4l in macrophages, we prepared myeloid specific deletion of Noc4l mice by bred Noc4l^{fl/fl} mice with the Lys M cre mice.
- 6) Insulin resistance was observed in Noc4l^{LKO} (myeloid specific deletion of Noc4l) mice.
- 7) Macrophage-specific deletion of Noc4l aggravated HFD-induced inflammation in mice.
- 8) Noc4l inhibited M1-like differentiation and activation of macrophage ex vivo.
- 9) Noc4l inhibited MyD88-independent TLR4/TRIF pathway in macrophages.
- 10) Noc4l localized on the early endosome-associated subcellular compartments and inhibited TLR4 endocytosis.
- 11) Noc4l directly interacted with the TIR domain of TLR4.
- 12) Molecular changes in adipose tissues of T2D patients were similar to that of Noc4l^{LKO} mice.

Secondly, the data in Figure 5 that forms the basis for the conclusions are not compelling and are lacking important controls:

(1) in Fig 5a, the authors should show the FLAG blot of the actual IP (not just the input) to confirm that NOC4L has actually been pulled down in the GFP control sample, and they should also show the full gel for the IP. GFP alone will run much lower than TLR4-EGFP (see input), and the authors have cropped the blot – so we don't know if any GFP has been pulled down in the control sample (i.e. GFP alone). This is a major concern.

Response: We appreciated the Reviewer 1's critical comments. Fig. 5a in previous manuscript did have some shortcomings as the reviewer points out. We have shown the blots of Flag and GFP both in the actual IP and Input, and the molecular markers were shown on the figure. GFP was found in the input blot but not in IP blot (lane 1). In the revised manuscript, we have answered the main questions (Fig. 6a in the revised manuscript).

(2) Moreover, another EGFP fusion protein (e.g. similar size to TLR4-EGFP) should be used as an additional control in these experiments to show that it is not pulled down with FLAG-NOC4L (C-terminus of NOC4L-FLAG could also be used as an additional IP control, since the data in Fig 5d suggest that this domain does not interact with TLR4).

Response: The Reviewer 1 suggested that another EGFP fusion protein (e.g. similar size to TLR4-EGFP) should be used as an additional control. It is quite reasonable, whereas to find a protein with a similar size to TLR4-EGFP is practical difficult. Firstly, TLR4 is a membrane protein with a large molecular weight. Secondly, there are few reports about NOC4L. Even if a similar size protein is found, it is not sure whether it can be used as a true negative control.

In addition, for negative control setting, we followed the common sense of the academic rules, please see the references ^{6,7}.

(3) Finally, with respect to Fig 5a, the argument that 24h LPS treatment downregulates the interaction seems strange – if this is the case, I would have thought this would be seen at a much more acute time point. More concerning is that HEK293T cells should not be LPS responsive unless both TLR4 and MD2 are transfected in to cells – so it is difficult to make sense of these data. Could this apparent decrease simply reflect sample to sample variation

and/or background issues?

Response: The Reviewer 1 pointed out that HEK293T should not be LPS responsive. As HEK293T cells might be contaminated by HeLa cells^{8,9}, in order to draw the solid conclusion, we carried out the additional experiments by using HeLa and RAW 264.7 cells and deleted the results from HEK293T cells. We found that Noc4l was interacted with TLR4 in HeLa and RAW 264.7 cells. The results were added in the Fig. 6b and Extended Data Fig. 3g of the revised manuscript.

(b) In relation to the above concerns, Fig 5b shows an endogenous IP, but there is a clear signal with the IgG control IP, and the inputs do not show the IgG control lane. Since the authors have Noc4l myeloid k/o mice available, they could perform IPs for the endogenous interaction on wild type vs knock-out macrophages. This would provide a much cleaner system for these studies i.e. an anti-NOC4L antibody should pull down TLR4 in wild type but not Noc4l knock-out cells.

Response: We thank the Reviewer 1 for these important suggestions. To confirm the interaction between Noc4l and TLR4, we performed endogenous IP with Noc4l antibody in HeLa and RAW 264.7 cells (Extended Data Fig. 3g and Fig. 6b in revised manuscript), TLR4 was found in the immunoprecipitate complex. To further confirm these findings, primary BMDM cells isolated from wild-type and Noc4l^{-/-} mice were subjected to immunoprecipitate assay with Noc4l antibody (in revised manuscript), and TLR4 was observed in wild-type group but not in Noc4l^{-/-} group. Based on these results, we confirmed that Noc4l interacted with TLR4. The results were presented in the revised Extended Data Fig. 3g and Figs. 6b-6c.

(c) The PLA data in Fig 5c lacks controls (e.g. an antibody of the same isotype as the anti-NOC4L antibody, but targeting another cytoplasmic protein), and again, PLA on wild type versus Noc4l k/o macrophages would provide reassurance that the PLA signal represents a bona fide NOC4L/TLR4 interaction. The PLA images should also include scale bars, as well as graphical quantification (not a single cell) i.e. quantification of PLA puncta/cell vs controls across multiple experiments.

Response: As PLA is an antibody-based method in which either a single or two proteins are immunolabeled first with two primary antibodies and then with different specific secondary antibodies conjugated to complementary oligonucleotides. Generally, if the primary antibodies can specifically recognize the corresponding antigen, the results are considered to be true and effective, no additional control is needed^{10,11}.

In order to make the solid conclusion, we carried out the additional experiments, which included not only wild type macrophage, but also Noc4l^{LKO} and Noc4l^{LOE} macrophage groups as the controls. The results showed that the interaction of Noc4l and TLR4 was disappeared in Noc4l^{LKO} and enhanced in Noc4l^{LOE} macrophages; we also made the quantification of the results. These results were added in revised Figs. 6d-6e.

Moreover, to confirm the specificity of TLR4 antibody, TLR4 mutant macrophages were used as a negative control group. The result showed that no positive signal was observed in both Noc4l/IgG and Noc4l/TLR4 groups (in revised Fig. 6f). With the results from Noc4l^{LKO} and TLR4 mutant macrophages controls, we confirmed that PLA signal does represent a *bona fide* NOC4L/TLR4 interaction.

(d) As far as I can tell, the methodology for Fig 5d has not been included in the manuscript, so it is very difficult to make any informed comments on these data (this information should be provided in the methods).

Response: We must confess that this work took about 8 years and the authors have graduated from school from time to time. We are sorry for losing the raw data of Fig 5d in previous manuscript. Due to the limited revised time, to re-perform the isotope labeling GST pull-down experiment seems unrealistic upon this moment. In order to confirm our previous results, alternative GST pull-down was carried out. Briefly, TLR4 deletions labeled with GST were purified in bacterial expression system and Flag-NOC4L-His was purified using eukaryotic expression system. The method was added to the MATERIALS AND METHODS in revised manuscript. The new GST-pulldown assay demonstrated that full length of NOC4L was directly interacted with TIR domain of TLR4 (in

revised Fig. 6g).

As previous data showed that N-terminus of NOC4L, not the C-terminus, was directly interacted with the TIR domain of TLR4, but we failed to get the consistent results from the C-terminal of NOC4L during this short revision period. In order to make the decent conclusion, we tuned down our conclusion that NOC4L was directly interacted with TIR of TLR4, rather than the N-terminus of NOC4L, not the C-terminus, was directly interacted with the TIR domain of TLR4 in revised manuscript.

3. If the argument is that NOC4L constrains the TRIF pathway, then this could be tested by examining inflammatory responses in macrophages from wild type, *Noc4l* k/o mice, *Trif* k/o mice and *Noc4l/Trif* double k/o mice (the authors have generated these mice - Extended Data Fig 2h). If the proposed model is correct, then amplified inflammatory responses that are apparent in *Noc4l* k/o macrophages should be abrogated by TRIF deletion. If this is not the case, then the authors would need to revise their model.

In fact, the authors present data in Extended Data Fig 2h that are not supportive of their model i.e. the effect of *Noc4l* deletion on blood glucose should be abrogated by TRIF deletion – but it is not. I am confused by the authors attempts to interpret these data in the results, as they do not seem consistent with the presented data. Also, there is no figure legend presented for the data in Extended Data Fig 2h.

Response: We thank the Reviewer 1's comment, we have carried out additional experiments to check the mRNA expressions of *TNFA*, *IL-6* and *IFN β* in BMDMs from wild type, *Noc4l* k/o, TRIF k/o, and *Noc4l/TRIF* double k/o mice stimulated with LPS. The results showed that the inflammatory reactions of *Noc4l* k/o BMDMs were significantly abrogated by TRIF deletion (in revised Figs. 5g-i). The in vitro experimental results showed that NOC4L is the upstream of TRIF and constrains the TRIF pathway.

As Reviewer 1 pointed out that Extended Data Fig 2h is not supportive the results of our model, to confirm the regulatory effect of NOC4L on TRIF signaling, up to 8 mice of each group for GTT was used in the revised manuscript. The trend of the AUC area were consistent with the previous Extended Data Fig 2h. As TRIF KO and *Noc4l*^{LKO} mice have higher GTT levels than wild type mice, in order to better quantify the role of *Noc4l* in GTT, we performed statistical statistics at each time point. We observed the elevated blood glucose by *Noc4l* deletion was significantly abrogated by TRIF deletion in mice, thus concluded that NOC4L could constrain the TRIF pathway in vivo, and we replaced the previous Figure with the revised Extended Data Fig. 2h.

4. Relating to the above, the authors show that macrophages from NOC4L over-expressing mice have elevated cell-surface TLR4 (Fig 6n). So, one would expect that LPS-induced IFN-beta and CCL5 expression, as well as IRF3 activation, should be reduced in macrophages from these mice versus controls. Have the authors assessed this? Perhaps this phenotype will only be revealed at sub-maximal LPS concentrations. Such experiments would provide stronger support for the proposed mechanism of NOC4L action.

Response: We thank to the Reviewer 1's comment, the additional experiments of BMDMs from WT, KO and OE mice were isolated and stimulated with different doses of LPS. We found that the mRNA levels of *IFN- β* and *CCL5* BMDMs from OE mice had no difference with WT BMDMs but were lowered than *Noc4l*^{LKO} BMDMs (Please see the Fig. 2). However, we don't have the clear answer for the phenomenon of inflammatory factors of the similar levels between WT and OE BMDMS groups upon this moment.

Fig. 2

5. Most of the macrophage studies are on BMM or peritoneal macrophages. Since the focus of the study is on obesity and metabolic disease, the authors should isolate adipose tissue macrophages from their *Noc4l* k/o mice and *Noc4l* fl/fl mice and confirm some of the key findings (e.g. surface expression of TLR4, TLR4 internalization, TRIF pathway activation, inflammatory genes etc). Such experiments are required to validate the author's claims that NOC4L does indeed limit adipose tissue macrophage inflammation.

Response: We thank the reviewer's constructive comments. We are sorry for ambiguous description in the previous manuscript, which may lead the misunderstanding of the manuscript. As *Noc4l* was observed highly expressed in the reproductive organs, immune organs and adipose tissue, spleen and is mainly localized on the macrophages of these tissues, thus, to study the particular molecular action of *Noc4l*, we constructed the knockout mouse by using Lysozyme M cyclization recombinase (Lys M cre) mice. Lys M cre mice is the most general tool to study the specific functions of macrophages^{12,13}. As Lys M cre is not ATMs specific but myeloid cell specific deletion, thus, deletion of *Noc4l* in macrophages results in a systemic malfunction of macrophages which may lead to the systemic inflammatory responses, rather than ATMs only. Some study had demonstrated that myeloid cells play an important role in regulating insulin response¹². Under this condition, the BMDM and peritoneal macrophages are more relevant cell types in this study. In order to get the better understanding for the authors, we clarified Lys M cre mice, and stated this both in the MATERIALS AND METHODS and Discussion sessions in the revised manuscript.

To further better understanding the precise role of *Noc4l* in ATM, our next work we will focus on ATM to unearth the mechanism by which *Noc4l* regulates inflammation and insulin response in the future. And we have toned down the effect of *Noc4l* in ATMs in the revised manuscript.

6. The authors attempt to connect the amplified TRIF signalling response to the in vivo phenotype, but they don't provide any data to support this connection. They should tone down their claims about the role of NOC4L in constraining TRIF signalling as the mechanism being responsible for the metabolic/inflammatory features that they observe in vivo (particularly in the Discussion) OR provide in vivo data to support their claims e.g. abrogation of the metabolic/inflammatory phenotypes in *Noc4l* k/o mice after crossing on to *Trif* k/o mice – see point #3 above. Moreover, there is clearly a hyper-inflammatory phenotype in *Noc4l*-deficient macrophages in the absence of LPS (e.g. IL-6, iNOS, MCP1: Fig 4f-g), so it would seem that control of TRIF signalling is unlikely to be the sole mechanism by which NOC4L acts – unless the authors are proposing that TRIF is constitutively active in the absence of NOC4L (in which case, the basal inflammatory phenotype should be lost in the absence of TRIF).

Response: Please see the answers in point # 3 of Reviewer 1.

Response to: A hyper-inflammatory phenotype in *Noc4l*-deficient macrophages in the absence of LPS.

We thank the Reviewer 1's comments. Indeed, the t-test methods of analysis data in previous manuscript are imprecise. We re-did statistics on the data by using two-way ANOVA, and the results showed that inflammatory phenotype of WT and KO BMDMs has no difference in absence of LPS or PA. We have rephrased the description of data in revised manuscript, and the methods of analysis

data has been added in the MATERIALS AND METHODS in revised manuscript.

7. Issues with Figure 1:

(a) Fig 1b – *Noc4l* expression in adipocytes vs SVF should be performed by qRT-PCR, rather than the non-quantitative gel that is shown.

As required by the reviewer, we have carried out the qRT-PCR experiment of *Noc4l* expression in adipocytes vs. SVF (Extended Data Fig. 1b in revised manuscript) and validated that *Noc4l* was predominantly expressed in SVF of adipose tissue.

(b) Fig 1c – can the authors provide images with an increased magnification? In particular, better quality images (more focused) of F4/80 (mouse) and *Noc4l* (human) is highly desirable. Quantification of the co-localization data would also be helpful.

According to the reviewer's suggestion, we have provided a figure with high magnification.

(c) Fig 1f – the blot showing that LPS decreases *Noc4l* expression over a time course is not convincing. The band on the blot is barely visible. Can the authors provide a blot with a stronger exposure, so that the downregulation is clearly apparent?

In response to the reviewer's request, we performed the new Western blot and replaced Fig 1f in the previous manuscript with a stronger exposure of *Noc4l* expression, and the result showed that LPS can decrease *Noc4l* expression over a time course (Fig. 2d in revised manuscript).

(d) Fig 1i – were THP-1 cells PMA-differentiated? If so, this should be stated in describing the data, and details provided in the methods (THP-1 cells are generally not LPS responsive unless they have been differentiated with PMA).

We have re-written this part in the figure legend of Fig. 2g and methods in line 428-429 in the revised manuscript according to the reviewer's suggestion.

8. Issues with Figure 2:

(a) Fig 2b – it would be preferable to re-run these samples on the one gel for a direct comparison. Also, a graph quantifying a larger number of samples (e.g. n=5 mice for each group) with appropriate statistical analysis is highly desirable (rather than just a representative blot with only n=2 per group).

We have changed Fig 2b with a picture that the proteins of *Noc4l* in adipose from CD and HFD diet were loaded on one gel, and the expression of *Noc4l* was quantified by the software Image J (Fig. 1c in the revised manuscript). The result showed that expression of *Noc4l* was decreased in HFD mice compared with CD mice.

(b) Fig 2e – the authors should confirm overexpression of *Noc4l* in adipose tissue (and other tissues, e.g. liver) with Lv-*Noc4l* delivery. And were there any differences in the response to insulin (i.e. insulin tolerance tests)?

According to the referee's request, we examined the overexpression efficiency of *Noc4l* in eWAT and liver tissues and confirmed that *Noc4l* was overexpressed in eWAT and liver tissues with Lv-*Noc4l* delivery (Figs. 1f-1g in the revised manuscript).

We also conducted the experiment of insulin tolerance test (ITT) and the result showed that delivery of Lv-*Noc4l* can ameliorate the insulin resistance (Fig. 1i in the revised manuscript).

(c) Fig 2i – were there any difference in oil-red-O staining or similar in the liver?

According to the referee's request, we performed oil-red-O staining in the livers of Lv-*Noc4l* and found that the accumulation of lipid droplets could not be detected in liver tissue of Lv-*Noc4l* group (Fig. 1n in the revised manuscript). This result was similar to the H&E staining.

9. Issues with Figure 4:

(a) Fig 4f – the definition of an “M1” macrophage is IFN-gamma+LPS-treated – the authors

should revise their nomenclature in describing these data (e.g. M1-like) or perform the experiments using IFN-gamma+LPS.

The wrong nomenclature was corrected as M1-like in the revised manuscript.

(b) Fig 4g – it would be appropriate to examine basal and LPS-induced levels of other inflammatory mediators at the protein level (not just IL-6).

We detected the protein level of TNF α in LPS-induced BMDM, and found that the inflammatory mediator was higher in Noc4l^{LKO} than Noc4l^{f/f} BMDMs (Fig. 4g).

(a) Fig 6c – the representative blot of total IRF3 looks like there's a decrease in total protein of IRF3 in response to LPS in Noc4l k/o cells at 45 and 60 min post-stimulation – is this real? Is IRF3 being degraded in response to LPS treatment? This could affect interpretation of the data in terms of increased phospho-IRF3, as assessed by quantification of pIRF3/total IRF3.

The total protein level of IRF3 was not degraded in response to LPS treatment because the internal reference protein adjustment is not very consistent. The internal reference protein was added in the revised Fig. 5c.

(b) Fig 6d – the authors state that IFN-beta expression is significantly increased in Noc4l k/o macrophages, but this is not the case when looking at the graph. Is there an asterisk missing?

We have added the asterisk on Fig. 5d in the revised manuscript.

(c) Fig 6e – this data set requires expression of a control protein to show that the effect of NOC4L on TRIF responses is specific (i.e. not apparent with a control protein) – to exclude issues such as promoter competition. Perhaps the C-terminus of NOC4L that doesn't interact with TLR4 could be used here? The figure also requires a no TRIF control to be included, so that one can see the baseline level of NF-kappaB activation.

We used the GFP protein as a negative control of NOC4L effect on TRIF response. The result showed that GFP protein can't inhibit the TRIF-induced NF- κ B activation, thus proving that the effect of NOC4L on TRIF response is specific. Moreover, we added a no TRIF control to show the baseline level of NF- κ B activation. As we failed to get the consistent result from C-terminal of NOC4L (answer to point # 2d in Reviewer 1), so using it as a negative control seems unreasonable.

(d) Fig 6f- the above issue (i.e. expression of a control protein) also applies to this figure panel.

Please see the answer for point # 10 (c) in Reviewer 1.

(e) Fig 6h -I require scale bars.

We added the scale bars in Fig 5k in the revised manuscript.

(f) Fig 6j – ideally, co-localization studies with Rabs should be performed in BMMs, since the localization of NOC4L varies depending on the cell type (nuclear vs cytoplasmic). Could this variation in localization reflect primary cells (e.g. BMMs) versus cell lines (mainly used for showing nuclear localization)? Have the authors investigated NOC4L localization in RAW264.7 (macrophage cell line) and/or non-macrophage primary cells?

According to the Reviewer 1's suggestion, we performed the co-localization experiment of Noc4l with Rab5 in BMDMs and the co-localization studies of Noc4l with EEA1 in RAW264.7. Both results showed that Noc4l was localized in endosomes in macrophages (BMDMs and RAW264.7 cells). Moreover, Noc4l was localized in the nuclear of non-macrophage (NIH-3T3, 3T3-L1, GC-1 and GC-2) cells, which can't be collocated with Rab5. In addition, we have not detected the localization of Noc4l in non-macrophage cells, like primary adipocyte.

11. Issue with Fig 7:

(a) Fig 7a – what are the expression levels of TRIF and CD68 in these T2D patients? Were these T2D patients obese or lean? Also, the IL-1beta data appear to be particularly striking in this data set. It is somewhat surprising that IL-1beta (and other mediators that are typically

increased/decreased in obesity/metabolic disease and linked to disease processes e.g. IL-1alpha, IL-10 etc) were not examined in the in vitro and in vivo mouse studies using NCO4L loss-of-function and gain-of-function.

Response: By checking the same expression profiling from the T2D patients, we obtained the increased expression of *TRIF*, whereas no significant change of *CD68* was observed. As *CD68* is highly expressed in the macrophages, microglia, osteoclasts and myeloid dendritic cells (DCs) ¹⁴, and *CD69* is expressed at low levels in freshly isolated peripheral blood monocytes but enhanced by stimulation overnight with GM-CSF and IFN- γ . The high expression of *CD 69* may reflect the inflammatory state of the body under T2D condition.

The BMI of nondiabetic and diabetic patients were 24.4861.2 and 25.0061.81, respectively. The difference between the two groups was insignificant (p= 0.81).

According to the reviewer's request, we detected the mRNA expressions of *IL-1 β* and *IL-10* in LPS (100 ng/ml) stimulated BMDMs from *Noc41^{fl/fl}* and *Noc41^{LKO}* mice at 6h. The results showed that expression of *IL-1 β* was increased and the expression of *IL-10* was decreased in *Noc41^{-/-}* macrophages compared with BMDMs from *Noc41^{fl/fl}* upon LPS stimulation. These results were added in Extended Data Fig. 2d in the revised manuscript.

12. Issues with Extended data Fig 2:

(a) Extended data Fig 2g - requires unstimulated controls so that one can see the baseline levels of inflammatory gene expression.

We added the results of unstimulated controls in Extended Data Fig 2k of the revised manuscript.

(b) Extended data Fig 2h – see concerns above re: interpretation and lack of figure legend.

We added the figure legend of the Extended data Fig. 2h in the revised manuscript.

13. Issues with Extended data Fig 3:

(a) Extended data Fig 3b-c – why switch between cell types for panel b vs c? It would be much more preferable to perform staining for lysosomes, ER and EEA all on RAW264.7 and/or BMM (not switch between HeLa cells and macrophages).

Response: As HeLa cell line is widely used in tumor research, biological experiments or cell culture and easily handled with exogenous gene transfection routinely, while the primary and cell lines of macrophages are known the most difficult cells for exogenous gens transfection. Generally, to study the gene function in macrophage in vitro, researchers normally pick some easily handled cells (like HeLa, HEK293T) to start their studies first. Once there are some results, then most specific cells (such as BMDMs and RAW264.7 cells) will be tested to confirm their findings. Thus, we first transfected the Hela cells to study the localization of *Noc41*, and found that *NOC4L* was collocated with *Rab5*. To further confirm the subcellular localization of *Noc41*, we used RAW264.7 cells/BMDMs to detect the collocation of endogenous *Noc41* and *EEA1/Rab5*.

14. Figure legends – all figures should indicate how many experiments individual data sets are combined from (for each panel). This is not apparent for some of the panels in some of the figures (especially in vitro data). Please also thoroughly check all legends to ensure that there are no mistakes in referring to panels, and in providing necessary information to understand each panel.

We have checked all figure legends carefully and described the results as much detail as possible. Moreover, we have added the information about how many experiments individual data are combined from.

15. Discussion – The authors should discuss the differences in GTT vs ITT data for in vivo studies. See also point #6 relating to modifying the Discussion to avoid over-interpretation of in vivo mechanisms without supporting data.

We have discussed the differences in GTT vs ITT in vivo studies in the revised manuscript in line 373-374.

16. The authors need to cite and discuss the recent paper by Zhu et al (Cell Reports, 2019; PMID: 31018134). Since Tregs are also present in adipose tissue and since there is obvious overlap with the current study (but in a different cell type), this published study has implications for understanding mechanisms by which NOC4L regulates inflammatory responses.

We cited and discussed the recent paper by Zhu et al (Cell Reports, 2019; PMID: 31018134).

Indeed, Treg plays an important role in the suppression of inflammation¹⁵. Treg accounts for 40% of CD4 T cells when mice were under the normal chow condition, and this cell population is higher in adipose than other tissues, such as liver, lung, and spleen. Moreover, obesity can significantly reduce the number of Tregs¹⁶. Lose of Tregs results in worsened metabolic parameters, such as increased fasting blood glucose levels and impaired insulin sensitivity¹⁷. Zhu et al. reported that deficient of Noc4l in Tregs resulting in impairing the activation of Tregs. Thus, we hypothesis that mice deficient of Noc4l in Tregs may develop IR, which was similar to the phenotype of Noc4l^{LKO} mice. This sentenced was added in the discussion session of the revised manuscript.

17. Some of the details in Methods are either very brief (e.g. monoclonal/polyclonal AbS labelling/GST pull downs). The Methods section should be carefully reviewed and updated. generation, nuclear/cytoplasmic extract generation) or are missing altogether (e.g. 35

The details of the method were supplemented in the revised manuscript.

MINOR:

1. Reference(s) are required for the description of RNAseq data for various tissues (lines 109-111).

We have added the reference of RNAseq data for various tissues in line 63-64 in the revised manuscript.

2. Fig 1d-e – include legend on x-axes (time for d, micromolar for e).

We have added the h and μM on x-axes on Fig. 2b and 2c in the revised manuscript.

3. Fig 2a & b – which fat pad from mice were used to assess Noc4l?

We used eWAT to assess the expression of Noc4l and rewrite the figure legend in Fig. 1 in the revised manuscript.

4. Fig 2c –What was the sex, age and background of these patients? This information should be included in the methods. Also, a BMI>30 is generally considered obese, not BMI>28. The authors should justify their BMI criteria.

The information of lean and obese group is as follows:

	Male	Female	BMI	Age
Lean	√		21.6	22
	√		23.3	54
		√	22.9	35
		√	23.7	52
		√	23.7	50
		√	22.4	26
Obese	√		29.8	29
	√		29.3	32
	√		32.1	57
		√	35.3	35
		√	33.7	40
		√	30.1	72

Difference definitions of obesity have been used in different countries¹⁸. In China, the two most commonly used BMI classification for adults is the WHO standard ($\text{BMI} \geq 30 \text{ kg/m}^2$) and the recently developed Chinese standard ($\text{BMI} \geq 28 \text{ kg/m}^2$)¹⁹. The lower BMI is recommended in China because a growing body of evidence suggests that people in China, as well as several other

Asian Pacific populations, has an elevated risk for obesity-related diseases or conditions at a lower BMI than Caucasians¹⁸. We added this information in the MATERIALS AND METHODS in the revised manuscript.

5. Fig 2d – it is unclear how many mice were used in these comparisons and whether the quantification of Noc4l/GAPDH was an average of all WAT tissues or specific ones (e.g. eWAT, iWAT and pWAT). It would be interestingly to know if there was differential expression of Noc4l in different fat depots.

The results represent the data of three mice of each group. The quantification of Noc4l/GAPDH was an average of all WAT tissues. Moreover, we also detected the expression of Noc4l in different fat depots and the result showed that there was no difference in different fat tissues. The result was added in the Extended Data Fig. 1a in the revised manuscript.

6. Fig 2e – the authors should show the AUC quantification graph (not just state a 16% decrease in the text).

We have added the quantification the graph of AUC (in revised Fig. 1h).

7. Fig 3h – insert legend with symbols for WT and OE.

We have inserted the legend of WT and OE on Extended Data Fig. 2l in the revised manuscript.

8. Extended Fig 2d – the authors should provide the list of 200 genes and/or upload the data in GEO database or similar.

We have provided the data in Extended Data Table 3.csv.

9. Fig 4h – is there statistical significance between 2 groups in control for Arg expression (as per text)? If so, the authors should include statistical significance on the graph.

There is statistical significance between 2 groups in control for *Arg1* expression and we have added the statistical significance on Fig. 4i in the revised manuscript.

10. Please indicate protein sizes on Western blots.

We have added the protein markers on Western blots.

Reviewer #2 (Remarks to the Author):

General Comments:

This manuscript by Qin et al investigates the contribution of NOC4L to adipose tissue macrophage (ATM) inflammation and insulin resistance during obesity. The authors demonstrate that NOC4L is decreased in obese humans and in mice and that NOC4L mRNA and protein levels in macrophages are decreased in response to obesity and inflammatory stimuli respectively. They show that lentiviral tail vein injection of NOC4L decreases weight gain during high fat diet challenge and that knocking out NOC4L in macrophages improves glucose tolerance and reduces inflammation during diet induced obesity. They go on to show that NOC4L interacts with toll-like receptor 4 (TLR4) to inhibit endocytosis and block the TRIF pathway to attenuate inflammatory cytokine production. Based on these findings, the authors conclude that NOC4L is a potential biomarker or therapeutic target for insulin resistance. Although the effects of NOC4L on the diet-induced phenotype are clear, and NOC4L is a novel player in regulating inflammation, there are numerous major concerns in the model systems and experimental rigor that lower the impact and interpretation of their results.

Major comments:

1) A central claim of this paper is that NOC4L knockout increases inflammation to worsen insulin resistance in mice during diet induced obesity. However, the perturbations used to alter NOC4L (lentivirus or -/-) also significantly affect body weight. What is the basis for this increase in adiposity? Metabolic cage experiments should be performed. Moreover, the relationship between increased body weight and insulin resistance is well documented in both mice and humans. Thus, how can the authors rule out the possibility that it's the change in body weight and not changes in ATM inflammation that are causing change to insulin resistance? Weight matching experiments, where insulin resistance is measured in wt and Noc4l-/- mice of similar weight, must be included to rule out this possibility. This is particularly important because macrophage inflammatory pathways in the brain are known to regulate feeding behavior.

Response: Thanks for the reviewer's constructive comments. We answer the question of the reviewer one by one as follows.

(1) What is the basis for this increase in adiposity?

As suggested by the reviewer, metabolic cage experiments were performed between Noc4l^{fl/fl} and Noc4l^{LKO} mice on CD and HFD diet. The metabolic rate of mice Noc4l^{LKO} was lower than that of Noc4l^{fl/fl} mice, especially at night (Figs. 3i-3j in the revised manuscript).

(2) How can the authors rule out the possibility that it's the change in body weight and not changes in ATM inflammation that are causing change to insulin resistance?

Thanks to the reviewer to put a very good question and advice.

In order to rule out the possibility of the effect of body weight on insulin resistance, we performed an insulin tolerance test (ITT) to assess the systemic insulin sensitivity on Noc4l^{LKO} and Noc4l^{fl/fl} mice with similar weight on HFD. Insulin tolerance of Noc4l^{LKO} mice was higher than the Noc4l^{fl/fl} weight matched mice with HFD (Fig. 4m in the revised manuscript).

Next, to further examine the role of inflammation in the induction of insulin resistance, we used a nonlethal dose of bacterial lipopolysaccharide (LPS) to induce acute inflammation. Following an overnight fast and LPS (1 mg/kg) intraperitoneal injection, we performed GTTs. All mice became hyperglycemic after LPS injection, but Noc4l^{LKO} mice were considerably more glucose tolerant than Noc4l^{fl/fl} mice after glucose loading (Fig. 4l in the revised manuscript).

These results suggested that the insulin sensitivity is due to the change in inflammatory state rather than change in body weight.

2) Although ATM inflammation is the main focus of the paper, there are no measurements that

directly demonstrate that ATM inflammation is altered in Noc4l^{-/-} mice. Inflammatory cytokine measurements are made on whole adipose tissue or plasma, neither of which directly demonstrates that ATMs are actually inflamed. Moreover, the results of the whole adipose tissue qRT-PCR (Fig. 4b) shows increases in F4/80 and CD68, raising the possibility that an increase in ATM number maybe more important than an increase in ATM inflammatory state per se (which is predicted by their mechanistic work). The authors should isolate ATMs from mice and quantify inflammatory cytokine expression within them under the various perturbations.

Response: As Lysozyme M cyclization recombinase is not ATM specific but myeloid cell specific promoter, thus, deletion of Noc4l in macrophages resulted in a systemic malfunction of macrophages which may lead to the systemic inflammatory responses, rather than ATM only. Please see the answer for point # 5 of Reviewer 1.

Based on the proinflammatory cytokines of BMDMs from WT and Noc4l^{-/-} mice stimulated with LPS, we concluded that LPS increased the inflammatory state of Noc4l^{-/-} mice.

3) The authors use two in vitro inflammatory perturbations: lipopolysaccharide (LPS) and the fatty acid palmitate to study macrophage inflammation. LPS is a well-established method to classically activate (M1) macrophages to mimic their phenotype during infection, while palmitate is a more physiologically relevant stimulus to study macrophage inflammation during diet-induced obesity. The authors treat these two stimulations as equal for much of the manuscript, and then focus their more detailed mechanistic work on LPS signaling. Recent studies suggest that palmitate induces a metabolically activated (MMe) phenotype that is dramatically different from M1 macrophages both in terms of protein expression and the mechanisms supporting inflammation (Kratz et al, Cell Metab, 2014; Coats et al, Cell Rep. 2017). Indeed, recent studies showed that although Tlr4^{-/-} attenuates inflammation in palmitate-treated macrophages, TLR4 does not directly bind palmitate and influences its signaling indirectly (Lancaster et al., Cell Metab., 2018). These papers bring to question the strategy of using LPS as a model system to uncover mechanisms that are at play during diet-induced obesity. Based on this rationale, it would be more appropriate for the authors to determine mechanisms of how Noc4l^{-/-} impacts inflammation in palmitate-challenged macrophages.

Response: We have seriously considered this issue:

To some extent, both LPS and FFA can induce the cytokine expression in adipose tissue²⁰. LPS induces M1 macrophage and palmitate induces a metabolically activated (MMe) phenotype that is dramatically different from M1 macrophages²¹.

To answer Reviewer 2's question, we used palmitate to induce MMe phenotype and detected the mRNA expressions of *ABCA1* and *PLIN2*, which are reported to involve in lipid metabolism (Kratz et al, 2014; Coats et al, 2017). Indeed, we found the increased expressions of *ABCA1* and *PLIN2* upon palmitate induction, whereas no difference in the expressions of *ABCA1* and *PLIN2* were observed between the Noc4l^{LKO} and Noc4l^{fl/fl} MMe (Please see the Fig. 3).

Moreover, even without HFD feeding, glucose tolerance of Noc4l^{LKO} mice was significantly impaired relative to the Noc4l^{fl/fl} mice with CD (Fig. 3d in the revised manuscript). These results suggested that Noc4l is more involved in LPS induced inflammatory pathway, but not in PA induced metabolic activation. In our case, LPS may play some kind of roles in mimic high fat diet induced inflammation and could be responsible for impaired metabolism.

Fig. 3

4) The authors present nice data to show the mechanism underlying NOC4L-TLR4 interactions, as it pertains to inflammatory signaling. However, these data were acquired with cells lines, most of which are not of the monocyte/macrophages lineage, and with a stimulus (LPS) that is not physiologically relevant to studying diet-induced obesity insulin resistance. Moreover, there are no confirmatory data in ATMs in vivo. It is therefore recommended for authors to confirm their mechanism in palmitate-challenged macrophages in vitro, and to provide evidence of similar effects in ATMs isolated from lean and obese wt and Noc4L^{-/-} mice.

Response: Indeed, ATM is the most suitable cell for studying adipose tissue inflammation. In our study, we prepared myeloid specific knockout mice by using the Lysozyme M cyclization recombinase (Lys M) cre mice, which deleted Noc4l in all kinds of macrophages, resulted in increased inflammation both systemically and locally. Under this condition, BMDMs and peritoneal macrophages are more suitable cells in our case. We stated this issue on the points #1 and 2 of Reviewer 2.

Additionally, we detected the interactions of NOC4L-TLR4 in RAW264.7 cell line and Noc4l^{fl/fl} and Noc4l^{LKO} BMDMs (Figs. 6b-6c in the revised manuscript). All these results showed that NOC4L interacted with TLR4.

Moreover, our results showed that NOC4L was more involved in M1 activation but not metabolic activation, please see the answer for point #3 of Reviewer 2.

5) One of the major claims of this manuscript is that the interaction between TLR4 and NOC4L is required to inhibit internalization of TLR4 and downstream inflammatory signaling. The authors use pulldowns and a proximity ligation assay to demonstrate that TLR4 and NOC4L interact via the N-terminal domain of NOC4L and the TIR domain of TLR4. However, the necessity of this interaction remains unexplored by the authors, leaving questions as to its relevance to the biology. Does inhibiting the interaction between these regions mimic the phenotype of the knockout mice/cell lines?

LPS triggers innate immune responses via Toll-like receptor (TLR) 4²², which contains two major pathways: the MyD88-dependent and MyD88-independent pathways (TRIF-dependent pathway)²³⁻²⁵ (Please see the Fig. 4).

Fig. 4

The adaptors of MyD88 and TRIF engage the same domain (TIR domain) of the TLR4, and are probably in competition with each other. LPS signals are silenced by mutating or blocking the TIR domain of TLR4. In our case, Noc4l was interacted with TIR domain of TLR4. We speculated that NOC4L and TRIF competitively bind TLR4 to inhibit downstream inflammatory pathways.

Even point mutant or molecular inhibitor can block the TIR domain of TLR4, these methods have the inhibitory effects on both on MyD88 and TRIF signal pathways, simultaneously, thus eliminating the inflammatory responses of downstream.

Next, there are few reports about NOC4L at present. Due to technical limitations, we had not found or synthesized a small molecule that can effectively inhibit the interaction domain of NOC4L.

But our results demonstrated from another perspective that this interaction is necessary for regulating inflammation. First, NOC4L interacted with TIR domain of TLR4. Second, Noc4l can inhibit the endocytosis of TLR4. Next, inhibition of the TLR4 endocytosis by Dynasore decreased the production of TNF α , and there was no difference between the Noc4l^{LKO} and Noc4l^{fl/fl} mice. Thus, we hypothesized that the interaction of NOC4L and TLR4 can inhibit the inflammatory response.

Other comments:

1) Lentiviral overexpression of proteins is a well characterized way to perform in vivo modifications, in the same way the LysMCre model has been used to study the relevance of proteins in myeloid cells. However, in both cases the specificity of the modifications may vary in different macrophages types. To provide more rigor to their perturbations, the authors need to confirm their overexpression and knockouts in isolated ATMs.

We demonstrated that Noc4l was effectively deleted from Noc4l^{LKO} ATMs at the mRNA level (Extended Data Fig. 1e in the revised manuscript). We examined the overexpression efficiency of Noc4l in fat and liver tissues and confirmed that Noc4l was overexpressed in fat and liver tissues with Lv-Noc4l delivery. However, we observed no increased expression of *Noc4l* in ATMs after the Lentivirus injection, this may probably due to the low overexpression efficiency via lentivirus transfection in vivo. In order to get the consistent results, we constructed the gene knockout and knock in mouse models.

2) Many of the figures and figure legends show a lack of attention to detail on the part of the authors. There are several instances where the cell lines being studied, the n for the experiment, the antibodies used for staining for IHC are not clear. It is recommended for the authors to revise all figure legends to improve clarity.

We have carefully revised the article and described the details of figures and figure legends.

Reviewer #3 (Remarks to the Author):

In this manuscript "Macrophage deletion of Noc4l triggers endosomal IRF3/IFN β signal and leads to insulin resistance", the authors determine the role of Noc4l in regulation of macrophage responses in adipose tissue during obesity. They identify regulation of TLR4 by Noc4l followed by control of INF β activity as a crucial pathway mediating adipose tissue inflammation and ultimately regulating insulin sensitivity during obesity.

Using different approaches ranging from in vitro to in vivo animal models and human adipose tissue, the authors conclude that Noc4l protects against adipose tissue inflammation and improves insulin sensitivity.

The manuscript reads well and contains an impressive amount of experiments. However, this also represents somewhat of a weakness, as the authors have attempted to cover many different aspects that sometimes leads to incomplete results and preliminary conclusions.

Please find below my comments and suggestions that may help to improve the quality of the current manuscript.

1. The authors show that noc4l controls TLR4 signalling by regulating its internalization. This also implies that TLR4-ligands are the main drivers of adipose tissue inflammation. In addition to fatty acids that are being debated as true TLR4 ligands, what other potential ligands may exist in adipose tissue? Also, LPS and PA treatment leads to downregulation of noc4l expression in macrophages. Although downregulation of noc4l is also observed in vivo in obese adipose tissue, do the authors have any data available related to dynamics of noc4l expression?

In other words, how is the expression regulated during DIO and are effects on gene expression seen early or late after the start of the HFD-intervention?

Response: We have consulted a large amount of literatures, and the well-known ligands of TLR4 are mainly LPS and PA so far. We could not clearly figure out other potential ligands of TLR4 that may exist in adipose tissue.

We did have the results for the dynamics of Noc4l expression in WT mice. Noc4l expression was gradually decreased during HFD induction. At the 7th week of induction, the downward trend was more obvious, and at 10 weeks (n=3/group), the expression decreased dramatically (please see the Fig. 5). It has been reported that the increased expressions of F4/80, CD11c, and CD11b by DIO started at week 8²⁶. Compared to Strissel' study, we found that the decrease of NOC4L level is no later than the occurrence of inflammation.

(Katherine J. Strissel et al,2007)

Fig. 5

Also, obesity promotes the influx of macrophages into the adipose tissue. However, noc4l expression is reduced. Would this mean that expression is lowered within macrophages residing in adipose tissue or that the infiltrating macrophages are of a different phenotype with low expression levels of noc4l?

Response: I am very grateful to the Reviewer 3 for his/her comments. As it has been reported that ATMs that reside in AT do not have CD11c whereas the ATMs that are newly recruited by obesity express CD11c. The obesity-induced changes in ATMs numbers are mainly due to an increase the triple-positive CD11b⁺ F4/80⁺ CD11c⁺ ATM subpopulation ²⁷. By sorting the F4/80⁺CD11b⁺CD11c⁻ and F4/80⁺CD11b⁺CD11c⁺ cells from eWAT of mice with HFD at 12weeks, we detected the expressions of *Noc4l* in each subpopulation. The result showed that the expressions of *Noc4l* were decreased in both resident and recruited macrophages compared to ATM in CD group (please see the Fig. 6).

As those NOC4L expression results from subpopulations lowered was not very relevant to our story, and we did not add this data in the revised manuscript.

Fig. 6

Also, the authors should provide more information about the study participants that were included. Age, BMI, medication, Hba1c, other relevant parameters?

The patient's relevant information is as follows. We only collected the information of gender, BMI and age. It will take a long time to recollect the other information.

	Male	Female	BMI	Age
Lean	√		21.6	22
	√		23.3	54
		√	22.9	35
		√	23.7	52
		√	23.7	50
		√	22.4	26
Obese	√		29.8	29
	√		29.3	32
	√		32.1	57
		√	35.3	35
		√	33.7	40
		√	30.1	72

2. The results section is very long and contains a long list of different experiments that sometimes lack coherence. I would also suggest to carefully reconsider the order of presenting the results. For example, I would first present results on regulation of noc4l in adipose tissue during obesity, before presenting data related to the regulation of noc4l by LPS and PA. Also, data on the interaction between noc4l and TLR4 presented in figure 5 is of interest from a biochemical point of view, yet is difficult to translate to in vivo relevance. Also, how would these results fit with the qPCR data on inflammatory gene expression in untreated ko BMDMs?

Response: We have adjusted the order of presenting the results. For details of the adjustment, please see the answer for point #2 of Reviewer 1.

We tried to study the physiological significance of the interaction between TLR4 and NOC4L, but due to the limitations of experimental conditions, we have no way to conduct related research. The relevant restrictions are explained on the point #5 of the Reviewer 2.

The answer for the hyper-inflammatory phenotype was explained and please see the answer for point #6 of Reviewer 1.

3. Based on the data presented in the second paragraph ending on page 9, the authors conclude that *noc4l* could directly improve IR in vivo. I think the authors should rephrase this text. Based on the findings, one could conclude that *noc4l* is downregulated in adipose tissue during obesity and that overexpression using a lentivirus, that primary leads to an increase in hepatic expression of *noc4l*, lowers glucose intolerance. However, overexpression also reduces BW and fat mass, both known to impact on glucose tolerance.

Hence, *noc4l* may not directly improve IR, but more indirectly via affecting the BW of the animals. Btw, any information on food intake or activity levels that may explain the difference in weight gain? Same holds true for the higher BW in the knockout animals. The authors need to provide more data related to a potential cause for the differences in BW. Again, by impacting on BW/fat mass all effects of *noc4l* on IR and adipose tissue inflammation could be largely independent from its effects on TLR4 signalling.

According to the Reviewer 3's question, we have rephrased this paragraph and toned down our conclusion according to the context of the revised manuscript.

The answer for body weight on insulin resistance, please see the answer for point #1 of Reviewer 2. In addition, there was no difference in activity levels between *Noc4l*^{fl/fl} and *Noc4l*^{LKO} mice. Moreover, we observed that *Noc4l*^{LKO} mice have lower food intake than *Noc4l*^{fl/fl} mice. Based on these results, we excluded the food intake and activity levels as the reasons for the body weight gain.

Previous study suggested that continuous subcutaneous infusion of low dose LPS for 4 weeks lead to increased fasted glycemia and insulinemia and whole-body, liver, and adipose tissue weight gain in mice²⁸. Thus, this weight gain may mainly due to chronic inflammation of *Noc4l*^{LKO} mice in our study.

4. Page 9, line 183. What type of macrophages have been isolated and tested? ATMs?

The bone marrow derived macrophages were tested for the deletion efficiency of *Noc4l* at mRNA and protein levels and we have corrected the description of this paragraph (line 126-127 in the revised manuscript). Moreover, we also detected the deletion efficiency of *Noc4l* in ATMs using RT-PCR, and our results showed that the expression of *Noc4l* was decreased in ATMs (Extended Data Fig. 1e in the revised manuscript).

5. The authors claim that *INFb* signalling is the key pathway impacting on insulin sensitivity, as stated in the title. However, no evidence in the context of the adipose tissue is presented in the manuscript. Although data presented in figure 6 show a role of *noc4l* in controlling *INFb* expression levels in BMDMs, what about *INFb* levels in vivo in adipose tissue in WT vs. myeloid-specific *noc4lko* animals? In addition, how do the authors conclude that *INFb* contributes to the development of insulin resistance in vivo? In general, the authors should tone down some of the conclusions made in the manuscript and also adapt the title. In addition, more work is needed to decipher the mechanism of action leading to differences in BW due to *noc4l*.

We thank the reviewer's critical comments and suggestions. Indeed, we overstated the effects of *IFNβ*, as *INFβ* is one of downstream targets of TRIF, there are some other inflammatory factors such as *TNFα* and *CCL5* also the target of TRIF^{23,29}.

Moreover, we checked the expression of *IFNβ* in adipose tissue, and found that its expression level is very low, and the number of peak cycles in qRT-PCR is about 35, and this result has higher uncertainty. We also consulted the literature and found that its expression in adipose tissue is indeed much lower than other inflammatory factors³⁰. Moreover, we prepared myeloid knockout mice, which play an important role not only in local but also in systemic inflammatory regulation. The expression of *IFNβ* from *Noc4l*^{LKO} BMDMs was higher than *Noc4l*^{fl/fl} BMDMs. In this way we toned down our conclusion from the title and the context in the revised manuscript.

The mechanism leading to differences in BW due to *Noc4l* has stated on point #3 of Reviewer 3.

6. The authors show and discuss the relevance of *noc4l* in controlling *m1* vs. *m2* polarization. Although this is of importance, the relevance of these phenotypes in adipose tissue *in vivo* may be somewhat limited. What about a role of *noc4l* in controlling metabolic activation of macrophages using triggers that are more relevant for adipose tissue?

Response: Indeed, macrophages in adipose tissue play a vital role in inflammation and insulin resistance. However, in our study, we prepared myeloid specific knockout mice by using the Lysozyme M cyclization recombinase (Lys M) cre mice, which deleted *Noc4l* in all kinds of macrophages, resulted the increased inflammation both systemically and locally. (please see the answer for point #5 of Reviewer 1).

The role of *Noc4l* in controlling metabolic activation of macrophages using triggers that are more relevant for adipose tissue, please see the point #3 of Reviewer 2.

Reference:

- 1 Milkereit, P. *et al.* A Noc complex specifically involved in the formation and nuclear export of ribosomal 40 S subunits. *J Biol Chem* **278**, 4072-4081, doi:10.1074/jbc.M208898200 (2003).
- 2 Mackmull, M. T. *et al.* Landscape of nuclear transport receptor cargo specificity. *Mol Syst Biol* **13**, 962, doi:10.15252/msb.20177608 (2017).
- 3 Fagerberg, L. *et al.* Analysis of the human tissue-specific expression by genome-wide integration of transcriptomics and antibody-based proteomics. *Mol Cell Proteomics* **13**, 397-406, doi:10.1074/mcp.M113.035600 (2014).
- 4 Qin, Y. *et al.* Targeted disruption of *Noc4l* leads to preimplantation embryonic lethality in mice. *Protein Cell* **8**, 230-235, doi:10.1007/s13238-016-0335-9 (2017).
- 5 Zhu, X. *et al.* *Noc4L*-Mediated Ribosome Biogenesis Controls Activation of Regulatory and Conventional T Cells. *Cell Rep* **27**, 1205-1220 e1204, doi:10.1016/j.celrep.2019.03.083 (2019).
- 6 Zhang, R. *et al.* HLA-B-associated transcript 3 (*Bat3*) stabilizes and activates p53 in a HAUSP-dependent manner. *J Mol Cell Biol*, doi:10.1093/jmcb/mjz102 (2019).
- 7 Samir, P. *et al.* DDX3X acts as a live-or-die checkpoint in stressed cells by regulating NLRP3 inflammasome. *Nature* **573**, 590-594, doi:10.1038/s41586-019-1551-2 (2019).
- 8 Lucey, B. P., Nelson-Rees, W. A. & Hutchins, G. M. Henrietta Lacks, HeLa cells, and cell culture contamination. *Arch Pathol Lab Med* **133**, 1463-1467, doi:10.1043/1543-2165-133.9.1463 (2009).
- 9 Nelson-Rees, W. A., Daniels, D. W. & Flandermeyer, R. R. Cross-contamination of cells in culture. *Science* **212**, 446-452, doi:10.1126/science.6451928 (1981).
- 10 Broderick, R., Nieminuszczy, J., Blackford, A. N., Winczura, A. & Niedzwiedz, W. TOPBP1 recruits TOP2A to ultra-fine anaphase bridges to aid in their resolution. *Nat Commun* **6**, doi:ARTN 6572 10.1038/ncomms7572 (2015).
- 11 Williams, M. J. *et al.* Obesity-linked homologues TfAP-2 and Twz establish meal frequency in *Drosophila melanogaster*. *PLoS Genet* **10**, e1004499, doi:10.1371/journal.pgen.1004499 (2014).
- 12 Arkan, M. C. *et al.* IKK-beta links inflammation to obesity-induced insulin resistance. *Nat Med* **11**, 191-198, doi:10.1038/nm1185 (2005).
- 13 Jais, A. *et al.* Myeloid-Cell-Derived VEGF Maintains Brain Glucose Uptake and Limits Cognitive Impairment in Obesity. *Cell* **165**, 882-895, doi:10.1016/j.cell.2016.03.033 (2016).
- 14 Greaves, D. R. & Gordon, S. Macrophage-specific gene expression: current paradigms and

- future challenges. *Int J Hematol* **76**, 6-15, doi:10.1007/bf02982713 (2002).
- 15 Josefowicz, S. Z., Lu, L. F. & Rudensky, A. Y. Regulatory T cells: mechanisms of differentiation and function. *Annu Rev Immunol* **30**, 531-564, doi:10.1146/annurev.immunol.25.022106.141623 (2012).
- 16 Feuerer, M. *et al.* Lean, but not obese, fat is enriched for a unique population of regulatory T cells that affect metabolic parameters. *Nat Med* **15**, 930-U137, doi:10.1038/nm.2002 (2009).
- 17 Becker, M., Levings, M. K. & Daniel, C. Adipose-tissue regulatory T cells: Critical players in adipose-immune crosstalk. *Eur J Immunol* **47**, 1867-1874, doi:10.1002/eji.201646739 (2017).
- 18 Wang, Y., Mi, J., Shan, X. Y., Wang, Q. J. & Ge, K. Y. Is China facing an obesity epidemic and the consequences? The trends in obesity and chronic disease in China. *Int J Obes (Lond)* **31**, 177-188, doi:10.1038/sj.ijo.0803354 (2007).
- 19 Zhou, B. F. & Cooperative Meta-Analysis Group of the Working Group on Obesity in, C. Predictive values of body mass index and waist circumference for risk factors of certain related diseases in Chinese adults--study on optimal cut-off points of body mass index and waist circumference in Chinese adults. *Biomed Environ Sci* **15**, 83-96 (2002).
- 20 Shi, H. *et al.* TLR4 links innate immunity and fatty acid-induced insulin resistance. *J Clin Invest* **116**, 3015-3025, doi:10.1172/JCI28898 (2006).
- 21 Kratz, M. *et al.* Metabolic dysfunction drives a mechanistically distinct proinflammatory phenotype in adipose tissue macrophages. *Cell metabolism* **20**, 614-625, doi:10.1016/j.cmet.2014.08.010 (2014).
- 22 Tanimura, N., Saitoh, S., Matsumoto, F., Akashi-Takamura, S. & Miyake, K. Roles for LPS-dependent interaction and relocation of TLR4 and TRAM in TRIF-signaling. *Biochem Biophys Res Commun* **368**, 94-99, doi:10.1016/j.bbrc.2008.01.061 (2008).
- 23 Kawai, T. *et al.* Lipopolysaccharide stimulates the MyD88-independent pathway and results in activation of IFN-regulatory factor 3 and the expression of a subset of lipopolysaccharide-inducible genes. *Journal of Immunology* **167**, 5887-5894 (2001).
- 24 Hoebe, K. *et al.* Identification of Lps2 as a key transducer of MyD88-independent TIR signalling. *Nature* **424**, 743-748, doi:10.1038/nature01889 (2003).
- 25 Kuzmich, N. N. *et al.* TLR4 Signaling Pathway Modulators as Potential Therapeutics in Inflammation and Sepsis. *Vaccines-Base* **5**, doi:ARTN 34 10.3390/vaccines5040034 (2017).
- 26 Strissel, K. J. *et al.* Adipocyte death, adipose tissue remodeling, and obesity complications. *Diabetes* **56**, 2910-2918, doi:10.2337/db07-0767 (2007).
- 27 Lumeng, C. N., Bodzin, J. L. & Saltiel, A. R. Obesity induces a phenotypic switch in adipose tissue macrophage polarization. *J Clin Invest* **117**, 175-184, doi:10.1172/JCI29881 (2007).
- 28 Cani, P. D. *et al.* Metabolic endotoxemia initiates obesity and insulin resistance. *Diabetes* **56**, 1761-1772, doi:10.2337/db06-1491 (2007).
- 29 Zanoni, I. *et al.* CD14 controls the LPS-induced endocytosis of Toll-like receptor 4. *Cell* **147**, 868-880, doi:10.1016/j.cell.2011.09.051 (2011).
- 30 Jain, P. *et al.* Systems biology approach reveals genome to phenome correlation in type 2 diabetes. *PLoS One* **8**, e53522, doi:10.1371/journal.pone.0053522 (2013).

Reviewer comments, second round –

Reviewers' comments:

Reviewer #2 (Remarks to the Author):

The revised manuscript is much improved and the authors have been responsive to reviewers comments. However, one major point was not addressed. It remains unclear whether the NOC4L knockout alters inflammatory cytokine expression in macrophages that are known to promote insulin resistance in vivo. This issue was raised by 2 reviewers and it has not been adequately addressed. I respectfully disagree with the authors claim that BMDMs and peritoneal macrophages are more appropriate here. Validation in ATMs (or Kupffer cells) are needed to validate the authors' mechanistic claims and strengthen their paper.

Reviewer #3 (Remarks to the Author):

The authors have responded to most of the comments. However, I still have several issues that need to be addressed. Based on the data, it seems that the pathway is particularly relevant for LPS-mediated inflammation. However, the relevance of LPS in driving metabolic inflammation is somewhat lower. It would be important to identify the potential drivers of the pathway in vivo modulating insulin sensitivity. Also, the authors haven't responded to the comments related to activation of ATMs. I understand that the Lysm model leads to knockdown in myeloid cells, but it would still be important to learn more about the consequences of the knockdown for adipose tissue resident macrophages and their inflammatory potential.

Reviewer #4 (Remarks to the Author):

In this study, Qin et al study the role of Noc4l in obesity and insulin resistance. They show that indeed Noc4l mRNA expression is decreased in obese mice and human adipose tissues, and they show that Noc4l knockout in macrophages results in increased body weight. Further, overexpression of Noc4l using lentiviral system rescues the insulin resistance and adipose tissue weights, however, it's unclear whether it also rescued total body weight of these mice? Mechanistically, they show that macrophage-specific knockout of Noc4l improves glucose processing, insulin resistance and diet-induced obesity in response to HFD. Further, they suggest that Noc4l interacts with TLR4 to alter endocytosis pathways in macrophages to reduce inflammatory cytokine gene expression. However, the evidence for macrophage-only Noc4l driving diet-induced obesity and also the interactions of Noc4l-TLR4 in macrophage endocytosis is weak. In my opinion, the authors need to provide additional experimental evidence to support their claims.

The authors have addressed some of the reviewers' comments and the manuscript overall does read better, however, there are still some major concerns that are outlined below:

1. Tregs are also known to play an important role in insulin resistance during diet-induced obesity. In fact, the decrease in Noc4l expression in SVF (i.e. most immune cells in the adipose such as iNKTs, Tregs, B cells, macrophages etc) or adipose tissue during HFD-feeding (~7-8 weeks onwards) coincides with the expected decrease in Tregs. Thus, it's plausible that the decrease in Noc4l expression may be due to the decrease in Tregs itself or other inflammatory cells such as iNKTs (another cell type that is reduced during HFD feeding). Further, given the role of Noc4l in Tregs that is already known, it would be important to clarify the role of Noc4l in Tregs and other inflammatory cells in the adipose, not just ATMs, during diet-induced obesity. The characterisation of these immune cells in the adipose tissue can be easily performed using flow cytometry sorting

and/or bead-isolation kits.

2. In a related note, the authors need to verify that Noc4l is indeed knocked down in adipose tissue macrophages (ATMs), not just BMDMs, from their Noc4lKO mice, especially considering that do they do observe both systemic and local inflammatory changes in these mice. Again, these experiments can be performed by isolating F4/80+ macrophages using flow cytometry sorting or bead-isolation kits, and standard qPCR and western blotting. It's quite possible that the BMDMs may be driving systemic inflammation that may have an effect on local tissues or vice versa. Please note, in the response to reviewers' (to reviewer #3, pt.4), the authors claim to have measured Noc4l in ATMs but in the figure they refer to the figure legend states BMDMs.

Also, importantly, in Fig 4b, the authors should measure the inflammatory gene expression in adipose tissue macrophages (ATMs; and/or other inflammatory cells) specifically to verify that indeed macrophage-specific KD of Noc4l is directly affecting inflammatory gene expression within these cells. As it's also possible that an increase in number of macrophage recruitment into the adipose tissue (which is indicated by CD68 and F4/80 increased gene expression) is the sole reason for the difference in inflammatory gene expression in adipose tissue.

3. The data showing the interaction between Noc4l and TLR4, and the role of Noc4l in endocytosis needs more experimental evidence. The authors have demonstrated interaction between TLR4 and Noc4l using IP expts in macrophages. However, the signal for Noc4l is WT BMDMs (Fig 6c) is so faint (relative to input), suggesting only a tiny fraction of Noc4l actually interacts with TLR4 under resting conditions This poses the question of the significance of such interaction etc. Thus, further experimentation is required to verify this mechanism, and some initial starting points could include the following:

- a. The authors should determine whether they can co-IP TLR4 and Noc4l from PA- (and LPS-) stimulated BMDMs to determine whether stimulation promotes or interferes with this interaction.
- b. To confirm localisation of Noc4l (and TLR4) to endosomes, sub-cellular fractionation of early and late endosomes, followed by western blotting. These experiments can also be performed in PA-stimulated samples. It is good practice to validate these pathways using multiple approaches.
- c. Fig 5: Were any of the downstream inflammatory signalling molecules (e.g. IKK or NF- κ B or their phosphorylation) affected by NOCL4LKO? This would further support the author's hypothesis that Noc4l affects inflammatory signalling.
- d. As this is the first time that the authors are showing an interaction between TLR4-Noc4l, it would indeed be good to use another protein EGFP fusion protein (e.g. TLR2 or another protein or a mutated/truncated TLR4) to show the absence of such interaction (i.e. it's not purely driven by an over-expression system but indeed specific interaction).
- e. Also, the above experiments can be performed in BMDMs derived from Noc4lKO mice. For example, to further confirm defecton TLR4 localisation and/or endocytosis. Further, the opposite (potentially) interactions could be shown in OE of Noc4l in BMDMs using lentiviral system (as authors have performed in the manuscript).
- f. In vivo, TLR4 surface expression can be verified from ATMs isolated from HFD-fed Noc4lKO mice and their control counterparts.
- g. The larger question remains as to how macrophage TLR4-dependent inflammation in Noc4lKO mice directly results in adipose tissue expansion and insulin resistance. This remains unclear.

4. The authors show that in response to HFD the Noc4lKO mice are metabolically challenged, where they have a decrease in VO₂, VCO₂, and energy expenditure (EE) during the night cycle (Fig 3). In their response to the reviewers, they mention that these mice also had a decrease in food consumption, however, they have not shown this data. It does make sense that if the mice do consume less food, they are likely to have decreased energy usage. However, the mechanisms by which these mice have increased adipose tissue and body weights are still unclear, and raise a very important question. Were there any differences in their excretion, especially fatty acids/lipids etc, as this could also potentially account for differences in their food absorption abilities and potential weight gain? Or potentially Noc4l deletion could also affect their appetite? Were there any weight differences in other organs (e.g. liver, spleen, muscle)? It's unlike that insulin resistance accounts for the body weight gain? This data, although interesting, is quite confusing, and further experiments are required to decipher the role of Noc4l in increasing adipose tissue and body weights.

5. Some critical in vitro macrophage experiments are also missing. For example, gene expression of inflammatory cytokines is not always reflective of cytokine production and secretion. Authors should indeed confirm the secretion of IL-6, TNF α etc in response to palmitic acid (and LPS) in BMDMs or PMs. Further, the authors should verify that Noc4l KD or OE does not impact naive macrophages ability to become anti-inflammatory 'M2-like' macrophages (i.e. measure 'M2-like' markers in response to IL-4 and/or IL-13).

Minor:

1. Some of the new data generated in response to the reviewers' comments were not included in the manuscript and would be critical to do so. For example, the measurement of Noc4l expression in CD11b \pm ATMs. Also, the patient information should be included as a summary table of number of females vs males, average age and average BMI etc.

2. Did they test their Noc4l antibodies in the knockout cells or tissues from mice to verify specificity of the antibody.

3. The authors should verify that Noc4l was not knockdown in other cells (e.g. T cells etc) in the Noc4lKO mice – i.e. there's no leak with the LysM cre system.

4. The data with HFD-fed OE mice in ITT (insulin sensitivity) is also confusing. The authors should discuss the limitations and explain this data. It may be worthwhile to perform ITT at a different time point (e.g. 10-12wks) to see if they do observe a difference. Overall, the authors should discuss the differences between OE and KO data and at least postulate the reasons why they do observe these differences in vitro and in vivo.

5. It would greatly assist the reviewers if the authors tracked changes or highlighted the changes in the actual manuscript file. Also, having the manuscript in portrait format would make it easier to read (rather than landscape).

6. New Fig 5a – it would be better to show a lighter and darker exposure of p-ERK as it's hard to see the bands in the current blot.

7. Lastly, there remain some minor grammatical and typographical issues with the current submission.

Responds to the reviewer's comments:

Reviewer #2 (Remarks to the Author):

The revised manuscript is much improved and the authors have been responsive to reviewers comments. However, one major point was not addressed. It remains unclear whether the NOC4L knockout alters inflammatory cytokine expression in macrophages that are known to promote insulin resistance in vivo. This issue was raised by 2 reviewers and it has not been adequately addressed. I respectfully disagree with the authors claim that BMDMs and peritoneal macrophages are more appropriate here. Validation in ATMs (or Kupffer cells) are needed to validate the authors' mechanistic claims and strengthen their paper.

Response: We thank for the Reviewer 2's constructive suggestion. According to his/her advice, we assessed the inflammatory cytokines expression from the isolated ATMs of *Noc4l^{fl/fl}* vs. *Noc4l^{LKO}* on HFD treatment. The mRNA expressions of *TNF α* and *IL-6* of ATMs of *Noc4l^{LKO}* were higher than those of *Noc4l^{fl/fl}*, which were shown in **Fig. 4i and line 231-232 labeled red** in the revised manuscript. Accumulation of ATMs in obese adipose is partly derived from bone marrow precursors (Hill et al., 2018; Weisberg et al., 2006). Thus, we confirmed that *Noc4l* deficiency in macrophage promotes inflammatory cytokines expression in ATM which is consistent with the results of BMDMs and peritoneal macrophages.

Reviewer #3 (Remarks to the Author):

The authors have responded to most of the comments. However, I still have several issues that need to be addressed.

Based on the data, it seems that the pathway is particularly relevant for LPS-mediated inflammation. However, the relevance of LPS in driving metabolic inflammation is somewhat lower. It would be important to identify the potential drivers of the pathway in vivo modulating insulin sensitivity.

Also, the authors haven't responded to the comments related to activation of ATMs. I understand that the *Lysm* model leads to knockdown in myeloid cells, but it would still be important to learn more about the consequences of the knockdown for adipose tissue resident macrophages and their inflammatory potential.

Response: We thank to the Reviewer 3's comment. LPS has been identified and accepted as one of the critical triggering factors in the early development of inflammation and metabolic deterioration (Cani et al., 2007; Cao et al., 2017; Konner and Bruning, 2011; Shi et al., 2006; Winer et al., 2016). LPS is produced in the gut of the host by the death of Gram-negative bacteria, followed by translocation into intestinal capillaries via a TLR4-dependent mechanism, and

transport to the target tissues, where it triggers signaling by binding to and activating TLR4 receptors (Cani et al., 2007) . LPS-TLR4 pathway is responsible for the heightened pro-inflammatory milieu in human obesity (Saltiel and Olefsky, 2017). High fat diet increases the proportion of LPS-containing microbiota in the gut, and under a 4-week high fat diet treatment chronically elevated plasma LPS levels were observed in mice which were defined as metabolic endotoxemia (Cani et al., 2007; Konner and Bruning, 2011). The upregulated inflammatory cytokines are key inducers of insulin resistance. Taken together, these studies confirmed that LPS plays the important roles in driving metabolic inflammation. Furthermore, we are also exploring other potential drivers of the pathway in vivo for modulating the insulin sensitivity.

In terms of the activation of ATMs, please see the answers in point for Reviewer 2. In addition, the decreased expression of Noc4l in ATMs was shown in **Extended Data Fig.1f and line 146-147 labeled red** in the revised manuscript.

Reviewer #4 (Remarks to the Author):

In this study, Qin et al study the role of Noc4l in obesity and insulin resistance. They show that indeed Noc4l mRNA expression is decreased in obese mice and human adipose tissues, and they show that Noc4l knockout in macrophages results in increased body weight. Further, overexpression of Noc4l using lentiviral system rescues the insulin resistance and adipose tissue weights, however, it's unclear whether it also rescued total body weight of these mice? Mechanistically, they show that macrophage-specific knockout of Noc4l improves glucose processing, insulin resistance and diet-induced obesity in response to HFD. Further, they suggest that Noc4l interacts with TLR4 to alter endocytosis pathways in macrophages to reduce inflammatory cytokine gene expression. However, the evidence for macrophage-only Noc4l driving diet-induced obesity and also the interactions of Noc4l-TLR4 in macrophage endocytosis is weak. In my opinion, the authors need to provide additional experimental evidence to support their claims.

The authors have addressed some of the reviewers' comments and the manuscript overall does read better, however, there are still some major concerns that are outlined below:

1. Tregs are also known to play an important role in insulin resistance during diet-induced obesity. In fact, the decrease in Noc4l expression in SVF (i.e. most immune cells in the adipose such as iNKTs, Tregs, B cells, macrophages etc) or adipose tissue during HFD-feeding (~7-8 weeks onwards) coincides with the expected decreased in Tregs. Thus, its plausible that the decrease in Noc4l expression may be due to the decrease in Tregs itself or other inflammatory cells such as iNKTs (another cell type that is reduced during HFD feeding). Further, given the role of Noc4l in Tregs that is already know, it would be important to clarify the role of Noc4l in Tregs and other

inflammatory cells in the adipose, not just ATMs, during diet-induced obesity. The characterisation of these immune cells in the adipose tissue can be easily performed using flow cytometry sorting and/or bead-isolation kits.

Response: We thank for the Reviewer 4's reminding. Indeed, Tregs and other immune cells do play the important roles in the process of insulin resistance during diet-induced obesity, but the key point of our story is focusing on the role of Noc4l in macrophages, Tregs and other immune cells definitely outside the scope of the current study. As Noc4l may play a critical role in Tregs in diet-induced insulin resistance (Zhu et al., 2019), we will explore it in our future study.

2. In a related note, the authors need to verify that Noc4l is indeed knocked down in adipose tissue macrophages (ATMs), not just BMDMs, from their Noc4lKO mice, especially considering that do they do observe both systemic and local inflammatory changes in these mice. Again, these experiments can be performed by isolating F4/80+ macrophages using flow cytometry sorting or bead-isolation kits, and standard qPCR and western blotting. It's quite possible that the BMDMs may be driving systemic inflammation that may have an effect on local tissues or vice versa. Please note, in the response to reviewers' (to reviewer #3, pt.4), the authors claim to have measured Noc4l in ATMs but in the figure they refer to the figure legend states BMDMs.

Also, importantly, in Fig 4b, the authors should measure the inflammatory gene expression in adipose tissue macrophages (ATMs; and/or other inflammatory cells) specifically to verify that indeed macrophage-specific KD of Noc4l is directly affecting inflammatory gene expression within these cells. As it's also possible that an increase in number of macrophage recruitment into the adipose tissue (which is indicated by CD68 and F4/80 increased gene expression) is the sole reason for the difference in inflammatory gene expression in adipose tissue.

Response: We thank to the Reviewer 4's comment. We are sorry for ambiguous description in the manuscript, which may lead the misunderstanding of the manuscript. We tested the knockout efficiency of Noc4l in ATM, besides BMDMs. The expression of Noc4l was also decreased in ATMs of Noc4l^{LKO} which was shown in **Extended Data Fig.1f and line 146-147** in the revised manuscript.

In terms of the inflammatory genes expression of ATMs, we also tested and supplemented this result. Please see the answers in point of Reviewer 2.

3. The data showing the interaction between Noc4l and TLR4, and the role of Noc4l in endocytosis needs more experimental evidence. The authors have demonstrated interaction between TLR4 and Noc4l using IP expts in macrophages. However, the signal for Noc4l is WT BMDMs (Fig 6c) is so faint (relative to input), suggesting only a tiny fraction of Noc4l actually interacts with TLR4 under resting conditions This poses the question of the significance of such

interaction etc. Thus, further experimentation is required to verify this mechanism, and some initial starting points could include the following:

Response: We thank to the Reviewer 4's comment. The importance of weak protein-protein interactions has been increasingly recognized. Weak protein-protein interactions become biologically important and play the crucial role in metabolic, regulatory, and signaling pathways (Qin and Gronenborn, 2014; Sukenik et al., 2017). In this study, we found that although the interaction between Noc4l and TLR4 is weak in the resting state of the cells, the interaction between Noc4l and TLR4 was strongly enhanced once the Raw264.7 cells were stimulated with LPS. This data is presented in the **Fig. 6d and line 356-357 labeled red** in the revised manuscript. Moreover, the results from the endogenous and exogenous IP and PLA assay have confirmed the interaction between Noc4l and TLR4. Therefore, we concluded that Noc4l regulates TLR4 signaling pathway by directly interaction with TLR4.

a. The authors should determine whether they can co-IP TLR4 and Noc4l from PA- (and LPS-) stimulated BMDMs to determine whether stimulation promotes or interferes with this interaction.

Response: We thank to the Reviewer 4's comment. We carried out the experiment for the interaction between TLR4 and NOC4L in RAW264.7 stimulated with LPS by co-IP, and observed that LPS promoted the interaction between TLR4 and NOC4L. The data is presented in the **Fig. 6d and line 356-357 labeled red** in the revised manuscript.

b. To confirm localisation of Noc4l (and TLR4) to endosomes, sub-cellular fractionation of early and late endosomes, followed by western blotting. These experiments can also be performed in PA-stimulated samples. It is good practice to validate these pathways using multiple approaches.

Response: Even we have tried many times in last months, unfortunately we didn't get the high purity of the sub-cellular fractionations of the early and late endosomes from the RAW264.7 cells. Instead, we have carried out the immunofluorescence co-localization assays at both endogenous and exogenous levels using early and late endosomes specific markers . Fluorescence colocalization analysis is more frequently used to determine whether two molecules associate with the same structures (Dunn et al., 2011), which is commonly accepted and routinely performed test in the academic field. Therefore, for identifying the localization of these two proteins, we carried out immunofluorescence co-localization assays exogenously and endogenously. The results demonstrated that Noc4l was located in the early endosome of macrophages by immunofluorescence experiments and were presented in **Figs. 5l and 5m** in the revised manuscript.

c. Fig 5: Were any of the downstream inflammatory signalling molecules (e.g. IKK or NF- κ B or their phosphorylation) affected by NOCL4LKO? This would further support the author's hypothesis that Noc4l affects inflammatory signalling.

Response: We tested the expression of I κ B α after LPS treatment. The data is presented in the **Fig. 5a** in the revised manuscript. The protein level of I κ B α was not affected by Noc4l deletion in BMDMs. As I κ B α is the downstream target of IKK, once cells receive various kinds of intracellular and extracellular stimuli, IKK is activated, which leads to I κ B α phosphorylation and ubiquitination. Then I κ B α protein is degraded and NF- κ B dimer is released and transferred to the nucleus of the cell (Liu et al., 2017). In the nucleus, NF- κ B binds to the target gene to promote the transcription of the target genes. Thus, we confirmed that the downstream inflammatory signaling molecules of NF- κ B were not affected by Noc4l deletion. Instead, we proved that Noc4l affects inflammatory cytokines expression by TRIF/IRF3 signaling.

d. As this is the first time that the authors are showing an interaction between TLR4-Noc4l, it would indeed be good to use another protein EGFP fusion protein (e.g. TLR2 or another protein or a mutated/truncated TLR4) to show the absence of such interaction (i.e. it's not purely driven by an over-expression system but indeed specific interaction).

e. Also, the above experiments can be performed in BMDMs derived from Noc4lKO mice. For example, to further confirm TLR4 localisation and/or endocytosis. Further, the opposite (potentially) interactions could be shown in OE of Noc4l in BMDMs using lentiviral system (as authors have performed in the manuscript).

Response to d and e: We thank to the Reviewer 4's comment. To prove the specific interaction between TLR4-Noc4l, we performed experiments exogenously and endogenously, respectively. First, we overexpressed FLAG-NOC4L and TLR4-EGFP in Hela cells and presented the interaction between NOC4L and TLR4 by IP (**Fig. 6a**). Second, in order to exclude the tag-induced artificial result, we further tested the endogenous interaction between NOC4L and TLR4 in RAW264.7 cells (**Fig. 6b**). Third, proximity Ligation Assay (PLA) is a powerful tool that allows in situ detection of endogenous proteins, protein modifications, and protein interactions with high specificity and sensitivity (Greenwood et al., 2015; Soderberg et al., 2006). The result of PLA further showed the specific interaction between TLR4-Noc4l in **Figs. 6e-6g**. Fourth, we demonstrated that no interaction between TLR4-Noc4l in BMDMs derived from Noc4l^{LKO} and TLR4 mutant mice, respectively in **Figs. 6c,6e and 6g**. Taken together, all these results indicated that Noc4l can specifically interact with TLR4.

f. In vivo, TLR4 surface expression can be verified from ATMs isolated from HFD-fed Noc4lKO mice and their control counterparts.

Response: We thank to the Reviewer 4's comment. Due to Covid-19 pandemic in the whole world from the beginning of 2020, all public laboratory platforms of our university and some other institutes were almost shut down until now, this task in vivo could not be accomplished upon this moment, unfortunately. Instead, we used bead-isolation kits as the reviewer suggested to recruit adipose tissue macrophages and test inflammatory genes expression. Consistent with the results from BMDMs, the expressions of *TNF α* and *IL-6* in *Noc41^{LKO}* mice were significantly higher than that in WT mice, and the results are presented in the revised manuscript **Fig.4i and line 231-232 labeled red**.

g. The larger question remains as to how macrophage TLR4-dependent inflammation in *Noc41LKO* mice directly results in adipose tissue expansion and insulin resistance. This remains unclear.

Response: The toll-like receptor 4 (TLR4) signaling pathway is one of the main triggers of the obesity-induced low-grade chronic inflammatory response (Rogero and Calder, 2018). Overweight and obese people showed increased expressions of TLR2 and TLR4 in adipose tissue in comparison with the lean people (Dasu et al., 2010). The TLR4 pathway increases the expressions of pro-inflammatory cytokines, such as *TNF α* , *IL-1*, and *IL-6*. Such increased proinflammatory cytokines during obesity, particularly *TNF α* , enhance the inhibitory effects of the serine phosphorylation of IRS-1, which in turn disrupting the insulin signal transduction and inducing insulin resistance (Guo, 2014; Hotamisligil and Davis, 2016).

In this study, we found that macrophage specific *Noc41* deficiency promotes TLR4 signaling and proinflammatory cytokines expression, which might inhibit the phosphorylation of IRS-1 resulting in insulin resistance in mice. In addition, the IRS-1 phosphorylation is under the investigation in our lab, and the results will be presented in the following manuscript.

4. The authors show that in response to HFD the *Noc41LKO* mice are metabolically challenged, where they have a decrease in VO_2 , VCO_2 , and energy expenditure (EE) during the night cycle (Fig 3). In their response to the reviewers, they mention that these mice also had a decrease in food consumption; however, they have not shown this data. It does make sense that if the mice do consume less food, they are likely to have decreased energy usage. However, the mechanisms by which these mice have increased adipose tissue and body weights are still unclear, and raise a very important question. Were there any differences in their excretion, especially fatty acids/lipids etc, as this could also potentially account for differences in their food absorption abilities and potential weight gain? Or potentially *Noc41* deletion could also affect their appetite? Were there any weight differences in other organs (e.g. liver, spleen, muscle)? It's unlike that insulin resistance accounts for the body weight gain? This data, although interesting, is quite confusing, and further

experiments are required to decipher the role of Noc4l in increasing adipose tissue and body weights.

Response: Thanks Reviewer 4 for these good suggestions. As we all know, obesity is associated with a chronic, low-grade inflammatory state, reflected by infiltration and activation of cells of the immune system in adipose tissue, as well as a systemically greater abundance of proinflammatory cytokines (Esser et al., 2014; Saltiel and Olefsky, 2017). Those cytokines in turn activate intracellular stress-signaling cascades associated with the transcription factor NF- κ B and the kinase JNK, which subsequently inhibit the action of insulin in the target tissues such as adipose tissue, liver, skeletal muscle and even the central nervous system (Guo, 2014; Hotamisligil and Davis, 2016). Prolonged inhibition of insulin signaling eventually leads to the development of insulin resistance, and ultimately, progression to overt type 2 diabetes mellitus (Donath and Shoelson, 2011; Mauer et al., 2014).

In this study, our key point is under the condition of HFD-induced obesity, Noc4l deficiency in macrophage significantly enhance inflammatory cytokines in adipose tissues which further promotes insulin resistance. As mentioned above, prolonged insulin resistance exacerbates HFD-induced obesity.

In the future, we will generate the neuron-specific ablation of Noc4l to further study whether Noc4l affects mice appetite. Additionally, we are also very interested in the role of gut microbiota in Noc4l increasing adipose tissue and body weights, which will be also under the investigation in future studies.

In addition, the weight of liver and spleen showed no difference between Noc4l^{LKO} and Noc4l^{fl/fl} mice. Please see the results in **Extended Data Fig.1j and line 153-154 labeled red** in the revised manuscript.

5. Some critical in vitro macrophage experiments are also missing. For example, gene expression of inflammatory cytokines is not always reflective of cytokine production and secretion. Authors should indeed confirm the secretion of IL-6, TNF α etc in response to palmitic acid (and LPS) in BMDMs or PMs. Further, the authors should verify that Noc4l KD or OE does not impact naive macrophages ability to become anti-inflammatory 'M2-like' macrophages (i.e. measure 'M2-like' markers in response to IL-4 and/or IL-13).

Response: The secretion results of IL-6, TNF α in response to LPS in BMDMs and the 'M2-like' markers in response to IL-4 had been presented in the previous manuscript. We are sorry for this ambiguous description in the manuscript, which may lead the misunderstanding of the manuscript. These results have been presented clearly in the revised **Figs.4g-4h and Fig.4j**, respectively.

Minor:

1. Some of the new data generated in response to the reviewers' comments were not included in the manuscript and would be critical to do so. For example, the measurement of Noc4l expression in CD11b+/- ATMs. Also, the patient information should be included as a summary table of number of females vs males, average age and average BMI etc.

Response: The patient information was shown in Extended Data Table 4. However, some of the new data such as the measurement of Noc4l expression in CD11b+/- ATMs are just response to the reviewers' comments, which are not relevant to the current study, or somehow even diffuse the focus of the current study. In this way, we excluded these data from the current manuscript.

2. Did they test their Noc4l antibodies in the knockout cells or tissues from mice to verify specificity of the antibody.

Response: The specificity of the antibody was tested in the knockout BMDMs and the results were shown in the **Extended Data Figs. 1b and 1d-1e**. The results indicated that the Noc4l antibody is specific.

3. The authors should verify that Noc4l was not knockdown in other cells (e.g. T cells etc) in the Noc4l^{LKO} mice – i.e. there's no leak with the LysM cre system.

Response: In 1999, Clausen and colleagues generated the LysM-Cre mouse and confirmed that no depletion was detected in lymphoid cells such as B and T lymphocytes (Clausen et al., 1999). T cells do play the important roles in the process of metabolism. In our study, we also compared the Noc4l expression in T cells between Noc4l^{fl/fl} and Noc4l^{LKO} mice and the results showed Noc4l expression showed no difference. The result of T cells is not relevant to the current study. So we didn't put the results in the manuscript.

4. The data with HFD-fed OE mice in ITT (insulin sensitivity) is also confusing. The authors should discuss the limitations and explain this data. It may be worthwhile to perform ITT at a different time point (e.g. 10-12wks) to see if they do observe a difference. Overall, the authors should discuss the differences between OE and KO data and at least postulate the reasons why they do observe these differences in vitro and in vivo.

Response: We thank to the Reviewer 4's comment. As mentioned above, the key point of our study is the role of inflammation in obesity-induced insulin resistance. However, in contrast to the macrophage specific Noc4l deficiency mice, no obvious difference of inflammatory cytokines expression in Noc4l macrophage-overexpressed mice compared to WT mice was observed

(Extended Data Fig. 4g). Therefore, we verified that insulin sensitivity is similar between WT and OE mice. We have discussed this issue in the DISCUSSION part of revised manuscript. Please see **line 429-435 labeled red.**

5. It would greatly assist the reviewers if the authors tracked changes or highlighted the changes in the actual manuscript file. Also, having the manuscript in portrait format would make it easier to read (rather than landscape).

Response: We have accepted the suggestions of reviewer 4 and highlighted the changes in the actual manuscript file.

6. New Fig 5a – it would be better to show a lighter and darker exposure of p-ERK as it's hard to see the bands in the current blot.

Response: We have replaced the p-ERK with a clear image. Please see **Fig. 5a in revised manuscript.**

7. Lastly, there remain some minor grammatical and typographical issues with the current submission.

Response: We have checked through the grammatical and typographical issues with the current submission.

References:

Cani, P.D., Amar, J., Iglesias, M.A., Poggi, M., Knauf, C., Bastelica, D., Neyrinck, A.M., Fava, F., Tuohy, K.M., Chabo, C., *et al.* (2007). Metabolic endotoxemia initiates obesity and insulin resistance. *Diabetes* 56, 1761-1772.

Cao, J., Peng, J., An, H., He, Q., Boronina, T., Guo, S., White, M.F., Cole, P.A., and He, L. (2017). Endotoxemia-mediated activation of acetyltransferase P300 impairs insulin signaling in obesity. *Nat Commun* 8, 131.

Clausen, B.E., Burkhardt, C., Reith, W., Renkawitz, R., and Forster, I. (1999). Conditional gene targeting in macrophages and granulocytes using LysMcre mice. *Transgenic Res* 8, 265-277.

Dasu, M.R., Devaraj, S., Park, S., and Jialal, I. (2010). Increased toll-like receptor (TLR) activation and TLR ligands in recently diagnosed type 2 diabetic subjects. *Diabetes Care* 33, 861-868.

Donath, M.Y., and Shoelson, S.E. (2011). Type 2 diabetes as an inflammatory disease. *Nat Rev*

Immunol 11, 98-107.

Dunn, K.W., Kamocka, M.M., and McDonald, J.H. (2011). A practical guide to evaluating colocalization in biological microscopy. *Am J Physiol Cell Physiol* 300, C723-742.

Esser, N., Legrand-Poels, S., Piette, J., Scheen, A.J., and Paquot, N. (2014). Inflammation as a link between obesity, metabolic syndrome and type 2 diabetes. *Diabetes Res Clin Pract* 105, 141-150.

Greenwood, C., Ruff, D., Kirvell, S., Johnson, G., Dhillon, H.S., and Bustin, S.A. (2015). Proximity assays for sensitive quantification of proteins. *Biomol Detect Quantif* 4, 10-16.

Guo, S. (2014). Insulin signaling, resistance, and the metabolic syndrome: insights from mouse models into disease mechanisms. *J Endocrinol* 220, T1-T23.

Hill, D.A., Lim, H.W., Kim, Y.H., Ho, W.Y., Foong, Y.H., Nelson, V.L., Nguyen, H.C.B., Chegireddy, K., Kim, J., Habberheuer, A., *et al.* (2018). Distinct macrophage populations direct inflammatory versus physiological changes in adipose tissue. *Proc Natl Acad Sci U S A* 115, E5096-E5105.

Hotamisligil, G.S., and Davis, R.J. (2016). Cell Signaling and Stress Responses. *Cold Spring Harb Perspect Biol* 8.

Konner, A.C., and Bruning, J.C. (2011). Toll-like receptors: linking inflammation to metabolism. *Trends Endocrinol Metab* 22, 16-23.

Liu, T., Zhang, L., Joo, D., and Sun, S.C. (2017). NF-kappaB signaling in inflammation. *Signal Transduct Target Ther* 2.

Mauer, J., Chaurasia, B., Goldau, J., Vogt, M.C., Ruud, J., Nguyen, K.D., Theurich, S., Hausen, A.C., Schmitz, J., Bronneke, H.S., *et al.* (2014). Signaling by IL-6 promotes alternative activation of macrophages to limit endotoxemia and obesity-associated resistance to insulin. *Nat Immunol* 15, 423-430.

Qin, J., and Gronenborn, A.M. (2014). Weak protein complexes: challenging to study but essential for life. *FEBS J* 281, 1948-1949.

Rogero, M.M., and Calder, P.C. (2018). Obesity, Inflammation, Toll-Like Receptor 4 and Fatty Acids. *Nutrients* 10.

Saltiel, A.R., and Olefsky, J.M. (2017). Inflammatory mechanisms linking obesity and metabolic disease. *J Clin Invest* 127, 1-4.

Shi, H., Kokoeva, M.V., Inouye, K., Tzameli, I., Yin, H., and Flier, J.S. (2006). TLR4 links innate immunity and fatty acid-induced insulin resistance. *J Clin Invest* *116*, 3015-3025.

Soderberg, O., Gullberg, M., Jarvius, M., Ridderstrale, K., Leuchowius, K.J., Jarvius, J., Wester, K., Hydbring, P., Bahram, F., Larsson, L.G., *et al.* (2006). Direct observation of individual endogenous protein complexes in situ by proximity ligation. *Nat Methods* *3*, 995-1000.

Sukenik, S., Ren, P., and Gruebele, M. (2017). Weak protein-protein interactions in live cells are quantified by cell-volume modulation. *Proc Natl Acad Sci U S A* *114*, 6776-6781.

Weisberg, S.P., Hunter, D., Huber, R., Lemieux, J., Slaymaker, S., Vaddi, K., Charo, I., Leibel, R.L., and Ferrante, A.W., Jr. (2006). CCR2 modulates inflammatory and metabolic effects of high-fat feeding. *J Clin Invest* *116*, 115-124.

Winer, D.A., Luck, H., Tsai, S., and Winer, S. (2016). The Intestinal Immune System in Obesity and Insulin Resistance. *Cell Metab* *23*, 413-426.

Zhu, X., Zhang, W., Guo, J., Zhang, X., Li, L., Wang, T., Yan, J., Zhang, F., Hou, B., Gao, N., *et al.* (2019). Noc4L-Mediated Ribosome Biogenesis Controls Activation of Regulatory and Conventional T Cells. *Cell Rep* *27*, 1205-1220 e1204.

Reviewer comments, third round –

Reviewer #2 (Remarks to the Author):

The authors have addressed my outstanding concerns. I would recommend for them to provide more details into the figure legend and the text to clarify i) how the experiment with ATMs was performed and ii) provide the reader with an understanding of what the bars in the figure represent.

Reviewer #3 (Remarks to the Author):

The authors have responded adequately to most of the comments. However, one issue still requires attention. As the authors have used LPS in their model and describe this as the main driver of obesity-induced inflammation, it would be of importance to measure endotoxin levels in their model. This is also relevant in light of recent studies showing the presence of a metabolic activated macrophage residing in obese adipose tissue that is different compared to the classically activated macrophage using LPS/IFN γ (reference: *Cell Metab.* 2014 Oct 7;20(4):614-25).

Reviewer #4 (Remarks to the Author):

The authors have addressed most of the revisions requested, but have glossed over several comments. Some critical information remains missing.

The authors have been asked to have measured secreted IL-6, TNF α in response to LPS in BMDMs but the authors have not presented that data in this manuscript, citing previous manuscript and they show relative expression data in Fig 4. The authors need to show secreted levels in this model, and if indeed, it was previously published, that reduces the novelty and impact of current study. Also, if indeed, ATMs are secreting these pro-inflammatory cytokines, the authors should culture extracted adipose tissue or isolated ATMs in vitro to demonstrate secreted protein, not just relative gene expression.

hhh

It also remains unclear as whether the inflammation occurs in other tissues (e.g. liver; Kupper cells) and that may be the mechanism for insulin resistance in this model. Thus, this should also be checked.

To provide mechanistic evidence, the authors were requested to test whether the phosphorylation (or ubiquitinated) of inflammatory molecules (e.g. IKK, NF- κ B) were affected in response to LPS with Noc4l deletion (or overexpression). However, they seemed to have measured expression, which doesn't reflect activity of these molecules. Overall, mechanistic insights into the pathway remains to be speculative presently, and I would argue more is required.

Did the authors measure whether Noc41 deletion affect IRS-1 phosphorylation?

If indeed Noc4l mice only affect HFD-diet induced obesity, then how do the authors explain the metabolic challenges these mice display (i.e. VO₂, VC0₂, EE)? The authors need to, at minimum, discuss this interesting but confusing data in the discussion. Do they postulate inflammation is causing these metabolic challenges?

Reviewer #2 (Remarks to the Author):

The authors have addressed my outstanding concerns. I would recommend for them to provide more details into the figure legend and the text to clarify i) how the experiment with ATMs was performed and ii) provide the reader with an understanding of what the bars in the figure represent.

Response: Thanks for Reviewer 2's suggestion. We have provided more details about ATM experiment in both Materials and Methods and the figure legend in Fig.4i. Please see the red labeled **line 579-586 and line 1096-1099** in the revised manuscript .

Reviewer #3 (Remarks to the Author):

1) The authors have responded adequately to most of the comments. However, one issue still requires attention. As the authors have used LPS in their model and describe this as the main driver of obesity-induced inflammation, it would be of importance to measure endotoxin levels in their model. **2)** This is also relevant in light of recent studies showing the presence of a metabolic activated macrophage residing in obese adipose tissue that is different compared to the classically activated macrophage using LPS/IFN γ (reference: Cell Metab. 2014 Oct 7;20(4):614-25).

Response: 1) We thank to the Reviewer 3's comment. Concerning of the endotoxin levels, several points need to be explained:

a) Most studies including ours didn't make use of the germ-free mice, therefore LPS (from Gram-negative bacteria) derived from gut microbiota must play a non-ignorable role to the host's metabolic phenotype. **b)** The plasma LPS levels rise with higher fat intake in mice and humans is commonly accepted in the academic field (Amar et al., 2008; Cani et al., 2007; Cao et al., 2017; Creely et al., 2007; Tremaroli and Backhed, 2012). Experimentally, plasma LPS level was increased in HFD-fed mice (Cao et al., 2017; Laurans et al., 2018) ; Clinically, plasma LPS level was increased in patients with type 2 diabetes (Tilg and Moschen, 2014) . **c)** According to our results, we demonstrated that Noc4l directly binds to TLR4, therefore we focused on how Noc4l elicits its action on TLR4 signal pathway in this study. We found that Noc4l deficiency in macrophage promotes M1-like macrophage activation leading to HFD-induced inflammation by enhanced LPS-induced TLR4/TRIF signaling under obesity and diabetes conditions. Based on these points, we think to measure the endotoxin levels in our model does not give additional information for our study.

2) Adipose tissue macrophages (ATMs) are a heterogeneous population of immune cells with a highly plastic phenotype according to microenvironment. Macrophages exhibit a diverse spectrum of metabolic characteristics. In general, ATMs include alternatively activated M2-like macrophages and classically activated M1-like macrophages. M1-like and M2-like represent the proinflammatory state of recruited ATMs and the anti-inflammatory state of resident ATMs, respectively. Interestingly, Kratz et al. identified

a distinct population of metabolically activated macrophages (MMe), following palmitate, glucose and insulin challenge which was distinct from bacterially activated M1 macrophages (Kratz et al., 2014), their work opens a novel window for the metabolic study.

In our study, we also measured the effect of Noc4l on MMe after the treatment of 500 μ M palmitate for 24h. The result showed that proinflammatory cytokines expressions were increased in BMDMs of KO mice compared to WT mice. These findings may suggest the role of Noc4l in regulating MMe functions during obesity. However, the mechanism of how Noc4l regulating MMe needs to be studied further in our future work. Meanwhile, we also addressed this in the discussion section of the revised manuscript in the red labeled **line 417-428**.

Reviewer #4 (Remarks to the Author):

The authors have addressed most of the revisions requested, but have glossed over several comments. Some critical information remains missing.

1) The authors have been asked to have measured secreted IL-6, TNF α in response to LPS in BMDMs but the authors have not presented that data in this manuscript, citing previous manuscript and they show relative expression data in Fig 4. The authors need to show secreted levels in this model, and if indeed, it was previously published, that reduces the novelty and impact of current study.

Response: We are sorry for the ambiguous description in the manuscript, which may lead the misunderstanding of the manuscript. We did measure the levels of the secreted IL-6, TNF α in response to LPS in BMDMs and the results were showed in **Fig.4g and 4h** in the previous manuscript. These results indicated that Noc4l deficiency in macrophage promotes inflammatory cytokines expression in BMDMs. Additionally, all these results are not published elsewhere.

2) Also, if indeed, ATMs are secreting these pro-inflammatory cytokines, the authors

should culture extracted adipose tissue or isolated ATMs in vitro to demonstrate secreted protein, not just relative gene expression.

Response: We thank to the Reviewer 4's good suggestions.

We are sorry that there are some technical problems and limits to measure the secreting pro-inflammatory cytokines from the isolated ATM of the Noc41^{fl/fl} and Noc41^{LKO} mice:

a) ATMs were isolated from the Noc41 mice after HFD-fed for 16w. The number of the isolated ATMs was quite little. b) As we all know that the isolated primary ATMs have no proliferation capability. If we culture the isolated ATMs in vitro for a short time (for example after attached to the plates), it was hard to detect the proinflammatory cytokines in the supernatant due to the detective limits of the current commercial ELISA kit. c) If the isolated ATMs were keep for a long term culture in vitro, their characteristics will be changed by the external culture stress which is not the suitable way to display the functions of the HFD-treated ATMs in vivo.

Based on these technical limits, after the isolation of ATMs, we directly measured the proinflammatory cytokines expression of the purified RNA from the isolated ATMs of the HFD-treated mice, which is not the best but rather the suitable way to display the ATMs function in vivo, alternatively. Additionally, we measured the proinflammatory cytokines in the sera of the HFD-treated Noc41^{fl/fl} and Noc41^{LKO} mice, which showed that Noc41 deficiency exaggerated the systemic inflammation (please see **Fig. 4a**). Lastly, due to the above technical limits, to measure the proinflammatory cytokines expressions from ATMs by qPCR is the commonly accepted and routinely performed test in the academic field (Han et al., 2013; Hernandez et al., 2014; Keiran et al., 2019; Ramkhelawon et al., 2014; Shin et al., 2017).

3) It also remains unclear as whether the inflammation occurs in other tissues (e.g. liver; Kupper cells) and that may be the mechanism for insulin resistance in this model. Thus, this should also be checked.

Response: We are sorry for our ambiguous description in the manuscript, which may lead the misunderstanding of the reader. We had measured the expressions of proinflammatory cytokines in livers in the previous figure (**Fig.4b**). The result showed that the expressions of both proinflammatory cytokines and macrophage markers had no differences between Noc41^{fl/fl} and Noc41^{LKO} mice. These results indicated that inflammation in Kupffer cells may not be the critical mechanism for insulin resistance.

4) To provide mechanistic evidence, the authors were requested to test whether the phosphorylation (or ubiquitinated) of inflammatory molecules (e.g. IKK, NF-κB) were affected in response to LPS with Noc4l deletion (or overexpression). However, they seemed to have measured expression, which doesn't reflect activity of these molecules. Overall, mechanistic insights into the pathway remains to be speculative presently, and I would argue more is required.

Response: It is well known that LPS triggers innate immune responses via TLR4, which contains two major pathways: the MyD88-dependent and MyD88-independent pathways (TRIF-dependent pathway). The key point of our story is Noc4l inhibits inflammatory cytokines expression by TRIF/IRF3 (MyD88-independent pathway) signaling to regulate insulin resistance. Moreover, we also demonstrated that Noc4l has no effect of MyD88-dependent signaling, thus downstream inflammatory signaling molecules of NF-κB were not affected by Noc4l deletion. Please see **Fig.7b** as follows:

5) Did the authors measure whether Noc4l deletion affect IRS-1 phosphorylation?

Response: The three major biochemical steps in insulin signaling are: tyrosine phosphorylation of the receptor and its direct substrates; activation of the lipid kinase, PI3K; and activation of multiple serine/threonine kinases, the most important of which is AKT (Haeusler et al., 2018). Therefore, AKT phosphorylation rather than IRS-1 phosphorylation is a direct readout of intracellular insulin signaling and has been measured in most of studies about insulin resistance to indicate the ability of insulin responses (Fan et al., 2016; Han et al., 2013; Hernandez et al., 2014; Keiran et al., 2019).

In our study, to confirm Noc4l deletion in macrophage impaired insulin signaling we also measured AKT phosphorylation after insulin injection, and the results showed that AKT phosphorylation was reduced in insulin target tissues including adipose tissue, muscle and liver. In this study, the phosphorylation of IRS-1 was not the mainly point which we focused. Whether Noc4l deletion impaired the AKT phosphorylation by inhibiting IRS-1 phosphorylation is under the investigation in our lab, and the results will be presented in the following manuscript.

6) If indeed Noc4l mice only affect HFD-diet induced obesity, then how do the authors explain the metabolic challenges these mice display (i.e. VO₂, VC₀₂, EE)? The authors need to, at minimum, discuss this interesting but confusing data in the discussion. Do they postulate inflammation is causing these metabolic challenges?

Response: We thank to reviewer 4's comment. We are sorry for our ambiguous description in the manuscript, which may lead the misunderstanding of the manuscript. The HFD-fed Noc4l^{LKO} mice showed a significant increase in total body weight compared to WT control. To understand the cause of increased adiposity in Noc4l^{LKO} mice, we analyzed energetic metabolism by indirect calorimetry. This result of VO₂, VCO₂, energy expenditure (EE) indicated that Noc4l deficiency reduced energy expenditure to cause increased weight gain upon HFD through a non-activity-based mechanism. It is known that diet can induce thermogenesis to keep energy balance (Kazak et al., 2017). Previous studies reported that alternatively activated macrophages express tyrosine hydroxylase (TH) and produce catecholamines to regulate HFD-induced obesity (Qiu et al., 2014). However, some studies indicated that macrophages isolated from adipose tissue of cold challenged mice did not express TH (Fischer et al., 2017; Spadaro et al., 2017). Therefore, we speculated that Noc4l deletion in macrophage may inhibit HFD-induced thermogenesis in the white adipose tissues to suppress energy expenditure. However, the mechanism of how Noc4l regulating energy balance needs to be studied further in our future work. Meanwhile, we also discussed this in the discussion section of the revised manuscript in the red labeled **line 447-455**.

References:

- Amar, J., Burcelin, R., Ruidavets, J.B., Cani, P.D., Fauvel, J., Alessi, M.C., Chamontin, B., and Ferrieres, J. (2008). Energy intake is associated with endotoxemia in apparently healthy men. *Am J Clin Nutr* 87, 1219-1223.
- Cani, P.D., Amar, J., Iglesias, M.A., Poggi, M., Knauf, C., Bastelica, D., Neyrinck, A.M., Fava, F., Tuohy, K.M., Chabo, C., *et al.* (2007). Metabolic endotoxemia initiates obesity and insulin resistance. *Diabetes* 56, 1761-1772.
- Cao, J., Peng, J., An, H., He, Q., Boronina, T., Guo, S., White, M.F., Cole, P.A., and He, L. (2017). Endotoxemia-mediated activation of acetyltransferase P300 impairs insulin signaling in obesity. *Nat Commun* 8, 131.
- Creely, S.J., McTernan, P.G., Kusminski, C.M., Fisher, M., Da Silva, N.F., Khanolkar, M., Evans, M., Harte, A.L., and Kumar, S. (2007). Lipopolysaccharide activates an innate immune system response in human adipose tissue in obesity and type 2 diabetes. *Am J Physiol Endocrinol Metab* 292, E740-747.
- Fan, R., Toubal, A., Goni, S., Drareni, K., Huang, Z., Alzaid, F., Ballaire, R., Ancel, P., Liang, N., Damdimopoulos, A., *et al.* (2016). Loss of the co-repressor GPS2 sensitizes macrophage activation upon metabolic stress induced by obesity and type 2 diabetes. *Nat Med* 22, 780-791.
- Fischer, K., Ruiz, H.H., Jhun, K., Finan, B., Oberlin, D.J., van der Heide, V., Kalinovich, A.V., Petrovic, N., Wolf, Y., Clemmensen, C., *et al.* (2017). Alternatively activated macrophages do not synthesize catecholamines or contribute to adipose tissue adaptive thermogenesis. *Nat Med* 23, 623-630.
- Haeusler, R.A., McGraw, T.E., and Accili, D. (2018). Biochemical and cellular properties of insulin receptor signalling. *Nat Rev Mol Cell Biol* 19, 31-44.
- Han, M.S., Jung, D.Y., Morel, C., Lakhani, S.A., Kim, J.K., Flavell, R.A., and Davis, R.J. (2013). JNK expression by macrophages promotes obesity-induced insulin resistance and inflammation. *Science* 339, 218-222.
- Hernandez, E.D., Lee, S.J., Kim, J.Y., Duran, A., Linares, J.F., Yajima, T., Muller, T.D., Tschop, M.H., Smith, S.R., Diaz-Meco, M.T., *et al.* (2014). A macrophage NBR1-MEKK3 complex triggers JNK-mediated adipose tissue inflammation in obesity. *Cell Metab* 20, 499-511.
- Kazak, L., Chouchani, E.T., Lu, G.Z., Jedrychowski, M.P., Bare, C.J., Mina, A.I., Kumari, M., Zhang, S., Vuckovic, I., Laznik-Bogoslavski, D., *et al.* (2017). Genetic Depletion of Adipocyte Creatine Metabolism Inhibits Diet-Induced Thermogenesis and Drives Obesity. *Cell Metab* 26, 693.
- Keiran, N., Ceperuelo-Mallafre, V., Calvo, E., Hernandez-Alvarez, M.I., Ejarque, M., Nunez-Roa, C., Horrillo, D., Maymo-Masip, E., Rodriguez, M.M., Fradera, R., *et al.* (2019). SUCNR1 controls an anti-inflammatory program in macrophages to regulate the metabolic response to obesity. *Nat Immunol* 20, 581-592.
- Kratz, M., Coats, B.R., Hisert, K.B., Hagman, D., Mutskov, V., Peris, E., Schoenfelt, K.Q., Kuzma, J.N., Larson, I., Billing, P.S., *et al.* (2014). Metabolic dysfunction drives a mechanistically distinct proinflammatory phenotype in adipose tissue macrophages. *Cell Metab* 20, 614-625.
- Laurans, L., Venteclef, N., Haddad, Y., Chajadine, M., Alzaid, F., Metghalchi, S., Sovran, B., Denis, R.G.P., Dairou, J., Cardellini, M., *et al.* (2018). Genetic deficiency of indoleamine 2,3-dioxygenase promotes gut microbiota-mediated metabolic health. *Nat Med* 24, 1113-1120.
- Qiu, Y., Nguyen, K.D., Odegaard, J.I., Cui, X., Tian, X., Locksley, R.M., Palmiter, R.D., and Chawla, A. (2014). Eosinophils and type 2 cytokine signaling in macrophages orchestrate development of

functional beige fat. *Cell* 157, 1292-1308.

Ramkhelawon, B., Hennessy, E.J., Menager, M., Ray, T.D., Sheedy, F.J., Hutchison, S., Wanschel, A., Oldebeken, S., Geoffrion, M., Spiro, W., *et al.* (2014). Netrin-1 promotes adipose tissue macrophage retention and insulin resistance in obesity. *Nat Med* 20, 377-384.

Shin, K.C., Hwang, I., Choe, S.S., Park, J., Ji, Y., Kim, J.I., Lee, G.Y., Choi, S.H., Ching, J., Kovalik, J.P., *et al.* (2017). Macrophage VLDLR mediates obesity-induced insulin resistance with adipose tissue inflammation. *Nat Commun* 8, 1087.

Spadaro, O., Camell, C.D., Bosurgi, L., Nguyen, K.Y., Youm, Y.H., Rothlin, C.V., and Dixit, V.D. (2017). IGF1 Shapes Macrophage Activation in Response to Immunometabolic Challenge. *Cell Rep* 19, 225-234.

Tilg, H., and Moschen, A.R. (2014). Microbiota and diabetes: an evolving relationship. *Gut* 63, 1513-1521.

Tremaroli, V., and Backhed, F. (2012). Functional interactions between the gut microbiota and host metabolism. *Nature* 489, 242-249.

Reviewer comments, fourth round –

Reviewer #2 (Remarks to the Author):

The authors have adequately addressed my concerns.

Reviewer #3 (Remarks to the Author):

Since the authors have clarified a previously unidentified biological function of NOC4L, namely the direct interaction with TLR4 to inhibit its endocytosis and blocking the TRIF pathway, the authors should also provide solid evidence for the ligands leading to activation of the pathway in vivo or at least discuss this issue in more detail. Although the authors refer to previous papers, it would still be important to measure potential TLR4-agonists in their experimental setup especially in light of using these statements in the manuscript (Activation of TLR4 signaling pathways triggered by LPS plays a predominant role in obesity-induced inflammation).

The authors also refer to results obtained from the MMe macrophages, but I am unable to find the results in the current manuscript.

For the human data presented in the last figure, did the authors correlated NOC4L expression with the results of the inflammatory genes? Hence, a person with a lower NOC4L expression should have higher inflammatory gene expression levels. Currently, it is difficult to interpret the role of NOC4L in human adipose tissue based on the results presented in the manuscript.

Reviewer #5 (Remarks to the Author):

The authors adequately responded to the reviewer 4.

The authors claimed that Noc4l negatively regulates MyD88-independent LPS response by inhibiting TLR4-internalization. The authors showed that Noc4l deletion enhances MyD88-independent LPS responses such as IFN- β production. This conclusion is, however, not consistent with the finding shown in Fig. 4, in which LPS-stimulated, Noc4l-deficient BMDMs showed higher production of TNF α . If TLR4 internalization is enhanced in Noc4l-deficient BMDMs, MyD88-independent cytokines such as IFN- β are expected to increase, but not TNF α , an MyD88-dependent cytokine. The authors need to explain.

Another concern is on the findings in Fig. 5p and 5q. Previous reports 1,2 showed that Dynasore inhibits MyD88-independent responses in wild-type BM-DMs, not MyD88-dependent responses. The results in Fig. 5p and 5q showed that Dynasore had no effect on LPS-dependent production of IFN β in wild-type BMDMs. The authors should explain why wild-type BMDMs were not sensitive to Dynasore treatment. Furthermore, the authors should explain a reason why TNF α production is decreased by Dynasore treatment in Noc4l-deficient BMDMs. Because TNF α production is induced by cell surface TLR4, TNF α production is expected to decrease in Noc4l-deficient BMDMs, in which TLR4 internalization is enhanced.

1 Kagan, J. C. et al. TRAM couples endocytosis of Toll-like receptor 4 to the induction of interferon-beta. *Nat Immunol* 9, 361-368, doi:ni1569 [pii] 10.1038/ni1569 (2008).

10.1038/ni1569 (2008).

2 Motoi, Y. et al. Lipopeptides are signaled by Toll-like receptor 1, 2 and 6 in endolysosomes. *Int Immunol* 26, 563-573, doi:10.1093/intimm/dxu054 (2014).

Reviewer #3 (Remarks to the Author):

1. Since the authors have clarified a previously unidentified biological function of NOC4L, namely the direct interaction with TLR4 to inhibit its endocytosis and blocking the TRIF pathway, the authors should also provide solid evidence for the ligands leading to activation of the pathway in vivo or at least discuss this issue in more detail. Although the authors refer to previous papers, it would still be important to measure potential TLR4-agonists in their experimental setup especially in light of using these statements in the manuscript (Activation of TLR4 signaling pathways triggered by LPS plays a predominant role in obesity-induced inflammation).

Response: We thank for Reviewer 3's good suggestion. We are sorry for the ambiguous description in the manuscript, which may lead the misunderstanding of the manuscript. Studies had reported that many factors, including LPS, produced from the gut microbiota and various lipid species that are elevated owing to diet or obesity (Miao et al., 2014; Reilly and Saltiel, 2017), play a key role in obesity-associated inflammation. Actually, we measured the plasma endotoxin concentration from the CD- and HFD-fed mice by an endotoxin detection assay kit. The results showed that the plasma endotoxin concentration of HFD-fed mice is higher than that of CD-fed mice, however, due to the high variation, there is no difference between $Noc41^{fl/fl}$ and $Noc41^{LKO}$ mice (**see below, n= 4**). Additionally, in our case, glucose tolerance of $Noc41^{LKO}$ mice was significantly impaired relative to the $Noc41^{fl/fl}$ mice with CD, while the blood level of palmitate has not been elevated (Fig.3d). Furthermore, we observed that glucose tolerance of $Noc41^{LKO}$ mice was significantly impaired relative to the $Noc41^{fl/fl}$ mice with a nonlethal dose of LPS injection intraperitoneally (Fig. 4m). So, under this condition, we would rather believe that LPS plays more important roles than palmitate does in our model. Thus, we put more attention on investigating how *noc4l* regulates LPS-induced signaling. Moreover, we found that *Noc4l* direct interacts with TLR4 to inhibit its endocytosis and blocks the TRIF pathway in macrophages. Palmitate is another important factor for obesity-induced inflammation. There exists a number of mechanisms about how palmitate regulates obesity-associated inflammation (Senn, 2006; Wen et al., 2011; Yin et al., 2015). We do agree that palmitate effects the TLR4 signal under the metabolic disorders. However, recent studies showed that palmitate does not directly bind TLR4, although $Tlr4^{-/-}$ attenuates inflammation in palmitate-treated macrophages (Lancaster et al., 2018). So how *Noc4l* regulates palmitate-associated signaling still needs to be further studied. Thus, we addressed this in the discussion section of the revised manuscript in the red labeled context **line 430-438**.

2. The authors also refer to results obtained from the MMe macrophages, but I am unable to find the results in the current manuscript.

Response: Kratz et al., showed that FFA such as palmitate plays a key role in triggering metabolic activation of macrophages (Kratz et al., 2014). In our study, we measured the effect of Noc4l on BMDMs after the treatment of 500 μ M palmitate for 24h. The result showed that the expressions of the proinflammatory cytokines were increased in BMDMs of the KO mice compared to WT mice (Fig. 4k). These findings may suggest the role of Noc4l in regulating MMe functions during obesity. However, the mechanism of how Noc4l regulating MMe needs to be studied further in our future work.

3. For the human data presented in the last figure, did the authors correlated NOC4L expression with the results of the inflammatory genes? Hence, a person with a lower NOC4L expression should have higher inflammatory gene expression levels. Currently, it is difficult to interpret the role of NOC4L in human adipose tissue based on the results presented in the manuscript.

Response: We thank so much for Reviewer 3's good suggestion. We supplemented the correlation analysis between NOC4L and inflammatory genes in human adipose tissue. The results were shown in the **Extended Data Fig.6 and below**. Due to the limited number of patients, the results indicated the significant negative correlation between NOC4L and some of inflammatory genes, and some of inflammatory genes in human adipose tissue have the negative tendency.

Reviewer #5 (Remarks to the Author):

The authors adequately responded to the reviewer 4.

1. The authors claimed that Noc4l negatively regulates MyD88-independent LPS response by inhibiting TLR4-internalization. The authors showed that Noc4l deletion enhances MyD88-independent LPS responses such as IFN- β production. This conclusion is, however, not consistent with the finding shown in Fig. 4, in which LPS-stimulated, Noc4l-deficient BMDMs showed higher production of TNF α . If TLR4 internalization is enhanced in Noc4l-deficient BMDMs, MyD88-independent cytokines such as IFN- β are expected to increase, but not TNF α , an MyD88-dependent cytokine. The authors need to explain.

Response: Thanks for Reviewer 5 comment. As seen in the MyD88 pathway, NF- κ B is also activated by the TRIF-dependent pathway. In addition to IRF3 activation, the downstream pathway of TRIF leads to NF- κ B activation and induction of inflammatory cytokines. It is noteworthy that production of inflammatory cytokines is also reduced in TRIF-deficient mice (Yamamoto et al., 2003a). It has been reported that TRIF triggered the activation of NF- κ B, referred to as “the late NF- κ B” (Kawai and Akira, 2007; Yamamoto et al., 2003b). We also cited these studies in our manuscript (please see line 280). And our luciferase result confirmed the TRIF-induced NF- κ B activation, and this TRIF-induced NF- κ B activation was inhibited by NOC4L expression in a dose-dependent manner (Fig. 5f). Collectively, TRIF-dependent pathways can activate both NF- κ B and IRF3, and are required for the induction of inflammatory cytokines in TLR4 signaling (Kawai and Akira, 2007; Liu et al., 2017; Yamamoto et al., 2003b).

2. Another concern is on the findings in Fig. 5p and 5q. Previous reports 1, 2 showed

that Dynasore inhibits MyD88-independent responses in wild-type BM-DMs, not MyD88-dependent responses. The results in Fig. 5p and 5q showed that Dynasore had no effect on LPS-dependent production of IFN β in wild-type BMDMs. The authors should explain why wild-type BMDMs were not sensitive to Dynasore treatment.

1 Kagan, J. C. et al. TRAM couples endocytosis of Toll-like receptor 4 to the induction of interferon-beta. *Nat Immunol* 9, 361-368, doi:ni1569 [pii] 10.1038/ni1569 (2008).

2 Motoi, Y. et al. Lipopeptides are signaled by Toll-like receptor 1, 2 and 6 in endolysosomes. *Int Immunol* 26, 563-573, doi:10.1093/intimm/dxu054 (2014).

Response: Thanks for Reviewer 5 comments. The reason for the different results may due to the different conditions between ours and Kagan's and Motoi's experiments. In our study, we just pretreated BMDMs with dynasore for 1h and then changed with the fresh medium to treat BMDMs using LPS (100ng/ml) for 6h in the absence of dynasore. While in Kagan's and Motoi's experiments, cells were pretreated with dynasore for 30min and then were treated with LPS (3ng/ml, 10ng/ml) or TLR ligands in the presence of dynasore for 40min and 24h, respectively.

3. Furthermore, the authors should explain a reason why TNF α production is decreased by Dynasore treatment in Noc4l-deficient BMDMs. Because TNF α production is induced by cell surface TLR4, TNF α production is expected to decrease in Noc4l-deficient BMDMs, in which TLR4 internalization is enhanced.

Response: This question is similar to the first one. As mentioned above, TRIF also can activate NF- κ B signaling and induce proinflammatory cytokines. Noc4l deficiency enhance TRIF-dependent signaling, including both IRF3 and NF- κ B. After treatment with dynasore to block TRIF signaling, the IFN β and TNF α expression were significantly reduced. These results further confirmed that Noc4l plays an important roles in TLR4 signaling by regulating TRIF-dependent signaling. Additionally, the Kagan, J. C. et al (the first paper reviewer 5 provided) also showed that dynasore treatment reduced LPS-induced inflammatory IL6 expression.

References:

Kawai, T., and Akira, S. (2007). Signaling to NF-kappaB by Toll-like receptors. *Trends Mol Med* 13, 460-469.

Kratz, M., Coats, B.R., Hisert, K.B., Hagman, D., Mutskov, V., Peris, E., Schoenfelt, K.Q., Kuzma, J.N., Larson, I., Billing, P.S., *et al.* (2014). Metabolic dysfunction drives a mechanistically distinct proinflammatory phenotype in adipose tissue macrophages. *Cell Metab* 20, 614-625.

Lancaster, G.I., Langley, K.G., Berglund, N.A., Kammoun, H.L., Reibe, S., Estevez, E., Weir, J., Mellett, N.A., Pernes, G., Conway, J.R.W., *et al.* (2018). Evidence that TLR4 Is Not a Receptor for Saturated Fatty Acids but Mediates Lipid-Induced Inflammation by Reprogramming Macrophage Metabolism. *Cell Metab* 27,

1096-1110 e1095.

Liu, T., Zhang, L., Joo, D., and Sun, S.C. (2017). NF-kappaB signaling in inflammation. *Signal Transduct Target Ther* 2.

Miao, H., Ou, J., Ma, Y., Guo, F., Yang, Z., Wiggins, M., Liu, C., Song, W., Han, X., Wang, M., *et al.* (2014). Macrophage CGI-58 deficiency activates ROS-inflammasome pathway to promote insulin resistance in mice. *Cell Rep* 7, 223-235.

Reilly, S.M., and Saltiel, A.R. (2017). Adapting to obesity with adipose tissue inflammation. *Nat Rev Endocrinol* 13, 633-643.

Senn, J.J. (2006). Toll-like receptor-2 is essential for the development of palmitate-induced insulin resistance in myotubes. *J Biol Chem* 281, 26865-26875.

Wen, H., Gris, D., Lei, Y., Jha, S., Zhang, L., Huang, M.T., Brickey, W.J., and Ting, J.P. (2011). Fatty acid-induced NLRP3-ASC inflammasome activation interferes with insulin signaling. *Nat Immunol* 12, 408-415.

Yamamoto, M., Sato, S., Hemmi, H., Hoshino, K., Kaisho, T., Sanjo, H., Takeuchi, O., Sugiyama, M., Okabe, M., Takeda, K., *et al.* (2003a). Role of adaptor TRIF in the MyD88-independent toll-like receptor signaling pathway. *Science* 301, 640-643.

Yamamoto, M., Sato, S., Hemmi, H., Uematsu, S., Hoshino, K., Kaisho, T., Takeuchi, O., Takeda, K., and Akira, S. (2003b). TRAM is specifically involved in the Toll-like receptor 4-mediated MyD88-independent signaling pathway. *Nat Immunol* 4, 1144-1150.

Yin, J., Wang, Y., Gu, L., Fan, N., Ma, Y., and Peng, Y. (2015). Palmitate induces endoplasmic reticulum stress and autophagy in mature adipocytes: implications for apoptosis and inflammation. *Int J Mol Med* 35, 932-940.

Reviewer comments, fifth round –

Reviewer #3 (Remarks to the Author):

The authors have adequately addressed most of the issues. However, I still feel the manuscript would benefit from a more balanced discussion of the role of LPS in inducing metabolic inflammation. The authors state in both the manuscript and rebuttal that LPS has been accepted as the inducer of inflammation during obesity. However, in the context of adipose tissue, fatty acids, and particularly palmitate would serve as a much more likely candidate to induce macrophage inflammation. The relevance of this pathway has also been shown by the authors in figure 4. Although palmitate levels in the circulation might not be altered, these fatty acids could still drive inflammation in obese adipose tissue. In general, the discussion needs some work with some statements that need to be changed. For example, the following statement is unclear to me: Thus, patients with diseases caused by infection with LPS containing Gram-negative bacteria or gut microbiota, which could further reduce *Noc4l* expression, could be more vulnerable to metabolic disorders.

I think the authors are mixing up two different aspects here. There is a difference between acute inflammatory reactions vs. long term inflammatory responses leading to metabolic complications.

Reviewer 5 (remarks to authors) -

Unfortunately, the reviewer is unable to agree with the authors responses. I agree with the authors that TRIF is required for both cell surface and endosomal TLR4 signals. In contrast, MyD88 is required only for the cell surface TLR4 signal. The authors should carefully look at the Fig. 4B (below) in Motoi's paper, showing that the effect of Dynasore on lipid A responses in BM-macrophages. Apparently, TNF- α is resistant to Dynasore treatment. Of course, IL-6 was also impaired by Dynasore treatment, as the authors pointed out.

This impairment might be due to the toxic effect of Dynasore, because CD14-deficiency impaired TLR4 internalization by LPS, leading to the decrease production of IFN- β , but NOT TNF- α or IL-6 (1)(Fig. 1 shown below). As long as TLR4 is ligated by LPS or lipid A, TLR4 internalization is linked with production of IFN- β , not TNF- α . If TLR4 internalization is enhanced, one expects that TNF- α behaves differently from IFN- β . The authors still need to explain why TNF- α production was increased as much as IFN- β production.

In this context, the conclusion that LPS-dependent TLR4 internalization is enhanced in Noc4I-deficient macrophages is not convincing at all. The authors claimed enhanced LPS-dependent TLR4 internalization in Fig. 5o. However, at least for the reviewer, LPS-dependent decrease in TLR4 does not seem to be strengthened in Noc4I-deficient macrophages. More convincing evidence for impaired TLR4 internalization is required.

Concerning Dynasore treatment, if the authors consider that differences in the Dynasore treatment condition causes opposite results, the authors need to show the results comparing treatment condition.

Even if the TRIF pathway is enhanced in Noc4I-deficient macrophage, the reviewer cannot accept the conclusion that the enhancement in the TRIF pathway is due to enhanced LPS-dependent TLR4 internalization. This is still a possibility the author can describe in Discussion, but not a conclusion.

1. Rajaiah, R., D. J. Perkins, D. D. Ireland, and S. N. Vogel. 2015. CD14 dependence of TLR4 endocytosis and TRIF signaling displays ligand specificity and is dissociable in endotoxin tolerance. *Proc Natl Acad Sci U S A* 112: 8391-8396.

Reviewer #5 (Remarks to the Author):

Q1. In contrast, MyD88 is required only for the cell surface TLR4 signal. The authors should carefully look at the Fig. 4B (below) in Motoi's paper, showing that the effect of Dynasore on lipid A responses in BM-macrophages. Apparently, TNF- α is resistant to Dynasore treatment.

Response: Partially consistent with Reviewer 5's point, we also observed that LPS-stimulated production of TNF α appeared to be "slightly" inhibited by Dynasore in wild-type (WT) BMDMs (Fig.5q, see below). However, this inhibition of Dynasore treatment in WT BMDM was not statistically significant, which is in line with an earlier independent study report (Bowen et al., 2012). (Fig.1, see below).

Fig.5q

Fig.1 cited from (Bowen et al., 2012)

However, a few published earlier studies have highlighted a critical role for the TRAM/TRIF signaling in the induction of interferon-related genes and proinflammatory cytokines expression (Rajaiah et al., 2015; Yamamoto et al., 2003a; Yamamoto et al., 2003b). In these studies, they demonstrated that both MyD88-dependent (TNF- α and IL-6) and TRIF-dependent (IFN- β and IP-10) cytokines induced by LPS were completely inhibited in the macrophages isolated from the TRIF^{-/-} mice (Fig.2 A and B, see below) (Rajaiah et al., 2015; Yamamoto et al., 2003a; Yamamoto et al., 2003b).

Fig.2 Impaired responses to LPS in TRIF-deficient cells. (Rajaiah et al., 2015; Yamamoto et al., 2003a)

In addition, Jiro Sakai and colleagues have found that the activation of the TNF α promoter requires TRIF-dependent signaling by using real-time single cell analysis in macrophages (Sakai et al., 2017).

All together, these results indicated that TRAM/TRIF plays an important role in the production of both MyD88- and TRIF-dependent cytokines/chemokines. Therefore, TNF α is not only a MyD88-dependent cytokine.

Thus, the decreased times of TNF α after Dynasore treatment is much higher in Noc41^{LKO} BMDMs compared with WT macrophages, which supports our point that Noc41 deficiency increases TRIF-dependent signaling.

Q2. Of course, IL-6 was also impaired by Dynasore treatment, as the authors pointed out. This impairment might be due to the toxic effect of Dynasore, because CD14-deficiency impaired TLR4 internalization by LPS, leading to the decrease production of IFN- β , but NOT TNF- α or IL-6 (1)(Fig. 1 shown below).

Response: We cannot fully agree with the Reviewer 5 on this point. A study published in Nature Immunology has shown that dynasore treatment reduced LPS-induced inflammatory IL6 expression (Kagan et al., 2008) (Fig. 3A shown below). Additionally,

even in Motoi's paper (Ref provided by reviewer 5) (Motoi et al., 2014)(Fig.3B shown below), Dynasore treatment also reduced lipid A-induced IL6 expression in macrophages. One still cannot rule out 100% that there could be some toxic effects of Dynasore, because CD14-deficiency impaired TLR4 internalization by LPS, leading to the decrease production of IFN- β , but NOT TNF- α or IL-6 (Rajaiah et al., 2015).

Fig.3 A and 3 B cited from (Kagan et al., 2008; Motoi et al., 2014), respectively

Q3. As long as TLR4 is ligated by LPS or lipid A, TLR4 internalization is linked with production of IFN- β , not TNF- α . If TLR4 internalization is enhanced, one expects that TNF- α behaves differently from IFN- β . The authors still need to explain why TNF- α production was increased as much as IFN- β production.

Response: TRIF activates both type I interferon and proinflammatory cytokine expression in TLR signaling. In addition to the MyD88-dependent pathway, NF- κ B is activated by the TRIF-dependent pathway. An increasing number of studies have proved that TNF α is not only a MyD88-dependent cytokine, but, TRIF also plays an important role in TNF α expression and production (Rajaiah et al., 2015; Yamamoto et al., 2003a; Yamamoto et al., 2003b). Our luciferase assay results confirmed the TRIF-induced NF- κ B activation, and overexpression of Noc4l blocked TRIF-induced NF- κ B activation, but cannot block MyD88-induced NF- κ B activation. We also illustrated here other molecules, which affect proinflammatory cytokines expression (IL6, TNF α) by regulating TRIF-dependent signaling, for example SHP-2 (An et al., 2006), SARM (Carty et al., 2006).

Taken together, our results demonstrated that TRAM/TRIF plays an important role in the production of both MyD88- and TRIF-dependent cytokines/chemokines. Additionally, please see the answer for Question 1 of the reviewer 5, too.

Q4. In this context, the conclusion that LPS-dependent TLR4 internalization is enhanced in Noc4l-deficient macrophages is not convincing at all. The authors claimed enhanced LPS-dependent TLR4 internalization in Fig. 5o. However, at least for the reviewer, LPS-dependent decrease in TLR4 does not seem to be strengthened in Noc4l-deficient macrophages. More convincing evidence for impaired TLR4 internalization is required.

Response: TLR4 surface staining as a directly readout for TLR4 endocytosis is routinely used in a lot of studies (Ghosh et al., 2015; Rajaiah et al., 2015; Tan et al., 2015; Zanoni et al., 2011). In Fig.5o, by using TLR4 surface staining, we clearly showed that TLR4 surface expression was significantly reduced after LPS treatment in Noc41-deficient macrophages compared with WT. Moreover, we also measured the interaction between Noc41 and TLR4 by immunoprecipitation (IP), Proximity Ligation Assay (PLA), and the expressions of their downstream signaling genes. We assumed that these results speak for themselves and provide convincing evidence that Noc41 deficiency enhances TLR4 internalization and activates TRIF-dependent signaling.

Q5. Concerning Dynasore treatment, if the authors consider that differences in the Dynasore treatment condition causes opposite results, the authors need to show the results comparing treatment condition.

Response: According to the reviewer 5's concern, we repeated the dynasore experiment according to Kagan's experiment protocol with slight modification (Kagan et al., 2008), in which cells were pretreated with Dynasore (80 μ M) for 30 min and then were treated with LPS (100ng/ml) for 6 h in the presence of Dynasore. The results are showed below and in line with Kagan and Motoi's results (Kagan et al., 2008; Motoi et al., 2014), which indeed indicate that differences in the Dynasore treatment condition causes different results.

Figure Legend: Relative expression of IFN β of BMDMs from *Noc41^{fl/fl}* and *Noc41^{LKO}* mice pre-treated with Dynasore (80 μ M) for 30 min and stimulated with LPS (100ng/ml) for 6 h in the presence of Dynasore.

Q6. Even if the TRIF pathway is enhanced in *Noc41*-deficient macrophage, the reviewer cannot accept the conclusion that the enhancement in the TRIF pathway is due to enhanced LPS-dependent TLR4 internalization. This is still a possibility the author can describe in Discussion, but not a conclusion.

Response: In line with reviewer 5's suggestion, we decided to tune down our conclusion that Noc4l-deficient in macrophage enhanced LPS-dependent TLR4 internalization to activate TRIF pathway.

In our revised manuscript, we changed and mentioned this in the Discussion part as follows: "Overall, we demonstrated here a possible mechanism explaining the molecular function of NOC4L in insulin sensitivity. NOC4L was localized in the endosome of BMDMs and played a potential role in TRIF-dependent pathway by directly interacting with TLR4. It might inhibit the endocytosis of TLR4, thus reducing the production of IFN β and proinflammatory cytokines, ameliorating the LSI and IR". (please see line 487-491, labeled red).

Reviewer #3 (Remarks to the Author):

The authors have adequately addressed most of the issues. However, I still feel the manuscript would benefit from a more balanced discussion of the role of LPS in inducing metabolic inflammation. The authors state in both the manuscript and rebuttal that LPS has been accepted as the inducer of inflammation during obesity. However, in the context of adipose tissue, fatty acids, and particularly palmitate would serve as a much more likely candidate to induce macrophage inflammation. The relevance of this pathway has also been shown by the authors in figure 4. Although palmitate levels in the circulation might not be altered, these fatty acids could still drive inflammation in obese adipose tissue. In general, the discussion needs some work with some statements that need to be changed. For example, the following statement is unclear to me: Thus, patients with diseases caused by infection with LPS containing Gram-negative bacteria or gut microbiota, which could further reduce Noc4l expression, could be more vulnerable to metabolic disorders.

I think the authors are mixing up to different aspects here. There is a difference between acute inflammatory reactions vs. long term inflammatory responses leading to metabolic complications.

Response: We thank so much for Reviewer 3's good suggestion.

We addressed the role of LPS in inducing metabolic inflammation in the discussion section of the revised manuscript in the red labeled context **line 430-443, as follows:** "Diet-induced metabolic endotoxemia is an important factor for the low-grade inflammation in obesity and metabolic diseases²². Structural changes to the intestinal epithelium in response to dietary alterations allow LPS produced from the gut microbiota to enter the bloodstream, resulting in an increase in the plasma levels of LPS. LPS binding to its receptor TLR4 induces the production of pro-inflammatory cytokines and, hence, leading to low-grade systemic inflammation. In our case, consistent with HFD-induced obesity model we observed that glucose tolerance of Noc4l^{LKO} mice was significantly impaired relative to the Noc4l^{fl/fl} mice with a nonlethal dose of LPS injection intraperitoneally. Besides LPS, the serum levels of saturated free fatty acids

such as palmitate are elevated in obesity, which might be an important player to activate proinflammatory pathways and induce cytokines expression. There exist a number of mechanisms about how palmitate regulates obesity-associated inflammation⁵³⁻⁵⁵. However, recent studies showed that palmitate does not directly bind TLR4, although Tlr4^{-/-} attenuates inflammation in palmitate-treated macrophages⁵⁶. So how Noc4l regulates palmitate-associated signaling is still required to be further studied.”

We deleted the statement “Thus, patients with diseases caused by infection with LPS containing Gram-negative bacteria or gut microbiota, which could further reduce Noc4l expression, could be more vulnerable to metabolic disorders” in the revised manuscript, red labeled **line 450-452**.

References:

An, H., Zhao, W., Hou, J., Zhang, Y., Xie, Y., Zheng, Y., Xu, H., Qian, C., Zhou, J., Yu, Y., *et al.* (2006). SHP-2 phosphatase negatively regulates the TRIF adaptor protein-dependent type I interferon and proinflammatory cytokine production. *Immunity* 25, 919-928.

Bowen, W.S., Minns, L.A., Johnson, D.A., Mitchell, T.C., Hutton, M.M., and Evans, J.T. (2012). Selective TRIF-dependent signaling by a synthetic toll-like receptor 4 agonist. *Sci Signal* 5, ra13.

Carty, M., Goodbody, R., Schroder, M., Stack, J., Moynagh, P.N., and Bowie, A.G. (2006). The human adaptor SARM negatively regulates adaptor protein TRIF-dependent Toll-like receptor signaling. *Nat Immunol* 7, 1074-1081.

Ghosh, M., Subramani, J., Rahman, M.M., and Shapiro, L.H. (2015). CD13 restricts TLR4 endocytic signal transduction in inflammation. *J Immunol* 194, 4466-4476.

Kagan, J.C., Su, T., Horng, T., Chow, A., Akira, S., and Medzhitov, R. (2008). TRAM couples endocytosis of Toll-like receptor 4 to the induction of interferon-beta. *Nat Immunol* 9, 361-368.

Motoi, Y., Shibata, T., Takahashi, K., Kanno, A., Murakami, Y., Li, X., Kasahara, T., and Miyake, K. (2014). Lipopeptides are signaled by Toll-like receptor 1, 2 and 6 in endolysosomes. *Int Immunol* 26, 563-573.

Rajaiah, R., Perkins, D.J., Ireland, D.D., and Vogel, S.N. (2015). CD14 dependence of TLR4 endocytosis and TRIF signaling displays ligand specificity and is dissociable in endotoxin tolerance. *Proc Natl Acad Sci U S A* 112, 8391-8396.

Sakai, J., Cammarota, E., Wright, J.A., Cicuta, P., Gottschalk, R.A., Li, N., Fraser, I.D.C., and Bryant, C.E. (2017). Lipopolysaccharide-induced NF-kappaB nuclear translocation is primarily dependent on MyD88, but TNFalpha expression requires TRIF and MyD88. *Sci Rep* 7, 1428.

Tan, Y., Zanoni, I., Cullen, T.W., Goodman, A.L., and Kagan, J.C. (2015). Mechanisms of Toll-like Receptor 4 Endocytosis Reveal a Common Immune-Evasion Strategy Used by Pathogenic and Commensal Bacteria. *Immunity* 43, 909-922.

Yamamoto, M., Sato, S., Hemmi, H., Hoshino, K., Kaisho, T., Sanjo, H., Takeuchi, O., Sugiyama, M., Okabe, M., Takeda, K., *et al.* (2003a). Role of adaptor TRIF in the MyD88-independent toll-like receptor signaling pathway. *Science* 301, 640-643.

Yamamoto, M., Sato, S., Hemmi, H., Uematsu, S., Hoshino, K., Kaisho, T., Takeuchi, O., Takeda, K., and Akira, S. (2003b). TRAM is specifically involved in the Toll-like receptor 4-mediated MyD88-independent signaling pathway. *Nat Immunol* 4, 1144-1150.

Zanoni, I., Ostuni, R., Marek, L.R., Barresi, S., Barbalat, R., Barton, G.M., Granucci, F., and Kagan, J.C. (2011). CD14 controls the LPS-induced endocytosis of Toll-like receptor 4. *Cell* 147, 868-880.

Reviewer comments, sixth round –

Reviewer #5 (Remarks to the Author):

No further comment.

REVIEWERS' COMMENTS

Reviewer #5 (Remarks to the Author):

No further comment.

Response: Thanks very much.